# Multipotent neural stem cells originating from neuroepithelium exist outside the mouse central nervous system

Dong Han [1,2] ✉, Wan Xu[2], Hyun-Woo Jeong [3], Hongryeol Park[3], Kathrin Weyer[4], Yaroslav Tsytsyura[5], Martin Stehling[6], Guangming Wu[1,7,8], Guocheng Lan[2], Kee-Pyo Kim [1,9], Henrik Renner[1], Dong Wook Han[10], Yicong Chen[2], Daniela Gerovska[11], Marcos J. Araúzo-Bravo[11,12], Jürgen Klingauf [5,13], Jens Christian Schwamborn [4], Ralf H. Adams [3,14], Pentao Liu [2,15] ✉ & Hans R. Schöler [1] ✉

Conventional understanding dictates that mammalian neural stem cells (NSCs) exist only in the central nervous system. Here, we report that peripheral NSCs (pNSCs) exist outside the central nervous system and can be isolated from mouse embryonic limb, postnatal lung, tail, dorsal root ganglia and adult lung tissues. Derived pNSCs are distinct from neural crest stem cells, express multiple NSC-specific markers and exhibit cell morphology, self-renewing and differentiation capacity, genome-wide transcriptional profile and epigenetic features similar to control brain NSCs. pNSCs are composed of Sox1+ cells originating from neuroepithelial cells. pNSCs in situ have similar molecular features to NSCs in the brain. Furthermore, many pNSCs that migrate out of the neural tube can differentiate into mature neurons and limited glial cells during embryonic and postnatal development. Our discovery of pNSCs provides previously unidentified insight into the mammalian nervous system development and presents an alternative potential strategy for neural regenerative therapy.

At the onset of vertebrate neurogenesis, neuroepithelial cells (NECs) in the neural tube, the earliest NSCs, switch their identity and generate radial glial cells that differentiate into neurons and glial cells in the central nervous system (CNS)[1]. In the adult mammalian brain, NSCs exist mainly in two regions: the subgranular zone in the dentate gyrus of the hippocampus and the subventricular zone, adjacent to the lateral ventricles[2], where neurogenesis persists throughout postnatal life[3]. NSCs can also be extracted from adult CNS regions where neurogenesis is not apparent, then expanded and differentiated into neurons and glia in culture[4–7]. Unlike the CNS, the peripheral nervous system (PNS) is derived from the neural crest cells (NCCs), a collection of progenitors that migrates out of the dorsal side of the neural tube[8]. Multipotent neural crest stem cells (NCSCs), rather than NSCs, are found outside

the CNS[9,10]. NCSCs, with a limited capacity for self-renewal, are different from NSCs in gene expression and differentiation potential[11–13]. Taken together, these results demonstrate that mammalian NSCs exist only in the CNS. Whether NSCs exist outside the CNS was unknown to date.

Here, we show that multipotent pNSCs with CNS NSC properties exist outside the mouse CNS, contrary to the long-standing dogma. Derived pNSCs closely resemble brain tissue-derived NSCs and are distinct from NCSCs. pNSCs are composed of Sox1+ cells and originate from NECs. pNSCs in situ have similar molecular features to brain NSCs. Furthermore, many pNSCs that migrate out of the neural tube can differentiate into neurons and limited glial cells during embryonic and postnatal development. Our finding of pNSCs provides previously unidentified insight into mammalian nervous system development.

## Results

### Derivation of peripheral NSCs by low-pH treatment

Initially, we attempted to repeat the experiment that described the stimulus-triggered acquisition of pluripotency based on low-pH treatment of somatic cells. However, regardless of which culture conditions and tissues we used, we failed to derive pluripotent cells. Finally, the respective papers were retracted a couple of months after they were published (https://www.nature.com/news/stap-retracted-1.15488), which stopped any further attempts at deriving pluripotent cells this way. However, to our surprise, with one culture condition, we were able to obtain a rare population of cells that shared features with NSCs. While most of the cells grew like fibroblasts and were negative for Sox2 and Olig2 (Extended Data Fig. 1a,b), we obtained NSC-like cell clusters when we used a low-pH medium to treat mouse passage-4 embryonic limb cells and passage-2 adult lung cells (Fig. 1a–d). We tentatively named these cells as 'embryonic limb low-pH NSCs' (ellNSCs) and 'adult lung low-pH NSCs' (allNSCs), respectively. EllNSCs and allNSCs were similar to control brain NSCs[14] or transcription factor (TF)-induced NSCs (iNSCs)[15] in morphology, passaging, proliferation, lack of teratoma formation, marker gene expression and genome-wide gene expression pattern (Fig. 1e–j and Extended Data Fig. 1c–f). DNA methylation analysis on the regulatory regions of NSC marker *Nestin* and the promoter region of fibroblast marker *Col1a1* revealed similar methylation profiles between low-pH-derived NSCs (ldNSCs) and control NSCs (Fig. 1k,l). These data suggest that ldNSCs have features in common with NSCs.

### Low-pH-derived pNSCs are multipotent

We next examined the differentiation capacity of ldNSCs to three main neural lineages in vitro. Under various conditions, ellNSCs and allNSCs differentiated into astrocytes, oligodendrocytes and neurons with similar differentiation efficiencies to brain NSCs (Fig. 2a–d and Extended Data Fig. 1g). As expected, most differentiated neurons were either GABAergic, glutamatergic or to a lesser extent cholinergic. Neurons from all three NSC types expressed vesicular glutamate transporter 1 (vGluT1), suggesting synapse development (Fig. 2e).

Neurons derived from ldNSCs also possess neuronal membrane properties. Both voltage-gated Na+ inward and K+ outward currents were elicited in ldNSC- and control NSC-derived neurons and the inward current could be blocked by tetrodotoxin (TTX) application (Fig. 2f and Extended Data Fig. 1h). Although spontaneous firing of action potentials was not observed, the neurons were excitable and elicited action potentials by current injections, demonstrating excitability (Fig. 2g). These data show that neurons derived from ldNSCs acquired electrical properties akin to those from control NSCs.

A hallmark of NSCs is their ability to survive and differentiate in vivo when transplanted into the adult mouse brain. Therefore, we transplanted karyotyped allNSCs labelled with mCherry into the cortex of adult mice. Analysis after 6 weeks revealed survival of grafts, with complete survival of the host and absence of tumour formation (Fig. 2h and Extended Data Fig. 1i,j). Like control NSCs, grafted allNSCs were able to differentiate into all three neural lineages, demonstrating that allNSCs exhibit similar characteristics as control NSCs after transplantation (Fig. 2i and Extended Data Fig. 1k–m).

### Derivation of pNSCs without low-pH treatment

Previously, NSC-like clusters could not be obtained from passage-4 embryonic limb cells without low-pH treatment (Fig. 1b, 'no treatment'). If endogenous NSCs were present in the limb tissues, culturing and passaging these cells in MEF medium that contains serum but not EGF and bFGF might induce their quiescence or differentiation. To test whether endogenous NSCs exist, freshly isolated embryonic limb cells were directly cultured in NSC medium for 3 weeks. Indeed, we obtained NSC-like cells (Fig. 3a–c and Extended Data Fig. 2a).

To assess whether pNSCs exist in adult lung tissues, we used a transgenic mouse line that expresses GFP under NSC-specific Nestin regulatory elements (Nes-GFP)[16]. Adult lung Nes-GFP+ cells were sorted and cultured directly in NSC medium for 3 weeks (Fig. 3d). While most GFP+ cells were fibroblast-like cells (Extend Data Fig. 2b), we observed GFP+ NSC-like clusters (Fig. 3e,f). Similar results were observed from postnatal tail Nes-GFP+ cells (Fig. 3g–i). Nes-GFP+ pNSCs were similar to brain NSCs in self-renewal, lack of teratoma formation, marker gene expression, genome-wide gene expression pattern and DNA methylation pattern on *Nestin* and *Col1a1* loci (Fig. 3j–p and Extended Data Fig. 2c). Together, these results demonstrate that pNSCs exist outside the CNS and can be isolated in vitro without low-pH treatment.

### pNSCs derived without low-pH treatment are multipotent

We next examined the multipotency of Nes-GFP+ pNSCs. We generated clonal brain NSC, adult lung and postnatal tail Nes-GFP+ pNSC lines (Fig. 4a). Under various differentiation conditions in vitro, lung and tail pNSCs could be differentiated into neurons, astrocytes and oligodendrocytes, with similar efficiencies to brain NSCs (Fig. 4b,c).

Next, clonal Nes-GFP+ adult lung and postnatal tail pNSCs labelled with H2B-tdTomato (Extended Data Fig. 2d) were transplanted into the cortex of postnatal day 1 (P1) mouse brain. All mice survived with no tumourigenesis. Immunohistochemistry after 6 weeks revealed that the engraftments did not express Nestin or cell-cycle marker Ki67 (Extended Data Fig. 2e). Notably, transplanted cells differentiated into all three neural lineages in vivo with similar differentiation efficiencies to brain NSCs (Fig. 4d–g). These results demonstrate that pNSCs derived without low-pH treatment exhibit characteristics similar to brain NSCs when transplanted into the mouse brain.

To exclude cell fusion between transplanted cells and endogenous cells, we transplanted wild-type postnatal tail pNSCs constitutively expressing Cre recombinase and H2B-GFP (Extended Data Fig. 2f,g) into the brain of P1 R26-loxp-Stop-loxp-RFP mouse[17]. If fusion occurred, fused cells would become GFP+/RFP+ after Cre excision of the loxP-flanked stop sequences (Extended Data Fig. 2h). We were only able to observe GFP+ but not GFP+/RFP+ neural derivatives (Extended Data Fig. 2i), excluding the possibility of cell fusion.

### Derived pNSCs do not originate from NCCs

NCSCs can be derived outside the CNS. To assess whether derived pNSCs are NCSCs, we stained Nes-GFP+ adult lung and postnatal tail pNSCs with NSC- and neural crest (NC)-specific markers. pNSCs expressed Sox2 and NSC-specific marker Olig2, but not NC-specific markers Sox10 and p75 (refs. 9,18). In contrast, NCSCs from adult wild-type (WT) mouse dorsal root ganglia (DRG) expressed Sox2, Sox10 and p75, but not Olig2 (Fig. 5a). Upon differentiation, pNSCs efficiently differentiated into

---

**Fig. 1 | Derivation and characterization of ellNSCs and allNSCs. a**, Schematic for deriving ellNSCs and morphology of early cluster. **b**, Number of NSC-like clusters obtained with different pH conditions from passage-4 embryonic limb cells; *n* = 12. **c**, Schematic for deriving allNSCs and morphology of early cluster. **d**, Number of NSC-like clusters obtained with different pH conditions from passage-2 adult lung cells; *n* = 10. **e**, LdNSCs stably maintained for 90 passages, representative of three biological replicates. **f**, Proliferation rates of passage-90 ldNSCs and control NSCs. The data represent mean ± s.d. (*n* = 3 biological replicates). **g**, Immunofluorescence of NSC markers in ellNSCs and allNSCs, representative of three biological replicates. Scale bar, 50 μm. **h**, RT–qPCR of NSC markers in samples. All data are calibrated to brain NSCs, whose expression is considered to be 1 for all genes. The data represent mean ± s.d. (*n* = 3 biological replicates). **i**, Heatmap of global gene expression across samples. Colour bar indicates gene expression in log_2 scale. *n* = 2 biological replicates, all replicates are shown. **j**, Hierarchical clustering of the cell lines based on **i. k,l**, Bisulfite sequencing PCR on the second intron of *Nestin* (**k**) and promoter of *Col1a1* (**l**) in samples. Open and filled circles represent unmethylated and methylated CpGs, respectively.

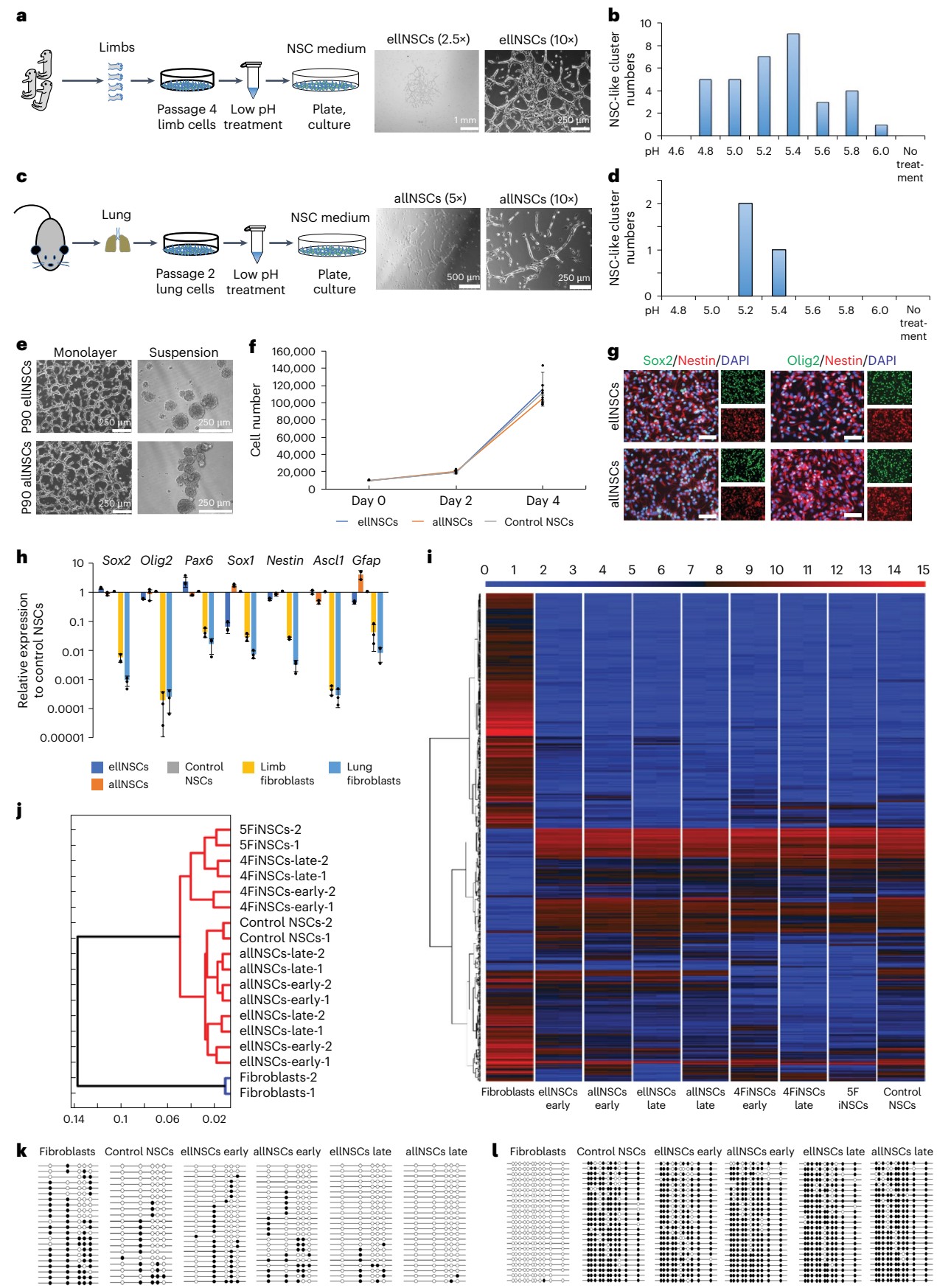

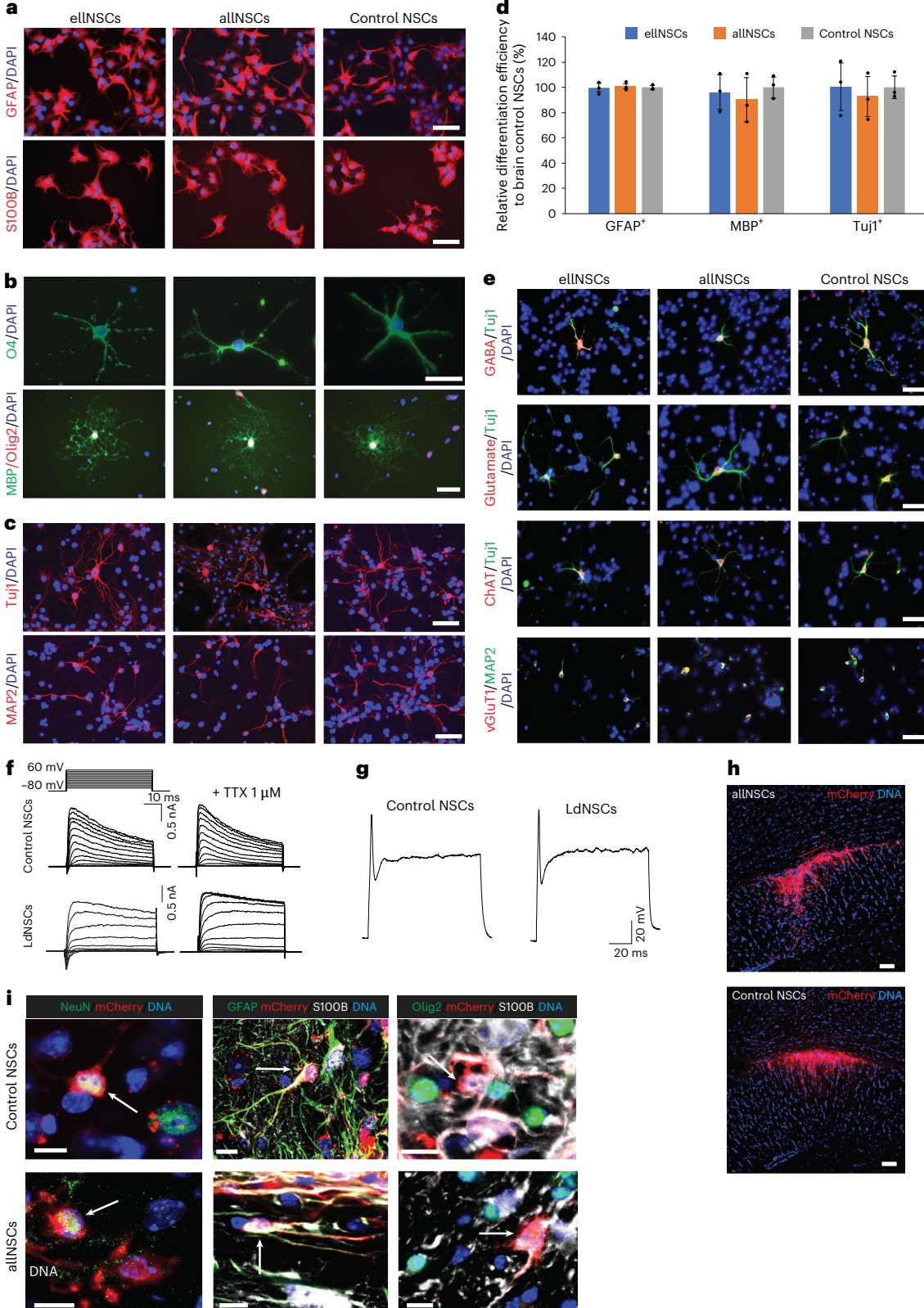

**Fig. 2 | Multipotency of ldNSCs in vitro and in vivo. a–c**, ellNSCs, allNSCs and control NSCs differentiated into astrocytes (GFAP⁺/S100B⁺) (**a**), oligodendrocytes (O4⁺/Olig2⁺/MBP⁺) (**b**) and neurons (Tuj1⁺/MAP2⁺) (**c**). Scale bar, 50 µm. **d**, In vitro differentiation efficiencies in samples, normalized to brain NSCs, determined via immunostaining with Tuj1 (neurons), GFAP (astrocytes) and MBP (oligodendrocytes). The data represent mean ± s.d. (*n* = 3 biological replicates). **e**, Immunofluorescence images of neurons derived from samples. Scale bar, 50 µm. ChAT, choline acetyltransferase; vGluT1, vesicular glutamate transporter

type 1. **f**, Voltage-clamp recordings from ldNSC and control NSC neurons in response to increasing voltage pulses. **g**, Current-clamp mode action potential of ldNSC and control NSC neurons in response to current injection (200 pA). **h**, Immunohistochemistry of transplanted allNSCs and control NSCs after 6 weeks. *n* = 3 mice each. Scale bar, 50 µm. **i**, Transplanted allNSCs and control NSCs differentiated into mature neurons (NeuN⁺/mCherry⁺), astrocytes (GFAP⁺/S100B⁺/mCherry⁺) and oligodendrocytes (Olig2⁺/S100B⁺/mCherry⁺), indicated by white arrows. Data represent three biological replicates. Scale bar, 10 µm.

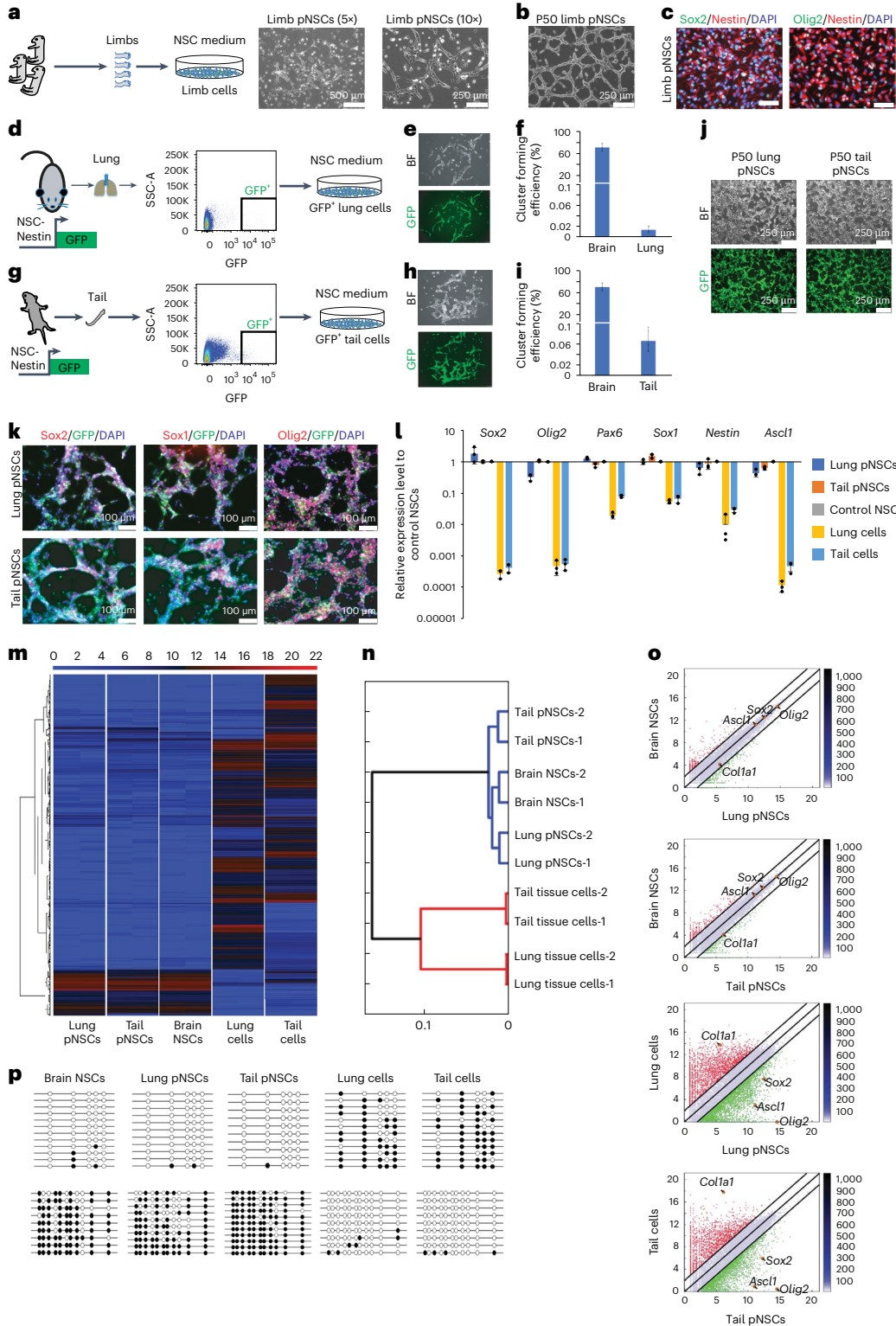

**Fig. 3 | Derivation of pNSCs without low-pH treatment. a**, Schematic and morphology of embryonic limb pNSC derivation without low-pH treatment. **b**, Limb pNSCs stably maintained for 50 passages. **c**, Immunofluorescence of limb pNSCs. Scale bar, 50 μm. **d**, FACS strategy for deriving adult lung pNSCs without low-pH treatment from Nes-GFP mouse. **e**, Morphology of Nes-GFP⁺ cells derived primary lung pNSC cluster. BF, bright-field. **f**, NSC-like cluster forming efficiency from sorted Nes-GFP⁺ lung cells. The data represent mean ± s.d. (*n* = 3 biological replicates). **g**, FACS strategy for deriving postnatal tail pNSCs without low-pH treatment from Nes-GFP mouse. **h**, Morphology of Nes-GFP⁺ cells derived primary tail pNSC cluster. **i**, NSC-like cluster forming efficiency from sorted Nes-GFP⁺ tail cells. The data represent mean ± s.d. (*n* = 3 biological replicates).

**j**, Lung and tail pNSCs stably maintained for 50 passages. **k**, Immunofluorescence of NSC markers in lung and tail pNSCs. **l**, RT–qPCR analysis across samples. All data are calibrated to brain NSCs, whose expression is considered to be 1 for all genes. The data represent mean ± s.d. (*n* = 3 biological replicates). **m**, Heatmap of global gene expression pattern of the samples. *n* = 2 biological replicates, all replicates are shown. **n**, Hierarchical clustering of the cell lines based on **m**. **o**, Pairwise scatter-plots of lung and tail pNSCs versus brain NSCs and tissue cells. *n* = 2 biological replicates; the depicted results are an integration of data derived from all different biological samples. **p**, Bisulfite sequencing PCR on *Nestin* (up) and *Col1a1* (down) in samples.

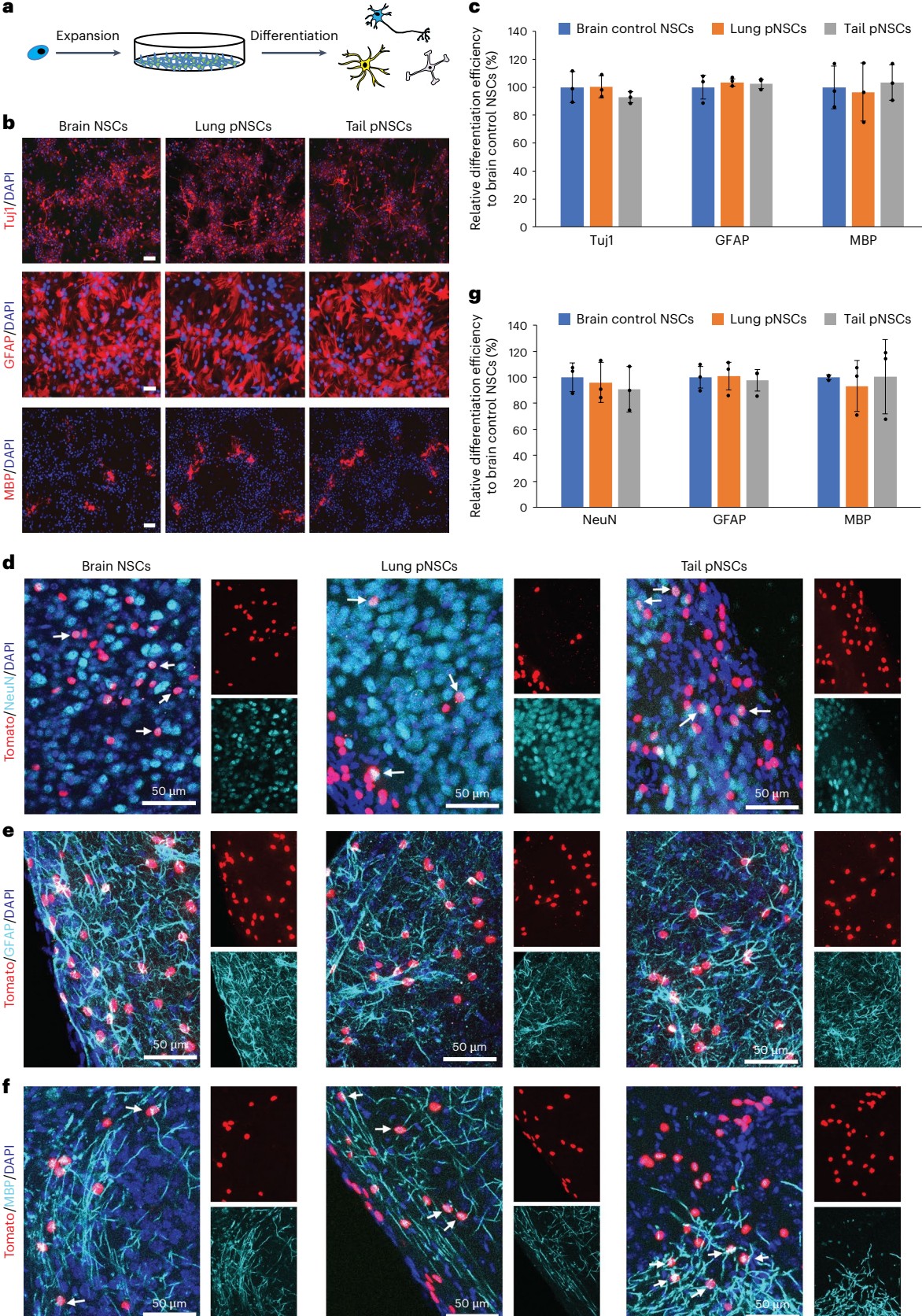

**Fig. 4 | Multipotency of pNSCs derived without low-pH treatment. a**, Scheme of the clonal pNSC line establishment and differentiation. **b**, Lung and tail pNSCs and brain NSCs differentiated into neurons (Tuj1), astrocytes (GFAP) and oligodendrocytes (MBP). Scale bar, 50 μm. **c**, In vitro differentiation efficiencies in samples, normalized to brain NSCs. The data represent means ± s.d. (*n* = 3

biological replicates). **d**–**f**, Transplanted pNSCs and brain NSCs differentiated into mature neurons (**d**), astrocytes (**e**) and oligodendrocytes (**f**). White arrows indicate positive cells. Scale bar, 50 μm. **g**, In vivo differentiation efficiencies in samples, normalized to brain NSCs. The data represent mean ± s.d. (*n* = 3 biological replicates).

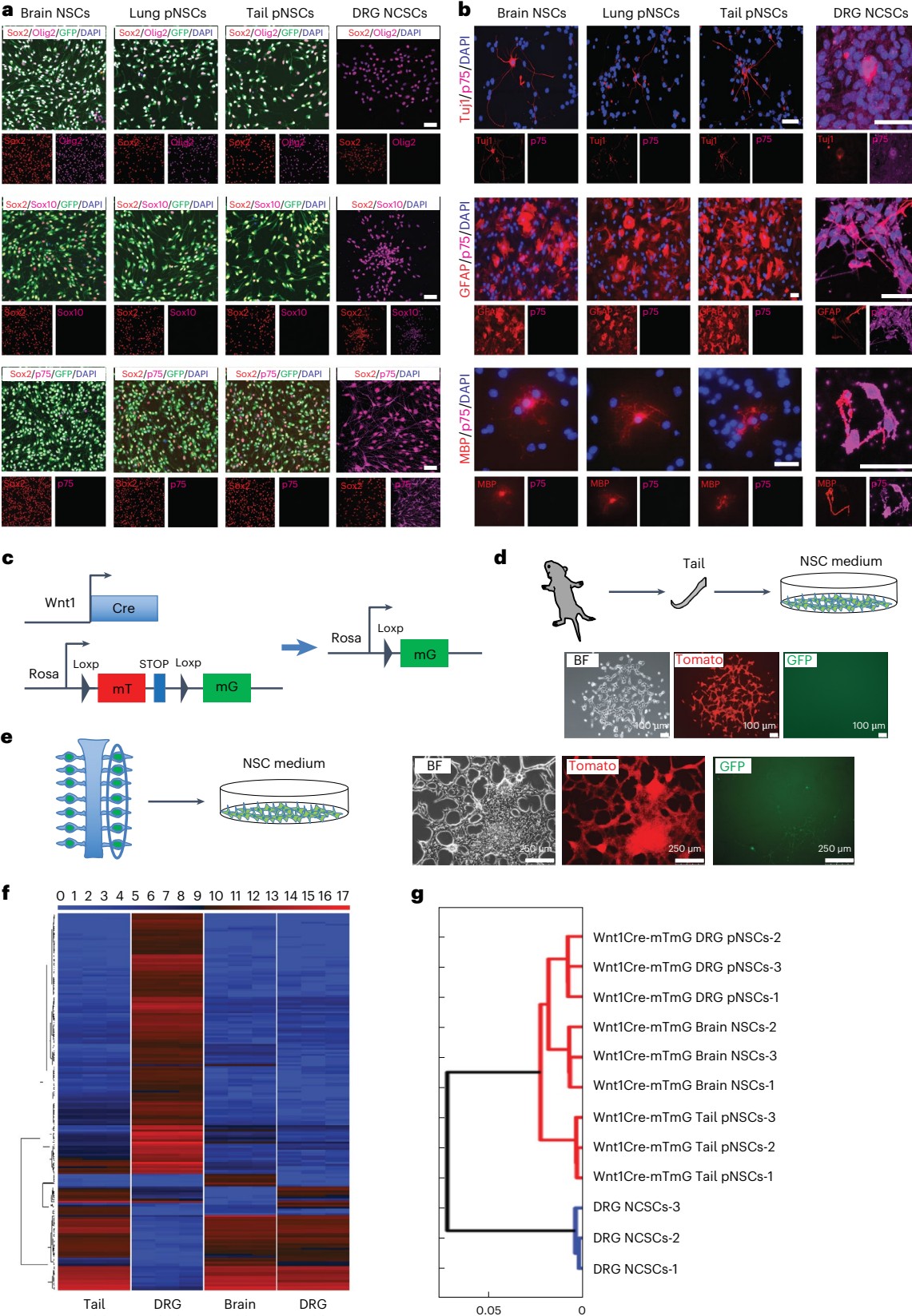

**Fig. 5 | pNSCs are distinct from NCSCs and do not originate from NCCs.**
**a**, Immunofluorescence of brain NSCs, lung pNSCs, tail pNSCs and DRG NCSCs. Data represent three biological replicates. Scale bar, 50 μm.
**b**, Immunofluorescence of differentiated neurons (Tuj1), astrocytes (GFAP) and oligodendrocytes (MBP) from brain NSCs, lung pNSCs, tail pNSCs and DRG

NCSCs. Data represent three biological replicates. Scale bar, 50 μm. **c**, Genetic tracing of NCCs and their progenies. **d,e**, Schematic for deriving postnatal tail (**d**) and DRG pNSCs (**e**) and morphology of primary pNSC clusters. **f**, Heatmap of global gene expression pattern across samples. *n* = 3 biological replicates; all replicates are shown. **g**, Hierarchical clustering of the cell lines based on **f**.

p75 negative Tuj1⁺ neurons, GFAP⁺ astrocytes and myelin basic protein (MBP)⁺ oligodendrocytes. In contrast, it was very difficult to observe Tuj1⁺ and MBP⁺ cells in NCSCs differentiation. Furthermore, NCSCs differentiated cells expressed p75 (Fig. 5b). These results demonstrate that pNSCs are different from NCSCs.

To further genetically confirm that pNSCs are not NCSCs and not from NCCs, we crossed Wnt1-Cre mice[19] with R26-mTmG mice[20] to label NCCs and their progenies (Fig. 5c). First, we tried to obtain pNSCs from postnatal Wnt1-Cre-mTmG tail tissues. Because we did not know whether pNSCs are derived from NCCs, we did not sort the GFP⁻ or GFP⁺ cells and simply plated out all the cells from tail tissue at low density and cultured them in NSC medium. Under these conditions, the GFP⁺ and GFP⁻ cells had exactly the same chance of forming NSC-like clusters. Although Wnt1-Cre may not label all NC-derived cells, the vast majority of NC-derived cells were GFP⁺. Notably, we derived only Tomato⁺/GFP⁻ pNSCs (Fig. 5d). We were also able to derive pNSCs from sorted GFP⁻ cells, indicating that pNSCs are not NCSCs and not from NCCs (Extended Data Fig. 3a). Similar results were obtained from postnatal DRG tissue. As there are many NC-derived cells in the DRG tissue, we also observed some GFP⁺ cells in the culture, but these GFP⁺ cells did not form the NSC-like clusters and were lost within several passages (Fig. 5e). Furthermore, we sorted the GFP⁺ cells from the DRG, treated them with low-pH medium, and cultured them in NSC medium. No NSC-like cluster formed (Extended Data Fig. 3b), strongly arguing against any trans- or de-differentiation of DRG GFP⁺ cells into pNSCs. The Tomato⁺/GFP⁻ pNSCs also share many features with brain NSCs (Fig. 5f,g and Extended Data Fig. 3c–i). Together, these results demonstrate that pNSCs do not originate from NCCs and are distinct from NCSCs.

## pNSCs are composed of Sox1⁺ cells

To further characterize the identity of in vivo pNSCs, we sought to perform single-cell transcription analyses. To enrich NSCs, we used a Nes-GFP transgenic mouse line. We performed single-cell RNA sequencing (scRNA-seq) on adult brain and lung Nes-GFP⁺ cells. Brain Nes-GFP⁺ cells were separated into ten different cell-type clusters and were identified using well-characterized markers. Abundant cell populations include quiescent NSCs (qNSCs), active NSCs (aNSCs), neuroblasts (NBs), oligodendrocyte progenitor cells (OPCs) and mature oligodendrocytes (mOLs). Other smaller populations included mural cells, endothelial cells (EC), ependymal cells (Epend), fibroblasts (Fibro), and microglia cells (Micro) (Fig. 6a, Supplementary Fig. 1a,b and Table 1). Pseudotemporal ordering[21] on single cells of qNSCs, aNSCs and NBs showed the developmental trajectory from qNSCs to aNSCs to NBs (Supplementary Fig. 1c,d). The cell populations and molecular features of the NSCs identified by our scRNA-seq analysis are consistent with those of previous reports[22–25].

We then examined lung Nes-GFP⁺ cells and identified 11 clusters. Most cells were non-neural lineage cells: pericytes (PCs), myofibroblasts (MyoFBs), matrix-fibroblasts (MatrixFBs), ECs, smooth muscle cells (SMC), epithelial cells (Epi) and lymphocytes (Lym) (Fig. 6b, Supplementary Fig. 1e,f and Table 2), indicating unspecific labelling of the Nes-GFP in lung tissues. Only one small cluster of cells expressed many NSC markers (Supplementary Fig. 2a,b). Nervous system development, neurogenesis and gliogenesis appeared in the

top Gene Ontology (GO) terms, confirming that this cell cluster is composed of NSCs (Supplementary Fig. 2c,d). However, several NSC markers expressed in this cluster cells are also expressed in NC-derived cells and many Schwann cell markers were also observed in these cells (Supplementary Fig. 2e), indicating that these cells could be a mixture of different cell types. We tentatively referred to these cells as neural cells. Unfortunately, the limited cell number prevented us from any further subcluster analysis.

To improve the scRNA-seq data of pNSCs, we used a Sox2-GFP knock-in (KI) mouse line. Only Sox2-GFP⁺ cells were able to form pNSC colonies, with the postnatal lung demonstrating higher efficiency (Fig. 6c and Extended Data Fig. 4a). We used postnatal lung cells for further experiments to increase the possibility of capturing pNSCs. Derived Sox2-GFP⁺ pNSCs could be maintained in passage, expressed NSC but not NC markers, showed similar transcriptome pattern to brain NSCs, and could differentiate into neurons, astrocytes and oligodendrocytes in vitro (Fig. 6d–g and Extended Data Fig. 4b,c).

Next, we performed scRNA-seq on postnatal lung Sox2-GFP⁺ cells. We identified 12 clusters (club cells (CC), basal cells (BC), Macrophage 1 (Macro1), Macrophage 2 (Macro2), myeloid cells, ciliated epithelial cells (ciliated Epi), Schwann cells (SC), fibroblasts (FB), ECs, neural cells and muscle cells (MC)) (Fig. 6h, Extended Data Fig. 4d and Supplementary Table 3). Of note, in the neural cell cluster, we found that some cells co-expressed *Sox1* and *Sox2* (Fig. 6i). NSCs and many other stem/progenitor cells express Sox2 (refs. 26–28) and Sox1 is the earliest known specific marker of the NSCs in the mouse embryo[29,30]. These results prompted us to investigate these double-positive cells in lung and tail tissues. In the postnatal and adult lung, the Sox1⁺/Sox2⁺ cells mainly distributed in the big bronchi epithelial cell wall (Fig. 6j and Extended Data Fig. 4e). In the postnatal mouse tail tissue, Sox1⁺/Sox2⁺ cells distributed under the skin as a short tube structure approximately in the middle of the tail in the longitudinal axis (Fig. 6k). Notably, we did not observe Sox1⁺ cells in the postnatal heart, indicating that pNSCs are not distributed in the entire mouse body (Extended Data Fig. 4f).

To confirm pNSCs are from Sox1⁺ cells, GFP⁺ and GFP⁻ cells from the postnatal lung, tail and DRG of Sox1-GFP KI transgenic mouse[30] were sorted out and cultured separately in NSC medium for 3 weeks. We observed NSC-like cluster cells only from GFP⁺ cells, but not from GFP⁻ cells (Fig. 6l,m and Extended Data Fig. 4g). Unexpectedly, the primary lung, tail and DRG pNSCs clusters from Sox1-GFP⁺ cells lost GFP signal, which was recapitulated in postnatal brain Sox1-GFP⁺ cells (Extended Data Fig. 4h), indicating the silencing of GFP signalling during in vitro culture[14]. Of note, pNSCs derived from Sox1-GFP⁺ cells expressed Sox1 (Extended Data Fig. 4i). pNSCs from Sox1-GFP⁺ cells were similar to brain NSCs in passaging, marker gene expression and differentiation ability (Extended Data Fig. 4i–m). Together, these data demonstrate that pNSCs are indeed composed of Sox1⁺ cells.

To reveal the intrinsic NSC properties of Sox1⁺ pNSCs in vitro without expansion, we isolated postnatal lung and tail Sox1-GFP⁺ cells and directly cultured them in a differentiation medium. Sox1-GFP⁺ cells without expansion could differentiate into Tuj1⁺ neurons, GFAP⁺ astrocytes and MBP⁺ oligodendrocytes, confirming their intrinsic NSC properties. The differentiated cells did not express p75, indicating that pNSCs do not differentiate into the NC lineage in vitro (Extended Data Fig. 4n).

**Fig. 6 | pNSCs are composed of Sox1⁺ cells. a,b,** Uniform manifold approximation and projection (UMAP) of 4,640 adult brain (**a**) and 5,981 lung (**b**) Nes-GFP⁺ cells. Colours represent cell population clusters. *n* = 1 biological replicate. **c,** FACS strategy for GFP⁺ and GFP⁻ cells from postnatal Sox2-GFP mouse lung tissue and morphologies of a primary lung pNSC cluster and GFP⁻ cells after culture. **d,** Sox2-GFP⁺ lung pNSCs stably maintained for 50 passages. **e,** Immunofluorescence of brain NSCs and lung pNSCs from Sox2-GFP mouse. Scale bar, 50 μm. **f,** Heatmap of global gene expression pattern across samples. *n* = 3 biological replicates for brain NSCs and lung pNSCs. *n* = 2 biological replicates for lung cells.

All replicates are shown. **g,** Hierarchical clustering of the cell lines based on **f. h,** UMAP of 28,170 postnatal lung Sox2-GFP⁺ cells. Colours represent cell population clusters. *n* = 1 biological replicate. **i,** Expression of *Sox1* and *Sox2* in postnatal lung Sox2-GFP⁺ neural cells cluster. **j,k,** Immunohistochemistry of WT postnatal lung (**j**) and tail (**k**) tissues. Data represent three biological replicates. Scale bar, 50 μm. **l,m,** FACS strategy for deriving lung (**l**) and tail pNSCs (**m**) from postnatal Sox1-GFP mouse, and morphologies of primary lung (**l**) and tail pNSC cluster (**m**) and GFP⁻ cells after culture.

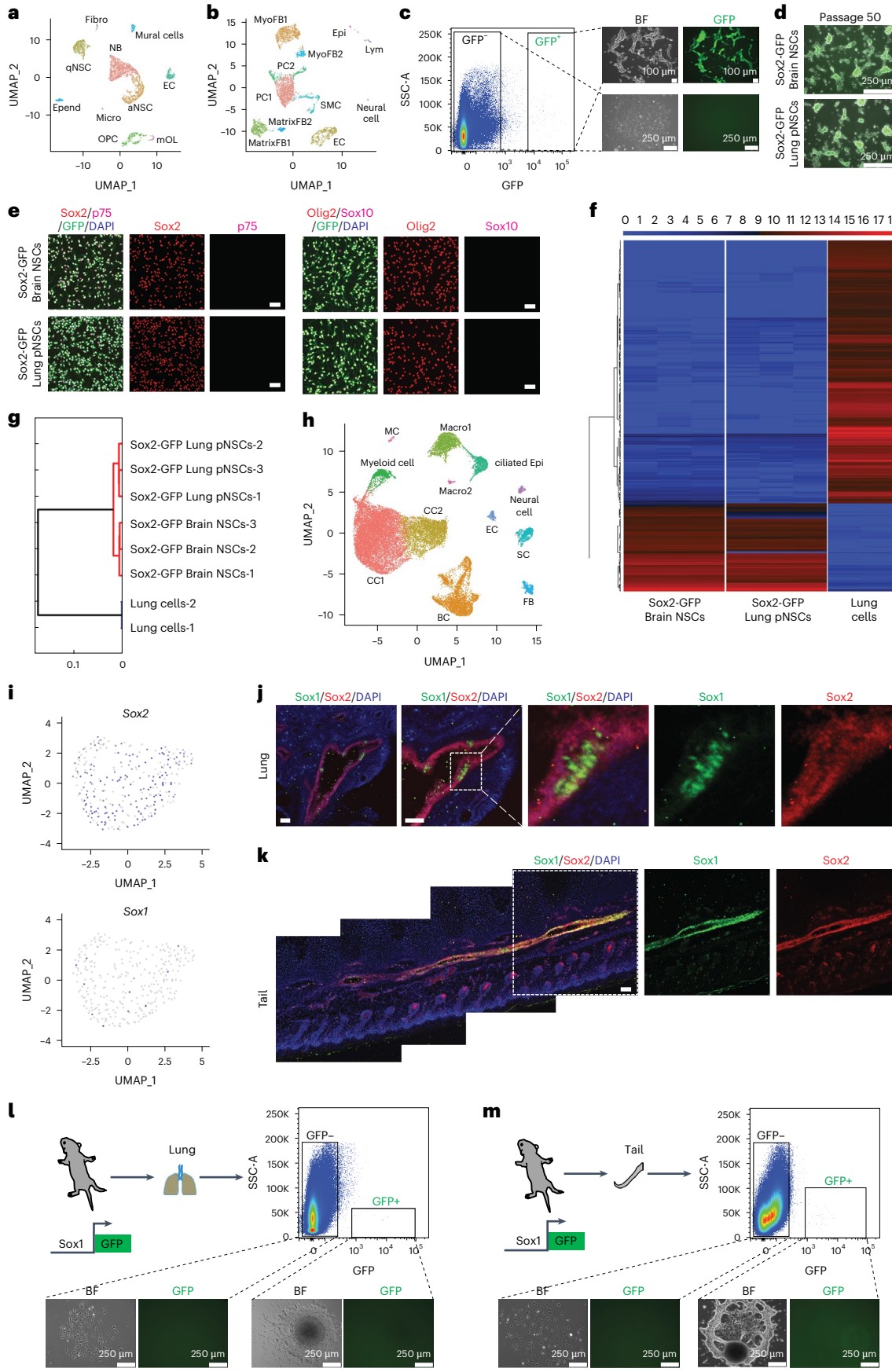

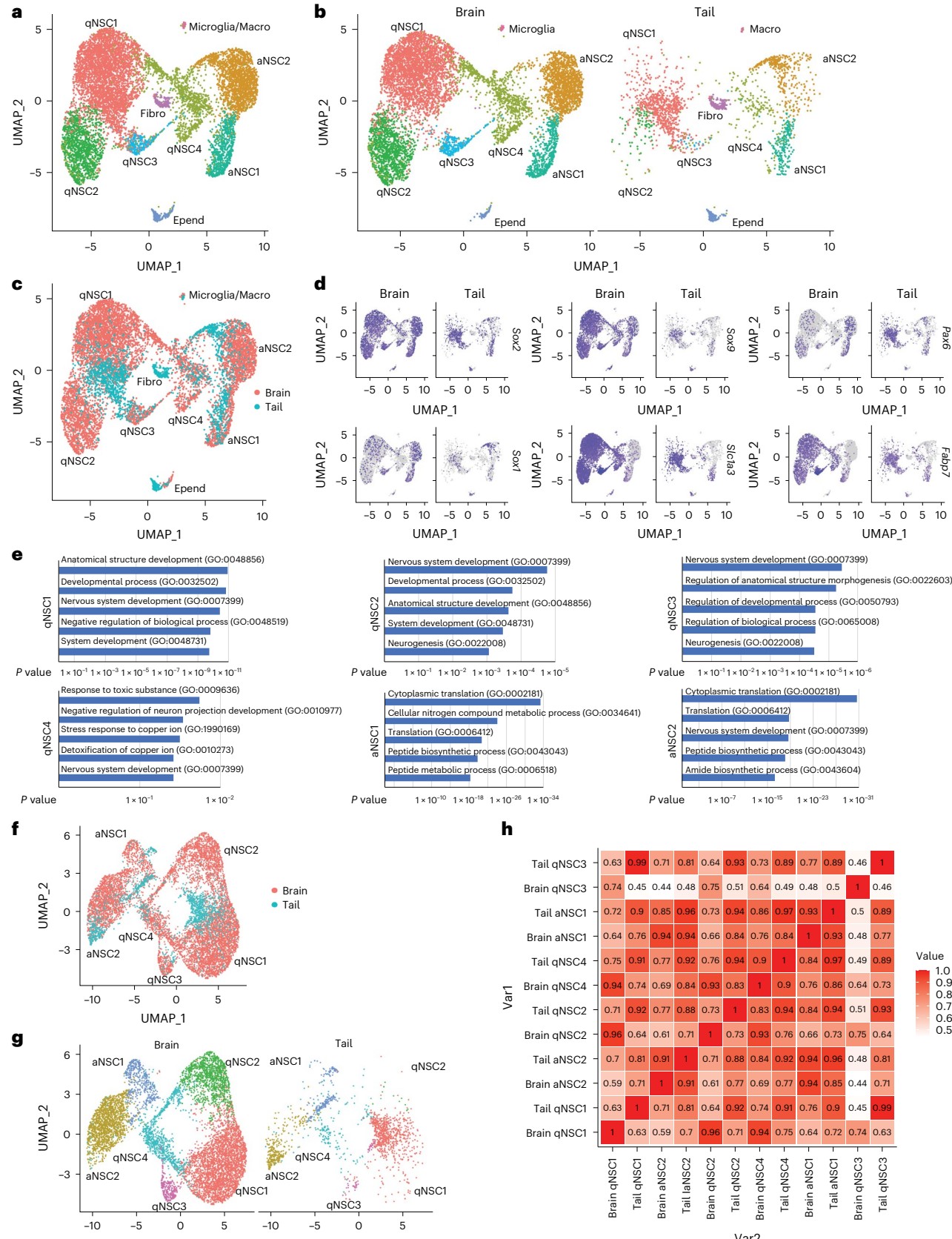

**Fig. 7 | Molecular profiling of pNSCs. a,b**, UMAP of Sox1-GFP⁺ cells from adult brain and postnatal tail tissues in combination (**a**) or separated (**b**). Colours represent cell population clusters. **c**, UMAP as in **a**, with colours representing cells from brain and tail. **d**, Expression of NSC markers in brain and tail clusters. **e**, Top five GO terms of NSC cluster-expressing genes. *P* value calculated by

Fisher's test. **f**, UMAP of Sox1-GFP⁺ NSCs from adult brain and postnatal tail tissues after removing non-neural cells, with colours representing cells from the brain and tail. **g**, UMAP of separated Sox1-GFP⁺ NSCs in **f**. **h**, Pearson correlation coefficients across clusters. *n* = 1 biological replicate for all the scRNA-seq samples.

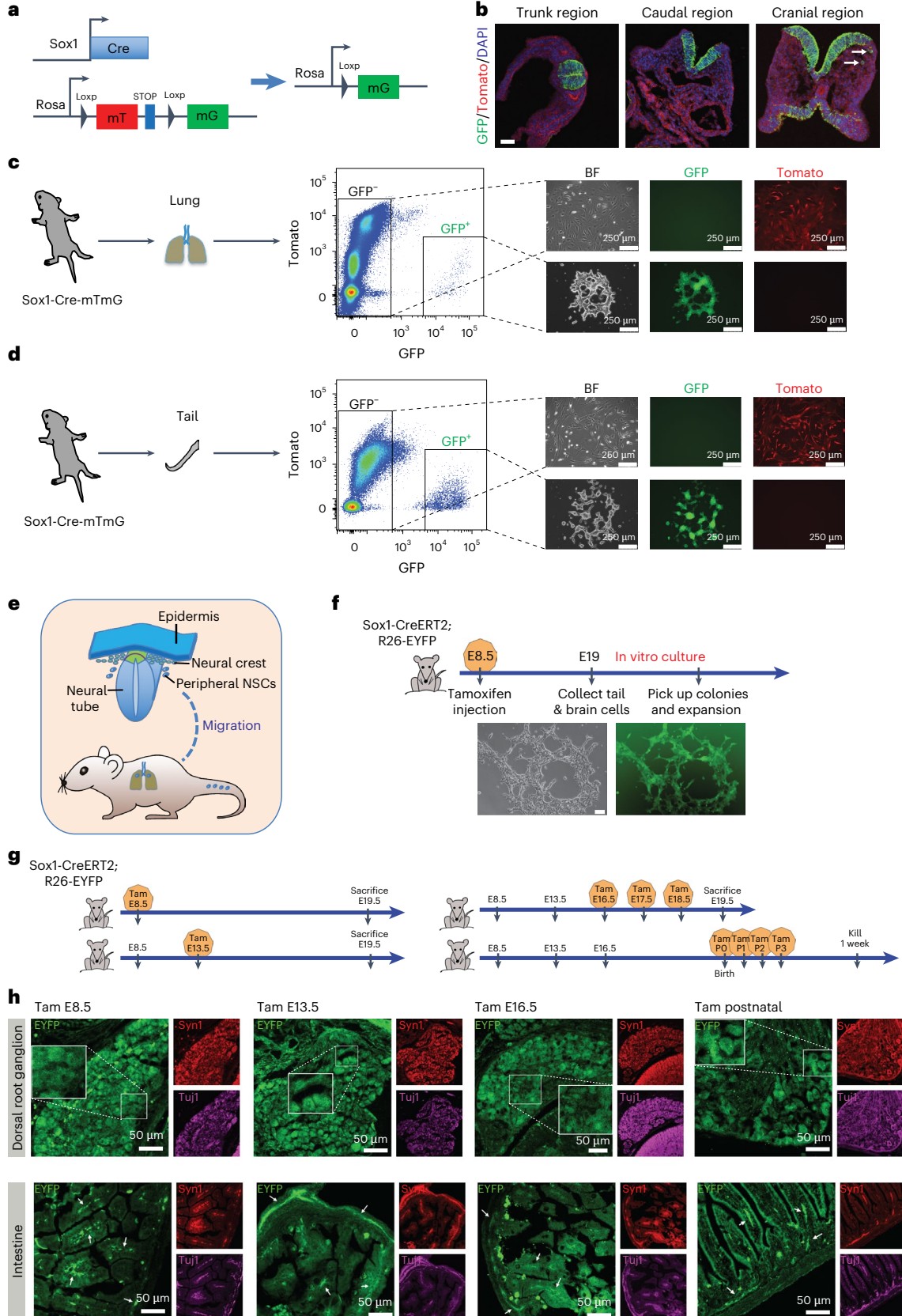

**Fig. 8 | Differentiation of pNSCs in situ during early development. a**, Genetic tracing of NECs and their progenies. **b**, Immunohistochemistry of E8.5 Sox1-Cre-mTmG mouse embryos. White arrows indicate migrated GFP⁺ cells. Data represent three biological replicates. Scale bar, 50 μm. **c,d**, FACS strategy for deriving lung (**c**) and tail pNSCs (**d**) from postnatal Sox1-Cre-mTmG mouse, and morphologies of primary lung (**c**) and tail pNSC cluster (**d**) and GFP⁻ cells after culture. **e**, Schematic of pNSCs' origination, migration and distribution. **f**, Strategy of tamoxifen injection to label NECs and morphology of a primary tail pNSC cluster. Scale bar, 50 μm. **g**, Strategy of tamoxifen injection to label pNSC progenies. **h**, Immunohistochemistry of DRG and intestine tissues across different time points. White arrows represent GFP⁺ mature neuronal cells. Data represent three biological replicates. Scale bar, 50 μm. Syn1, Synapsin 1.

## Molecular profiling of pNSCs

We next sought to investigate the molecular profile of Sox1[+] pNSCs in situ. As such populations were relatively rare in the lung, we focused on tail pNSCs. We performed scRNA-seq experiments on the postnatal tail and adult brain Sox1-GFP[+] cells. Cells from both brain and tail were pooled together and separated into nine different cell-type clusters. Abundant cell populations expressed many known NSC or intermediate neural progenitor (IPC) markers and were identified as qNSC1, qNSC2, qNSC3, qNSC4, aNSC1 and aNSC2. Other very minor populations were identified as ependymal cells (Epend), fibroblasts (Fibro) and microglia/macrophages (Macro). We suspect that this may be due to flow cytometry noise or unspecific sorting (Fig. 7a, Extended Data Fig. 5a–c and Supplementary Table 4).

Notably, the majority of tail Sox1-GFP[+] cells were clustered together with brain NSCs (Fig. 7b,c). They expressed many brain NSC markers (Fig. 7d and Extended Data Fig. 5d). Nervous system development and neurogenesis appeared in the top five GO terms (Fig. 7e). Compared with Schwann cells from the Sox2-GFP[+] scRNA-seq dataset, pNSCs highly expressed NSC-specific markers, but not Schwann cell/NC-derived cell markers (Extended Data Fig. 5e,f), indicating that pNSCs are different from Schwann cells. These results demonstrate that tail Sox1-GFP[+] cells show similar molecular features to brain NSCs.

The majority of tail Sox1[+] cells were clustered together with brain qNSCs, indicating that they are quiescent (Extended Data Fig. 6a). Ki67 stained positive for roughly 13.1% of tail Sox1[+] pNSCs (n = 520 cells), not only confirming our scRNA-seq data but also indicating the self-renewal of certain pNSCs (Extended Data Fig. 6b). Pseudotemporal ordering revealed the development trajectory of qNSC–aNSC in both tail pNSCs and brain NSCs (Extended Data Fig. 6c), indicating the ongoing neurogenesis process from tail pNSCs at the postnatal stage.

As non-neural cells in our Sox1-GFP[+] scRNA-seq analysis might lead to misrepresentation of cell clustering, we reanalysed our data after excluding those cells. Tail pNSCs still clustered together with brain NSCs (Fig. 7f,g). Majority of genes expressed in pNSCs were also present in brain NSCs, including important NSC markers (Extended Data Fig. 6d). GO analysis of conserved marker genes enriched biological processes involved in nervous system development, neuron differentiation and brain development. Only in the aNSC1 cluster are the conserved marker genes more strongly associated with cell cycle and cell division genes, consistent with the state of active NSCs (Extended Data Fig. 6e).

However, pNSCs did not completely overlap with brain NSCs. The Pearson correlation coefficients of qNSC4, aNSC1 and aNSC2 clusters between brain NSCs and tail pNSCs were >0.9, indicating high similarity. However, the Pearson correlation coefficients of qNSC1, qNSC2 and qNSC3 clusters between brain NSCs and tail pNSCs were much lower (Fig. 7h), suggesting that these cells in the tail only partially resemble brain NSCs. In addition, many brain NSC markers were not expressed in tail pNSCs (Extended Data Fig. 6d). GO analysis revealed that these genes mainly fall into processes related to protein translation, metabolic processes, transcription regulation and cell cycle, which may indicate the different status of pNSCs and brain NSCs. Several other GO terms indicate involvement in nervous system and brain development, which is consistent with the developmental potential of brain NSCs (Extended Data Fig. 6f). This demonstrates that pNSCs are similar but are not identical to brain NSCs, which is expected, as these cells most likely follow different developmental trajectories and interact with different microenvironments.

We also compared the molecular profile of Sox1[+] pNSCs with that of other tissue-resident stem cells from published data: lung AT1 and AT2 stem cells and E13.5 embryonic hair follicle epithelium stem cells[31,32]. These cells are not comparable to pNSCs (Extended Data Fig. 7a,b), suggesting that pNSCs are a unique group of NSCs that are distinct from other groups of tissue-resident stem cells.

## LdNSCs are from endogenous Sox1[+] pNSCs

To determine the origin of ldNSCs, we performed low-pH treatment on Sox1-GFP[−] and Sox1-GFP[+] postnatal lung cells. We only observed NSC-like cell cluster from Sox1-GFP[+] cells (Extended Data Fig. 7c,d), indicating that ldNSCs are actually derived from endogenous Sox1[+] pNSCs.

Quiescence is not a singular, but a graded state. To test whether low-pH treatment can trigger quiescent pNSCs activation and increase the efficiency of pNSC derivation, we treated tail and brain Sox1-GFP[+] cells with low-pH which was able to increase the efficiency of tail pNSC and brain NSC derivation, suggesting that the graded quiescence of pNSCs/NSCs can be activated by low-pH treatment (Extended Data Fig. 7e).

## pNSCs originate from neuroepithelial cells

We have proven that pNSCs are not from NCCs. To assess whether pNSCs originate from NECs, we crossed Sox1-Cre mouse[33] (RBRC05065) with R26-mTmG mouse to permanently label NECs and their progenies (Fig. 8a). In E8.5 embryos, GFP expression was detected exclusively in NECs except for few GFP[+] cells that started to migrate out of the neural fold in the cranial region, confirming the efficient and specific labelling of this system (Fig. 8b and Extended Data Fig. 7f,g). Notably, pNSCs from Sox1-Cre-mTmG postnatal lung and tail tissues were GFP[+]/Tomato[−] and could only be derived from GFP[+] cells (Fig. 8c,d). They could be maintained in passage and were similar to brain NSCs in marker gene expression and differentiation ability (Extended Data Fig. 7h–l). These results demonstrate that pNSCs originate from Sox1[+] NECs (Fig. 8e).

To rule out a late onset of Sox1 expression in the peripheral tissue, we crossed Sox1-CreERT2 mice[34] with R26-EYFP mice[35] and injected tamoxifen only once into E8.5 pregnant female mice to ensure that only the earliest NECs can be labelled. At E19, the pregnant mice were killed and Sox1-CreERT2-R26-EYFP-positive pups' tail tissues were dissociated into single cells and cultured to derive tail pNSCs. Indeed, we were able to derive EYFP[+] tail pNSCs (Fig. 8f and Extended Data Fig. 8a–c). We also noted EYFP[−] tail pNSC clusters from the same experiments (Extended Data Fig. 8d), likely because of low Cre-loxP recombination efficiency caused by low tamoxifen dosage injected into the pregnant female mice. Similarly, EYFP[+] and EYFP[−] brain NSCs were also observed (Extended Data Fig. 8e).

## Differentiation of pNSCs in situ during early development

We showed that some Sox1[+] NECs migrate out of the neural tube and become pNSCs. To assess pNSC functionality, postnatal Sox1-Cre-mTmG mouse tissues were stained with neuronal and glial cell markers. In many tissues, GFP[+] cells were positive for neuronal marker Tuj1, indicating the neuronal differentiation of pNSCs. However, in these tissues, most GFP[+] cells did not express glial cell markers GFAP, MBP or MPZ (Extended Data Fig. 8f,g). Only a few GFP[+] cells in lung, tail and trunk muscle tissues expressed weak GFAP or MPZ (Extended Data Fig. 8h), indicating the limited glial differentiation potential of pNSCs. Notably, heart tissue exhibited differentiated GFP[+] neurons but no Sox1[+] cells, indicating complete differentiation of migrated pNSCs (Extended Data Figs. 4f and 8f). These results demonstrate that many pNSCs differentiate into neurons and limited glial cells during embryonic development, but some pNSCs do not differentiate and maintain a stem state until adult age.

To examine the stages of development that pNSCs contribute to, we crossed Sox1-CreERT2 mice with R26-EYFP mice and injected tamoxifen at different time points across development (Fig. 8g). In some tissues, such as heart and lung, a substantial neuronal contribution of pNSC progeny was observed only at earlier time points. Such cells were rarely observed when tamoxifen was injected at E16.5 or postnatally, suggesting that the deposited pNSCs have completed their differentiation and contribution in early organogenesis (Extended Data Fig. 9a,b). However, in other tissues, such as DRG, intestine and thymus, the contribution of pNSC progenies to neurogenesis persisted even after birth (Fig. 8h and Extended Data Figs. 9c and 10a,b). Although

neurogenesis of pNSCs in the postnatal tail was not as high, we observed mature neurogenesis from pNSCs after birth, clearly demonstrating the neurogenic niche of pNSCs in postnatal tail (Extended Data Fig. 10c). Of note, EYFP+ neurons were also positive for functional neuronal markers, suggesting a functional contribution of pNSCs in the endogenous environment (Fig. 8h and Extended Data Figs. 9 and 10a–c). No EYFP+ cells were observed in DRG and intestine tissues without tamoxifen injection, ruling out leaky Cre recombination (Extended Data Fig. 10d).

In conclusion, our data demonstrate that pNSCs contribute to neurogenesis in multiple organs, following diverse developmental trajectories. This ongoing and robust neurogenesis underscores the intrinsic NSC properties of pNSCs and their crucial role in PNS development.

## Discussion

Our research challenges the conventional belief that NSCs are exclusive to the CNS by revealing the presence of pNSCs outside the CNS. These pNSCs closely resemble brain-derived NSCs and demonstrate potential therapeutic utility through engraftment and differentiation in the mouse brain.

During vertebrate development, NECs in the neural tube form the CNS, while NCCs form the PNS[36]. Here we identified pNSCs that also contribute to the formation of the PNS and are not derived from the NC. The Wnt1-Cre mouse line is commonly used for lineage tracing studies of NCCs[37,38]. Although Wnt1 lineage tracing labels midbrain cells in tandem, our experiments revealed that pNSCs are not derived from Wnt1-Cre+ cells, indicating they do not originate from the NC. Instead, we found that pNSCs come from NECs, demonstrating that NECs also migrate out of the neural tube to contribute to PNS formation.

pNSCs, although rare, are not stray stem cells that evade canonical development, but rather endogenous NSCs in the periphery. They contribute continuously and substantially to neurogenesis in different tissues and are distributed at specific locations along the bronchi and tail. Transcriptomic analysis shows strong overlap between pNSC and brain NSC. pNSCs, distinct from other resident stem cells, are authentic NSCs that are an integral part of mouse embryonic and postnatal development. Our data also suggest that the maintenance of pNSCs in different tissues might be on the tissues' own initiative.

The 'niche' supporting the NSCs affects the behaviour of these cells[39]. Cumulative evidence suggests that NSCs in the CNS are mostly quiescent[40,41]. Our scRNA-seq data showed that the majority postnatal tail pNSCs were also quiescent. Active NSCs were also identified in postnatal tail pNSCs. Developmental trajectory analysis showed the transition from quiescent to active NSCs in both tail pNSCs and brain NSCs. We also found that pNSCs continuously differentiate into mature neurons during development, indicating ongoing neurogenesis from pNSCs.

We isolated pNSCs from tissues physically distant from the CNS, ruling out contamination. Previous studies have demonstrated that NCSCs from embryonic DRG tissue can be reprogrammed into NSCs[11,42]. Our research revealed that pNSCs are distinct from NCSCs and do not originate from NCCs and their progenies. Previous studies derived NCSCs in vitro by using the neurosphere formation method, which may have included pNSCs that may be dominantly expanded, giving a false reprogramming impression[11,42]. Furthermore, pNSCs could be derived only from Sox1+ cells, which is the earliest known NSC-specific marker[29,30]. These data suggests the endogenous stem cell origin of pNSCs. Notably, pNSCs share similarities with brain NSCs, not Schwann cells, confirming their NSC identity and ruling out reprogramming from NC-derived cells. These findings highlight the derivation of pNSCs is distinct from reprogramming processes.

The potential of pNSCs as reparative therapeutics for neural diseases and injuries remains to be explored. It would also be interesting to determine whether pNSCs exist in humans. Our finding that pNSCs reside outside the CNS challenges the current notion that NSCs exist only in the CNS. This facilitates insight into neural development and

suggests a higher level of cellular plasticity for the nervous system than had previously been observed. pNSCs, which are more accessible than NSCs in the CNS, could potentially be used for neural regenerative therapy or in the treatment of neurological diseases.

## Online content

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

[1]Department of Cell and Developmental Biology, Max Planck Institute for Molecular Biomedicine, Münster, Germany. [2]Stem Cell & Regenerative Medicine Consortium, School of Biomedical Sciences, Li Ka Shing Faculty of Medicine, The University of Hong Kong, Hong Kong, China. [3]Department of Tissue Morphogenesis, Max Planck Institute for Molecular Biomedicine, Münster, Germany. [4]Developmental and Cellular Biology, Luxembourg Centre for Systems Biomedicine (LCSB), University of Luxembourg, Belvaux, Luxembourg. [5]Department of Cellular Biophysics, Institute for Medical Physics and Biophysics, University of Münster, Münster, Germany. [6]Flow Cytometry Unit, Max Planck Institute for Molecular Biomedicine, Münster, Germany. [7]Third Affiliated Hospital of Guangzhou Medical University, Guangzhou, China. [8]Division of Basic Research, Guangzhou National Laboratory, Guangzhou, China. [9]Department of Medical Life Sciences, College of Medicine, The Catholic University of Korea, Seoul, Republic of Korea. [10]NUOXINTE Biotechnology, Suzhou, China. [11]Group of Computational Biology and Systems Biomedicine, Biogipuzkoa Health Research Institute, San Sebastian, Spain. [12]IKERBASQUE, Basque Foundation for Science, Bilbao, Spain. [13]IZKF Münster and Cluster of Excellence EXC 1003, Cells in Motion (CiM), Münster, Germany. [14]University of Münster, Medical Faculty, Münster, Germany. [15]Center for Translational Stem Cell Biology, Hong Kong, China. ✉e-mail: dhan23@hku.hk; pliu88@hku.hk; office-schoeler@mpi-muenster.mpg.de

## Methods

### Mice and cells

Embryonic limb cells were derived from embryos at embryonic day 13.5. Adult brain and lung cells were derived from ~4–6-week-old WT or transgenic mice. Postnatal brain, lung, tail and DRG cells were derived from ~postnatal day 1–5 WT or transgenic mice. Limb, lung, tail, DRG or whole brain tissues were cut and minced using a pair of scissors and incubated with 0.25% trypsin (Invitrogen) at 37 °C for 10 min. After trypsinization, 3× volume amount of MEF medium (high-glucose DMEM (Sigma) containing 10% FBS (Biochrom), 1% GlutaMAX (Gibco), 1% sodium pyruvate (Sigma), 1% nonessential amino acids (Sigma), 0.1% β-mercaptoethanol (Gibco) and 1% penicillin–streptomycin (Sigma)) was added and the entire suspension was pipetted up and down to dissociate the tissues. The dissociated tissues were centrifuged and resuspended in MEF or NSC medium, and cells were plated onto six-well plates and cultured in a humidified incubator at 37 °C under 5% $CO_2$. Passage-4 embryonic limb cells and passage-2 adult lung cells from WT mice were maintained and passaged in MEF medium.

All mice used were bred and housed at the mouse facility Max Planck Institute (MPI) in Muenster and the Centre for Comparative Medicine Research (CCMR) at The University of Hong Kong (HKU), and animal handling was conducted in accordance with MPI or HKU CCMR animal protection guidelines. The protocols for animal handling and maintenance for this study were approved by the Landesamt für Natur, Umwelt und Verbraucherschutz Nordrhein-Westfalen under the supervision of a certified veterinarian in charge of the MPI animal facility (protocols Az 81-02.05.50.17.014, Az 84-02.04.2016.A525, Az 81-02.04.2017.A376 and Az 84-02.05.2016.A494) and by the Government of the Hong Kong Special Administrative Region Department of Health (protocol 23-208 in DH/HT&A/8/2/3 Pt.55). C57BL/6, CDl and B6C3HFl mice were bred in-house. The background of WT mice used in this study was CDl mice or mixtures between CD1 and C3H. All transgenic mice were bred with WT mice in-house for experiments. For the teratoma assay, SCID mice (2–3 months old) were used. For the in vivo transplantation experiments, NOD.Cg-*Prkdc*<sup>scid</sup> *Il2rg*<sup>tm1Wjl</sup>/SzJ mice or C57BL/6 mice were used. Mice were maintained in the animal facility with a controlled temperature of 22 °C, 40–60% humidity, a 14–10-h light–dark photoperiod and free access to water and food.

### Generation of pNSCs

For generation of low-pH NSCs, passage-4 embryonic limb cells, passage-2 adult lung cells from WT mice were dissociated by trypsin. The trypsin treatment was stopped by adding MEF medium. The cells were then washed twice in D-PBS (without calcium chloride and magnesium chloride) (Sigma). Then, $0.5 \times 10^6$ cells were treated with 500 µl of low-pH HBSS solution (Biochrom) (titrated to different pH by hydrochloric acid) for 25 min at 37 °C in water bath, and then centrifuged at 200*g* at room temperature for 5 min. After the supernatant was removed, precipitated cells were resuspended and plated in a well of a six-well plate without coating in simple defined NSC medium: DMEM/F-12 (1:1) (Gibco) with 2% B27 without vitamin A, 1% N2 (both from Gibco), 1× GlutaMAX (Gibco), 1× penicillin–streptomycin (Sigma), supplemented with 10 ng ml⁻¹ EGF and 10 ng ml⁻¹ human bFGF (both from Invitrogen). Freshly isolated GFP⁻ and GFP⁺ postnatal lung cells after FACS from Sox1-GFP mice were treated with low-pH medium in the same way and then the cells were resuspended and plated in gelatin-coated six-well plate in NSC medium for 3 weeks. The medium was changed every 48 h. A single pNSC cluster was manually picked and cultured in a 96-well plate individually. As observed from the pictures of the pNSCs, the pNSC clusters were usually clonal due to the low colony formation efficiency. Consequently, we were able to pick each pNSC cluster with pipette tips. We made every effort to extract only the pNSC cluster cells by aspirating the central cells of the NSC clusters. While we could not 100% guarantee that we were sampling only pNSC cells without touching surrounding cells, the few surrounding cells

that were inadvertently included were minimal and would be lost in multiple passages.

For generation of pNSCs without low-pH treatment, freshly isolated embryonic limb cells and postnatal tail cells from WT mice, freshly isolated adult lung or postnatal tail GFP⁺ cells from Nes-GFP mice, freshly isolated adult lung or postnatal lung GFP⁺ and GFP⁻ cells from Sox2-GFP mice, freshly isolated postnatal lung or postnatal tail GFP⁺ and GFP⁻ cells from Sox1-GFP mice, freshly isolated postnatal lung or postnatal tail GFP⁺ and GFP⁻ cells from Sox1-Cre-mTmG mice were cultured on gelatin-coated dishes in NSC medium for 3 weeks. Medium was changed every 48 h. A single pNSC cluster was manually picked and cultured in a 96-well plate individually.

For generation of brain control NSCs, whole brain tissues from adult or postnatal mice were cut and minced using a pair of scissors, and incubated with 0.25% trypsin (Invitrogen) at 37 °C for 10 min. After trypsinization, 3× volume amount of MEF medium was added, and the entire suspension was pipetted up and down to dissociate the tissues. The dissociated tissues were centrifuged and resuspended in NSCs medium, and cells were plated onto six-well plates coated with gelatin, and cultured in a humidified incubator at 37 °C under 5% $CO_2$. After several passages, pure brain NSCs can be established.

### Quantitative RT–PCR

Total RNA was extracted using the RNeasy Mini kit (QIAGEN). cDNA synthesis was performed using the High-Capacity cDNA Reverse Transcription kit (Applied Biosystems). Transcript levels were determined using iTaq SYBR Green Supermix with ROX (Bio-Rad). Gene expression was normalized to the housekeeping gene *Gapdh* expression and calculated using the delta Ct algorithm. All data were calibrated to control NSCs, whose expression was considered to be 1 for all genes. Error bars in the figures represent s.d. of biological triplicates. Primer sequences are listed in Supplementary Table 5.

### Microarray and data analysis

RNA samples to be analysed by microarrays were prepared using QIAGEN RNeasy columns with on-column DNA digestion. Then, 300 ng of total RNA per sample was used as input into a linear amplification protocol (Ambion) that involved synthesis of T7-linked double-stranded cDNA and 12 h of in vitro transcription incorporating biotin-labelled nucleotides. Purified and labelled cRNA was then hybridized for 18 h onto MouseRef-8 v2 expression BeadChips (Illumina) following the manufacturer's instructions. After washing as recommended, the chips were stained with streptavidin-Cy3 (GE Healthcare) and scanned using the iScan reader (Illumina) and accompanying software. Samples were exclusively hybridized as biological replicates. The bead intensities were mapped to gene information using BeadStudio 3.2 (Illumina). Background correction was performed using the Affymetrix Robust Multi-array Analysis background correction of the lumiExpresso v.2.24.0 package of R-Bioconductor v.3.0 model[43]. Variance stabilization was performed using $\log_2$ scaling, and gene expression normalization was calculated with the method implemented in the lumi package of R-Bioconductor 3.0. Data post-processing and graphics were performed with in-house-developed functions in MATLAB (R2020b). Hierarchical clustering of genes and samples was performed with the one minus correlation metric and the unweighted average distance (UPGMA) (also known as group average) linkage method of MATLAB (R2020b) software. The molecular signatures were taken from the gene set collection C5 of the v.3.0 of the Molecular Signatures Database (MSigDB)[44]. The significance of the gene set of the different expressed genes was analysed using an enrichment approach based on the hypergeometric distribution. The significance (*P* value) of the gene set enrichment was calculated using the hypergeometric distribution. The multi-test effect influence was corrected by controlling the false discovery rate using the Benjamini–Hochberg correction at a significance level of α = 0.05.

## Bisulfite sequencing

To determine DNA methylation status at the promoter region of *Col1a1* and enhancer region of *Nestin*, bisulfite conversion was carried out on 2 µg of isolated genomic DNA from cells with the EpiTect Bisulfite kit (QIAGEN) according to the manufacturer's protocol. The bisulfite-converted DNA was amplified by PCR using the primers, Col1a1 promoter F: GTTAGGTAGTTTTGATTGGTTGG, Col1a1 promoter R: ACAATAACCCCTAAAAAAACAAAAA, Nestin enhancer F: TAAAGAGGTTGTTTGGTTTGGTAGT and Nestin enhancer R: CTATTCCACTCAACCTTCCTAAAA. The PCR products were cloned into the pCRII TOPO vector (Invitrogen) according to the manufacturer's protocol. Individual clones were sequenced by GATC-biotech, which has since been acquired by Eurofins Genomics (part of Eurofins Scientific) (https://eurofinsgenomics.eu/en/custom-dna-sequencing/). Sequences were analysed using the Quantification Tool for Methylation Analysis (http://quma.cdb.riken.jp).

## In vitro differentiation of NSCs and NCSCs

For general neural differentiation, NSCs or NCSCs were seeded at a density of 100,000 cells per well on poly-D-lysine (Sigma)-coated 24-well plates in NSC medium. For neurons differentiation, the next day, the medium was replaced by neural differentiation medium (DMEM/F-12 (1:1) with 2% B27 w/o vitamin A, 1% N2, 1× GlutaMAX, 1× penicillin–streptomycin). On day 6, 10 ng ml$^{-1}$ BDNF, 10 ng ml$^{-1}$ GDNF and 100 µM dbcAMP (all from PeproTech) were added and the cells were cultured for another 14 days. For directed astrocyte differentiation, the medium was replaced by neural differentiation medium, supplemented with 10% FBS (Biochrom) the next day, and the cells were cultured for another 5 days. For oligodendrocyte differentiation, the next day, the medium was replaced by neural differentiation medium. On day 6, 30 ng ml$^{-1}$ T3 (Sigma) and 200 mM ascorbic acid (PeproTech) were added and the cells were cultured for another 14 days. During culture, the medium was changed every other day.

## Immunocytochemistry

For immunofluorescence, cells were fixed in 4% paraformaldehyde (Electron Microscopy Sciences), washed three times with PBS (Sigma), and then incubated in PBS containing 0.1% Triton-X-100, 3 mg ml$^{-1}$ BSA fraction V (both from Sigma) and 10% FBS (Biochrom) for 45 min at room temperature. The cells were then incubated with primary antibodies overnight at 4 °C. Primary antibodies consisted of mouse anti-Nestin (Millipore, MAB353C3, 1:200), rabbit anti-Sox2 (Cell Signalling Technology, 23064, 1:1,000), goat anti-Sox2 (Santa Cruz, sc-17320, 1:500), rabbit anti-Olig2 (Millipore, AB9610, 1:1,000), goat anti-Sox1 (R&D, AF3369, 1:200), mouse anti-Tuj1 (Sigma, T8660, 1:1,000), rabbit anti-Tuj1 (BioLegend, 802001, 1:1,000), rabbit anti-MAP2 (Santa Cruz, SC-20172, 1:1,000), rabbit anti-GFAP (Millipore, AB5804, 1:500), rabbit anti-S100B (Abcam, ab41548, 1:200), mouse anti-O4 (Millipore, MAB345, 1:100), rat anti-MBP (Abcam, ab7349, 1:500), rabbit anti-glutamate (Sigma, AB5018, 1:2,000), rabbit anti-GABA (Millipore, ABN131, 1:500), rabbit anti-ChAT (Millipore, AB143, 1:500), mouse anti-vesicular glutamate transporter 1 (vGluT1) (Millipore, MAB5502, 1:100), rabbit anti-p75 (Sigma, AB1554, 1:500) and rabbit anti-Sox10 (Abcam, ab155279, 1:500). The day after incubation with primary antibodies, the cells were washed three times with PBS containing 0.1% BSA (Sigma) and further incubated with secondary antibodies (Invitrogen) for 60 min at room temperature. Nuclei were detected by 4,6-diamidino-2-phenylindole (DAPI) (Invitrogen) staining. Images were acquired using a Leica DMI6000B inverted fluorescence microscope equipped with a Hamamatsu Orca-R2 charge-coupled device camera.

## Electrophysiological recordings

Voltage-clamp recordings were performed in control NSCs- or ldNSCs-derived neurons after 2–3 weeks of differentiation, using the whole-cell configuration of the patch-clamp technique[45]. Signals were amplified using HEKA EPC-10 patch-clamp amplifier (HEKA Elektronik Dr. Schulze), sampled at 10 kHz and filtered at 3 kHz via four-pole Bessel low pass filter. Data were acquired using PatchMaster 2.4 software (HEKA Elektronik Dr. Schulze). The patch pipettes were fabricated from borosilicate glass (Science Products) on a P-1000 micropipette puller (Sutter Instruments) and had a resistance of 3–5 MΩ when filled with pipette solution. The intracellular recording solution contained (all Sigma) 125 mM potassium aspartate, 10 mM NaCl, 1 mM EGTA, 15 mM phosphocreatine, 4 mM Mg-ATP, 0.4 mM Na$_2$-GTP, 10 mM D-glucose and 10 mM HEPES adjusted to pH 7.4 with KOH. The extracellular bath solution contained (all Sigma) 130 NaCl mM, 10 mM NaHCO$_3$, 2.4 mM KCl, 2.5 mM CaCl$_2$, 1.3 mM MgCl$_2$, 10 mM D-glucose and 10 mM HEPES adjusted to pH 7.4 with NaOH. Tetrodotoxin (Alomone Labs) was added to the bath solution at 1 µM (if needed). Series resistance (10–20 MΩ), pipette and cell capacitance, and liquid junction potential were cancelled electronically. All experiments were performed at room temperature. Recordings of current–voltage relationship ('I–V curves') were performed in voltage-clamp mode at a holding potential of −80 mV. Values of membrane potential and voltage responses to current injections were recorded in current-clamp mode. Data analysis and visualization were performed with Patcher's Power Tool routine (developed by F. Mendez and F. Würriehausen, MPI BPC) for IgorPro (WaveMetrics) and SciDAVis program (http://scidavis.sourceforge.net/index.html).

## Karyotyping

The confluent monolayer of allNSCs cultured in 25T flask was treated with 10 µg ml$^{-1}$ colcemid (Gibco) for 4 h to arrest cells at the metaphase stage. The cells were washed gently three times with 0.075 M KCl hypotonic solution (Merck) and 1% sodium citrate (Merck), and then treated with hypotonic solution at 37 °C for 25 min. The cells were dissociated by shaking the flask and then collected into a 15-ml conical tube. After centrifugation at 300g for 10 min, the supernatant was removed, and the cell pellet was washed three times with fixative solution (methane:acetic acid ratio of 3:1, Merck). The cell pellet was suspended with fixative solution and dropped on a cold wet slide. The slide was treated with trypsin and then stained with Giemsa (Sigma-Aldrich). The slide was observed under the light microscope.

## FACS

In brief, adult or postnatal mice were killed and lung, brain or tail tissues were cut and minced using a pair of scissors, and incubated with 0.25% trypsin (Invitrogen) at 37 °C for 10 min. After trypsinization, a 3× volume amount of MEF medium was added and the entire suspension was pipetted up and down to dissociate the tissues. The dissociated tissues were centrifuged and resuspended in FACS medium (D-PBS without calcium and magnesium (Sigma) supplemented with 0.3% BSA (Sigma)). Flow cytometry was performed using either FACS Canto (BD Biosciences) or FACS Aria II (BD Biosciences) and the data were analysed using either eBD FACSDiva (BD Biosciences) or FlowJo software (TreeStar).

## Cell labelling

LdNSCs and the corresponding adult brain NSCs were labelled with mCherry using pLVTHM (Addgene, 12247) lentivirus infection system. To produce lentiviruses, HEK293 cells were plated at a density of 3 × 10$^6$ per 10-cm dish. On the next day, the cells were transfected with 3 µg psPAX2 (Addgene, 12260), 1.5 µg pMD2.G (Addgene, 12259) and 4.5 µg lentiviral vector pLVTHM using 27 µl Fugene 6 in 600 µl Opti-MEM per dish. Virus-containing supernatants were collected at 48 h post-transfection, filtered through a 0.45-mm PVDF filter, concentrated at 20,000 rpm for 2 h using an optima L-100 XP ultracentrifuge (Beckman Coulter), resuspended in 1 ml Knockout DMEM and stored at −80 °C until use. LdNSCs and the corresponding brain NSCs were infected with pLVTHM-mCherry virus, then mCherry-positive cells were sorted by FACS.

pNSCs derived without low-pH treatment and the corresponding brain NSCs were labelled with CAG-promoter driven H2B-Tomato-IRES neomycin or H2B-EGFP-IRES neomycin using the lipofectamine transfection method. pNSCs and brain NSCs were plated at a density of $3 \times 10^6$ per 10-cm dish. The next day, cells were transfected with 5 μg CAG-H2B-Tomato-IRES neomycin or CAG-H2B-EGFP-IRES neomycin piggyBac expression vector and 2 μg PBase vector using 21 μl lipofectamine 2000 (Invitrogen) in 600 μl Opti-MEM per dish. After selection by neomycin, Tomato or GFP-positive cells were sorted by FACS.

## Transplantation, perfusion, sectioning and immunohistochemical analysis of ldNSCs

In vivo analysis was conducted as described previously[46]. In brief, mCherry-labelled control NSCs and allNSCs were trypsinized and resuspended into single cells in DMEM-F-12 at a density of $10^5$ cells per μl. With the help of a Hamilton 7005KH 5-μl syringe a total of $3 \times 10^5$ cells were transplanted into the cortex of each hemisphere of adult mice ($n = 6$, NOD. Cg-*Prkdc^scid Il2rg^{tm1Wjl}*/SzJ, 8 weeks). Stereotactic coordinates were defined by using the Franklin and Paxinos mouse brain atlas (in relation to the bregma: anteroposterior (AP) 1.1 mm; mediolateral (ML) ±0.84 mm; and dorsoventral (DV) −2.5 mm below the skull).

Perfusion, sectioning and immunohistochemical analysis were performed 6 weeks after transplantation. Mice were deeply anaesthetized by intraperitoneal injection of 0.017 ml 2.5% Avertin (100% stock solution: 10 g 2, 2, 2-tribromoethanol 10 ml tert-amyl alcohol) per gram of body weight and killed by perfusion. Brains tissues were fixed overnight at 4 °C in 4% paraformaldehyde in PBS. Later, 40-μm thick sections were prepared using a vibratome (Leica,) and blocked for at least 1 h in TBS (0.1 M Tris, 150 mM NaCl, pH 7.4) containing 0.5% Triton-X-100, 0.1% sodium azide, 0.1% sodium citrate and 5% normal goat serum. Immunostainings were performed by incubation of the sections with primary antibodies diluted in the blocking solution for 48 h at 4 °C on a shaker, followed by incubation with the secondary antibody diluted in the blocking solution for 2 h at room temperature. Finally, sections were mounted in AquaMount (DAKO). The primary antibodies used were as follows: rabbit anti-S100B (Abcam, ab41548, 1:200), chicken anti-glial fibrillary acidic protein (Millipore, AB5541, 1:1,000), rabbit anti-Oligo2 (Millipore, AB9610, 1:400) and mouse anti-neuronal nuclei (NeuN) (Millipore, MAB377, 1:400). Alexa fluorophore-conjugated secondary antibodies (Invitrogen) and Hoechst 33342 (Invitrogen) were used to visualize primary antibodies and nuclei, respectively. Sections were analysed using a Zeiss LSM 710 confocal microscope. All confocal images are represented as maximum intensity projections unless stated otherwise.

## Transplantation, perfusion, sectioning and immunohistochemical analysis of pNSCs derived without low-pH treatment

Designed cells with trypsin were collected and concentrated as $10^5$ cells per μl in D-PBS. P1 C57Bl6 pups were used for cell transplantation. After hypo-thermo anaesthesia, pups were quickly installed at the self-made stand and lightly tied with tapes for fixing the position. A 10-μl microliter syringe (Hamilton, Model 701RN) with a 33G sharp end needle was used for the injection. Digital just for mouse stereotaxic instruments (Stoelting, 5173D) were used for positioning. To target the cortex, the needle moved from bregma to the posterior as follows: ML 1 mm, AP 1 mm and DV −2 mm. Not to be affected by skin tension, the target area was pinched with 31G needles. The injection was performed with a microinjector (legato 130) as 1 μl min⁻¹ for 3 μl. We waited for 1 min after the injection to reduce the chance of backflow and slowly pulled out the needle. Pups stayed on the 37 °C hot plate til they were woken and brought back to the mother.

Perfusion, sectioning and immunohistochemical analysis were performed 6 weeks after transplantation. Mice were anaesthetized by intraperitoneal injection of xylazine (Bayer, Rompun 2%; 10 mg kg⁻¹)

and ketamine (Zoetis, Ketavet 100 mg ml⁻¹; 100 mg kg⁻¹) dissolved in saline and killed by perfusion. Brain tissues were fixed overnight at 4 °C in 4% paraformaldehyde in PBS. Later, 80-μm thick sections were prepared using a vibratome (VT 1200S, Leica). After heat-induced epitope retrieval (85 °C, 45 min) in sodium citrate buffer (10 mM sodium citrate and 0.05% Tween 20, pH 6.0), vibratome sections were blocked and permeabilized by overnight incubation in blocking buffer (10% normal donkey serum (Abcam, ab7475), 0.1% BSA (Sigma, P6148) and 0.3% Triton-X-100 (Sigma, T8787) in PBS) at 4 °C. Primary antibodies were diluted in blocking solution and incubated overnight at 4 °C. Mouse anti-NeuN (Millipore, MAB377, 1:400), rat anti-MBP (Abcam, ab7349, 1:400), chicken anti-GFAP (Merck, AB5541, 1:500), rabbit anti-GFAP (Thermo Fisher Scientific, RB087A1, 1:400), goat anti-tdTomato (SIC-GEN, AB8181-200, 1:400), chicken anti-GFP (AVES LABS, AB_2307313, 1:400), rabbit anti-RFP (Biomol, 600-401-379, 1:400), rabbit anti-Ki67 (Abcam, ab15580, 1:400) and mouse anti-Nestin (Millipore, MAB353C3, 1:200) were used as primary antibodies. After staining with primary antibodies, vibratome sections were washed once in 0.5% Triton-X-100 in PBS and three times in PBS (10 min at room temperature each) and incubated overnight with suitable donkey-raised, species-specific, Alexa Fluor-coupled secondary antibodies (all from Invitrogen) diluted (1:1,000) in blocking buffer. After secondary antibody incubation, vibratome sections were stained with DAPI (Invitrogen) diluted at 1 μg ml⁻¹ in PBS for 20 min at room temperature to visualize nuclei, then washed as already described before mounting in Fluoromount G (Southern Biotech, 0100-01). Sections were analysed using a Zeiss LSM 780 confocal microscope. All confocal images are represented as maximum intensity projections unless stated otherwise. For quantification, multiple sections (five to six) from one animal and three independent animals were examined for each group. The efficiency of differentiation into neurons, astrocytes and oligodendrocytes from brain control NSCs and pNSCs was calculated by the average number of NeuN⁺, GFAP⁺ and MBP⁺ cells among H2B-Tomato cell number in each animal. The relative differentiation efficiency was normalized to brain control NSCs injections.

## Inducible genetic experiments

Sox1-CreERT2 mice were crossed with Rosa26-EYFP reporter mice to lineage trace Sox1-positive cells and their progenies. Tamoxifen was intraperitoneally injected into Sox1-CreERT2-R26-EYFP pregnant mothers at E8.5, E13.5 and E16.5 time points, and they were killed at E19.5 for analysis. Newborn pups were also injected with tamoxifen and killed after 1 week to study postnatal development. Tamoxifen (Sigma, T5648) was dissolved in corn oil (S50856, Yuanye) at a concentration of 20 mg ml⁻¹ and kept at 4 °C up to 1 week. For Cre-mediated recombination was induced at E8.5 and E13.5 prepatent mother, tamoxifen was injected once with 3 mg 40 g⁻¹ body weight. For recombination induced at E16.5 prepatent mother, tamoxifen was injected every 24 h for 3 days with 3 mg 40 g⁻¹ body weight. For newborn pups, 50 μg tamoxifen was intraperitoneally injected every 24 h during four consecutive days starting at postnatal day 0 (P0). Pups were analysed at P7.

## Immunohistochemistry of different tissues

Lung, tail, trunk, heart, thymus and intestine tissues were collected and washed once with PBS. The tissues were then fixed overnight at 4 °C in 4% paraformaldehyde in PBS. Later, 10- or 20-μm thick sections were prepared using a cryostat. The sections were blocked in blocking solution (0.25% triton in PBS + 10% normal donkey serum + 0.1% BSA) for 1 h at room temperature. Immunostainings were performed by incubation of the sections with primary antibodies diluted in the blocking solution overnight at 4 °C, followed by incubation with the secondary antibody diluted in the blocking solution for 2 h at room temperature. Finally, sections were mounted in AquaMount (DAKO). Rabbit anti-Sox2 (Cell Signalling, 23064, 1:400), goat anti-Sox1 (R&D, AF3369, 1:400), rabbit anti-Ki67 (Abcam, ab15580, 1:400), rat anti-MBP (Abcam, ab7349,

1:200), chicken anti-GFAP (Merck, AB5541, 1:1,000), rabbit anti-GFAP (Thermo Fisher Scientific, RB087A1, 1:400), chicken anti-GFP (AVES LABS, AB_2307313, 1:400), mouse anti-Tuj1 (Sigma, T8660, 1:400), rabbit anti-Tuj1 (BioLegend, 802001, 1:400), rabbit anti-MPZ (Thermo Fisher Scientific, PA5-37179, 1:400), rabbit anti-synapsin 1 (Sigma, S193, 1:300), rabbit anti-GABA (Millipore, ABN131, 1:250), rabbit anti-ChAT (Millipore, AB143, 1:250) and rabbit anti-TH (Santa Cruz, sc-14007, 1:250) were used as primary antibodies. Alexa fluorophore-conjugated secondary antibodies (Invitrogen) and DAPI (Invitrogen) were used to visualize primary antibodies and nuclei, respectively. Sections were analysed using a Zeiss LSM 780 confocal microscope. All confocal images are represented as maximum intensity projections unless stated otherwise.

## RNA-seq

RNA quality was verified on an Agilent Bioanalyzer Nano Eukaryote chip. Then, 1 mg of total RNA was used for polyA selection with NEBNext PolyA messenger RNA magnetic isolation module and subsequent cDNA synthesis, Illumina Tru-seq adaptor ligation and library preparation were performed with NEBNext Ultra RNA Library Prep kit (NEB) according to the manufacturer's instructions. Library quality and concentration were determined using an Agilent Bioanalyzer DNA1000 chip and a Qubit fluorometer. Libraries were sequenced on Illumina NextSeq500 in single-end mode. HISAT2 (v.2.2.1)[47] was used to align the RNA-seq reads for each of the samples to the mouse reference genome GRCm38 (https://cloud.biohpc.swmed.edu/index.php/s/grcm38_tran/download) and Cufflinks (v.2.2.1)[48] was used to annotate them. The counts of aligned reads to each gene were calculated with HTSeq (htseq-count v.1.99.2)[49]. In-house software was used to merge the expression results into a single text file used in the downstream analysis in MATLAB (R2020b) software (MathWorks). We performed quantile normalization to equalize the data and stabilized them through the $\log_2$ transform of the data plus one. The heatmap of the most highly variable transcripts, the hierarchical clustering dendrograms (calculated using the unweighted pair group method with arithmetic mean and Euclidean distance measure) and principal-component analyses were performed using in-home functions developed in MATLAB (R2020b) software (MathWorks)[50].

## Single-cell RNA-seq library preparation and sequencing

After FACS sorting, the Nestin-GFP+, Sox1-GFP+ or Sox2-GFP+ cells from brain, lung or tail tissue were counted using a Luna-II automated cell counter (Logos Biosystems) and Chromium (10x Genomics) for Nestin-GFP+ or BD Rhapsody Express system (BD) for Sox1-GFP+ and Sox2-GFP+ were used for the single cell capture and mRNA collection. Single-cell whole transcriptome libraries were prepared according to the manufacturer's instructions using Chromium Single Cell 3′ Library & Gel Bead kit v.3 (10x Genomics) or BD Rhapsody WTA Reagent kit (BD, 633802) and sequenced on the Illumina NextSeq500 using High Output kit v.2.5 (150 cycles, Illumina) aiming sequencing depth of >20,000 reads per cell for each sample.

## Single-cell RNA-seq data analysis

Sequencing data were processed with UMI-tools (v.1.0.1)[51], aligned to the mouse reference genome (mm10) with STAR (v.2.7.1a)[52] and quantified with Subread featureCounts (v.1.6.4)[53]. Alternatively, the FASTQ format of sequencing raw data were processed with BD Rhapsody WTA Analysis pipeline (v.1.11) on SevenBridges Genomics online platform (SevenBridges).

Data normalization, detailed analysis and visualization were performed using Seurat (v.3.1.5)[54] if not specified otherwise. For initial quality control of the extracted gene–cell matrices, we filtered cells with parameters nFeature_RNA > 200 and nFeature_RNA < 4,000 for number of genes per cell and percent.mito < 40 for percentage of mitochondrial genes and genes with parameter min.cell = 3. Filtered matrices were normalized by the logNormalize method with scale factor = 10,000.

Variable genes were found using FindVariableFeatures function with parameters of select.method = 'vst' and nfeatures = 2,000 and then trimmed the genes related to cell cycle (GO:0007049). FindIntegrationAnchors and IntegrateData with default options were used for the data integration and Standard workflow using ScaleData and RunPCA functions were processed. Statistically significant principal components were determined by JackStraw method and used for Uniform Manifold Approximation and Projection nonlinear dimensional reduction.

Unsupervised hierarchical clustering analysis was performed using the FindClusters function in the Seurat package. We tested different resolutions between 0–0.09 in 0.01 increments and 0.1–0.9 in 0.1 increments and selected the final resolution using clustree R package to decide the most stable as well as the most relevant for our previous knowledges. Cellular identity of each cluster was determined by finding cluster-specific marker genes using FindAllMarkers function with parameters of min. pct = 0.25, logfc.threshold = 0.25 and only.pos = TRUE. For subclustering analysis, we isolated specific cluster(s) using subset function, extracted data matrix from the Seurat object using GetAssayData function and repeated the whole analysis pipeline from data normalization.

Differentially expressed genes were identified using the non-parametric Wilcoxon rank-sum test by the FindMarkers function of the Seurat package. FindConservedMarkers function was used to identify canonical cell-type marker genes that are conserved across tissues. Pearson correlation coefficients between clusters were calculated using cor() R function. The FeaturePlot, VlnPlot and DotPlot functions of the Seurat package were used for visualization of selected genes.

The Monocle (v.2.14)[22] R package was used for pseudotime trajectory analysis. All information from the Seurat objects was imported to Monocle CDS objects and then performed dimensionality reduction using its DDRTree method with parameters max_components = 2 and norm_method = 'log'. All cluster-specific markers determined earlier for each cluster were used as ordering genes for the trajectory. GO analysis was performed using PANTHER (v.17.0)[55] Fisher's test with false discovery rate correction.

To exclude non-neural cells in our Sox1-GFP+ scRNA-seq analysis, first, we removed ependymal cells, microglia/macrophages and fibroblasts from the data and re-performed the analysis. Then, we excluded Krt1/Krt10 double-positive lung epithelial cells and Olig2/Plp1 double-positive OPC/oligodendrocytes. Finally, we used only the 'NSC' populations to do the cluster analysis.

## Statistics and reproducibility

No statistical method was used to predetermine sample size. The sample size in different experimental groups was determined based on previous studies or preliminary studies. No data were excluded from the analyses. Data distribution was assumed to be normal but this was not formally tested. The statistical test types and biological $n$ numbers used are detailed in the relevant figure captions. The Investigators were not blinded to allocation during experiments and outcome assessment.

## Reporting summary

Further information on research design is available in the Nature Portfolio Reporting Summary linked to this article.

## Data availability

The microarray, bulk RNA-seq and single-cell RNA-seq data in this study are available from Gene Expression Omnibus under accession numbers GSE151649, GSE213158 and GSE213133. The publicly available datasets used in this study are available from Gene Expression Omnibus under accession numbers GSE30500, GSE131498 (GSM3781644) and GSE211713 (GSM6499593). The mouse reference genome for RNA-seq analysis is GRCm38 (https://cloud.biohpc.swmed.edu/index.php/s/grcm38_tran/download). All data supporting the findings of this study are available from the corresponding author on reasonable request. Source data are provided with this paper.

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

## Acknowledgements

We thank M. Haustein for tissue section preparation; K. Müller for single-cell RNA-seq experiments; L. Lüken for FACS experiments; M. Sinn for microarray sample preparation; and V. Episkopou, R. Lovell-Badge, S. Malas, G. Lapathitis, S. Gao, J. Chen, T. Braun, J. Kim, RIKEN BioResource Research Center, Centre for Comparative Medicine Research of the University of Hong Kong for help with transgenic mice. Finally, we thank A. Malapetsas for proofreading. This work was funded by the Max Planck Society's White Paper-Project 'Animal testing in the Max Planck Society'. This work was supported by Health@InnoHK, Innovation Technology Commission.

## Author contributions

D.H. conceived the study, performed the experiments, interpreted the data, and wrote the manuscript. W.X. performed the immunohistochemistry experiments; H.W.J. and Y.C. performed the scRNA-seq experiments and analysed the data; H.P., K.W. and J.C.S. performed the in vivo transplantation experiments; Y.T. and J.K. performed the electrophysiological experiments; M.S. performed the FACS sorting experiments; G.W. and G.L. performed the mice experiments; K.P.K. performed the bisulfite sequencing experiments; H.R. performed in vitro NSC differentiation experiments; D.W.H. performed karyotyping experiment; M.J.A.-B. and D.G. analysed the microarray and bulk RNA-seq datasets; R.H.A. supervised the scRNA-seq and transplantation experiments; P.L. supervised the immunohistochemistry and mice experiments; and H.R.S. supervised the study and edited the manuscript.

## Funding

## Competing interests

The authors declare no competing interests.

## Additional information

**Extended data** is available for this paper at https://doi.org/10.1038/s41556-025-01641-w.

**Correspondence and requests for materials** should be addressed to Dong Han, Pentao Liu or Hans R. Schöler.

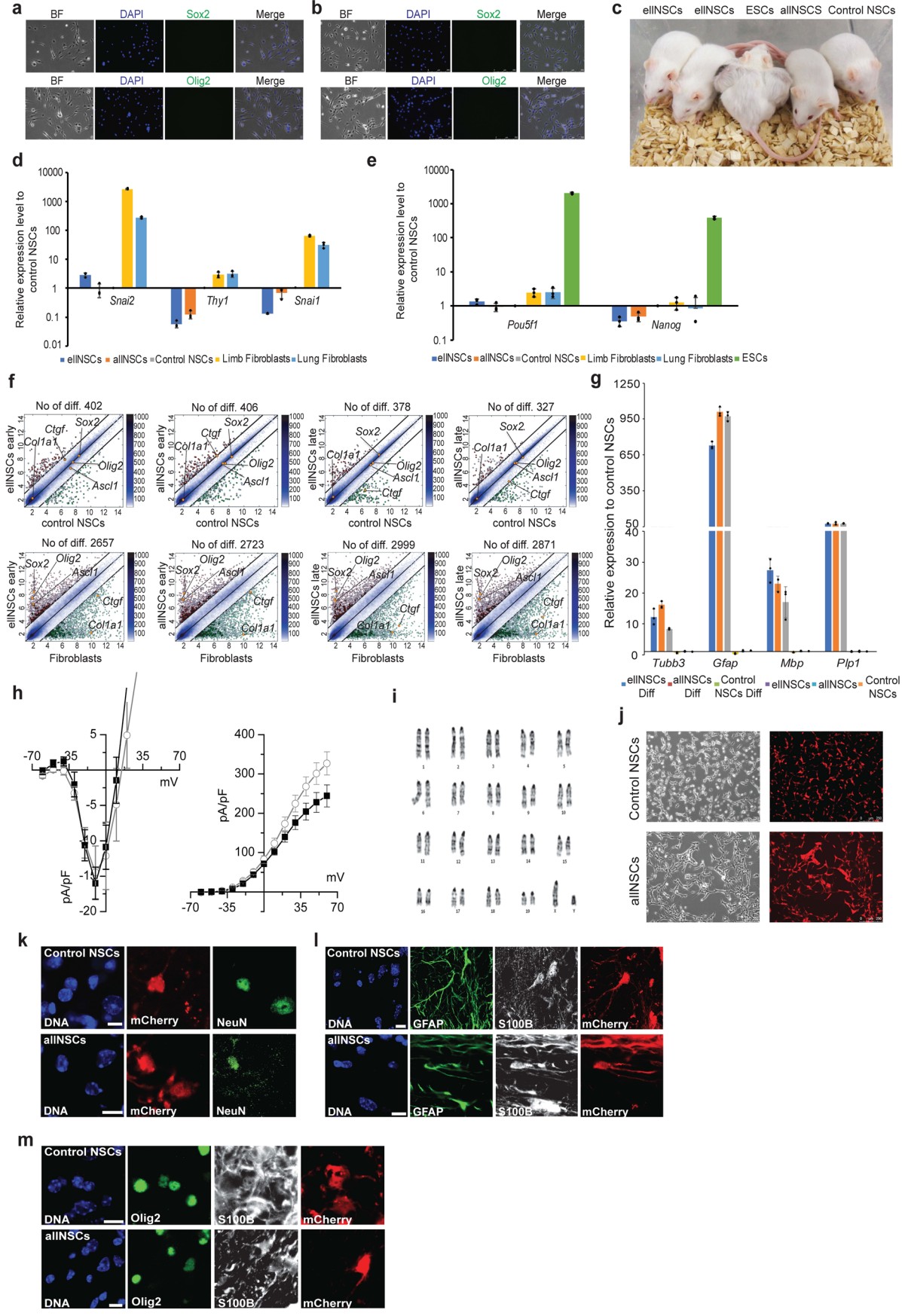

**Extended Data Fig. 1 | See next page for caption.**

**Extended Data Fig. 1 | Derivation, characterization and multipotency assessment of ldNSCs. a,b**, Morphology and immunofluorescence microscopy images of embryonic limb cells (**a**) and adult lung cells (**b**) cultured in NSC medium. **c**, ellNSCs and allNSCs, control NSCs and ESCs were injected into SCID mice. Only ESCs, but not ellNSCs, allNSCs and control NSCs, formed teratomas 4 weeks after injection. **d,e**, Expression levels of somatic (**d**) and pluripotency genes (**e**) in different samples. All data are calibrated to brain NSCs, whose expression is considered to be 1 for all genes. The data represent means ± s.d. ($n = 3$ biological replicates). **f**, Pairwise scatter plots of the global gene expression microarray profiles of ellNSCs and allNSCs versus control NSCs and fibroblasts. The bar to the right indicates the scattering density; the higher the scattering density, the darker the blue colour. Gene expression levels are depicted in $log_2$ scale. The number of differentially expressed genes is indicated on top of each scatter plot. $n = 2$ biological replicates, the depicted results are an integration of data derived from all different biboloical samples. **g**, RT-qPCR analysis of indicated markers in spontaneous differentiated cells. All data are calibrated to brain NSCs, whose expression is considered to be 1 for all genes. The data represent means ± s.d. ($n = 3$ biological replicates). **h**, I-V curves of inward (left) and outward (right) components of transmembrane currents in ldNSCs (open grey circles, $n = 25$) and control NSCs (solid black squares, $n = 15$) derived neurons. The data represent means ± SEM. **i**, Karyotypic analysis of allNSCs. **j**, allNSCs and control NSCs were labelled with mCherry. Data represent 3 biological replicates. **k-m**, The in vivo differentiation potential of transplanted control NSCs and allNSCs were determined by immunostaining for mCherry and cell fate specific markers. Differentiating mCherry⁺/NeuN⁺ neurons (**k**), mCherry⁺/GFAP⁺/S100B⁺ astrocytes (**l**), and mCherry⁺/Olig2⁺/S100B⁺ oligodendrocytes (**m**) were observed. Data represent 3 biological replicates. Scale bar, 10 μm.

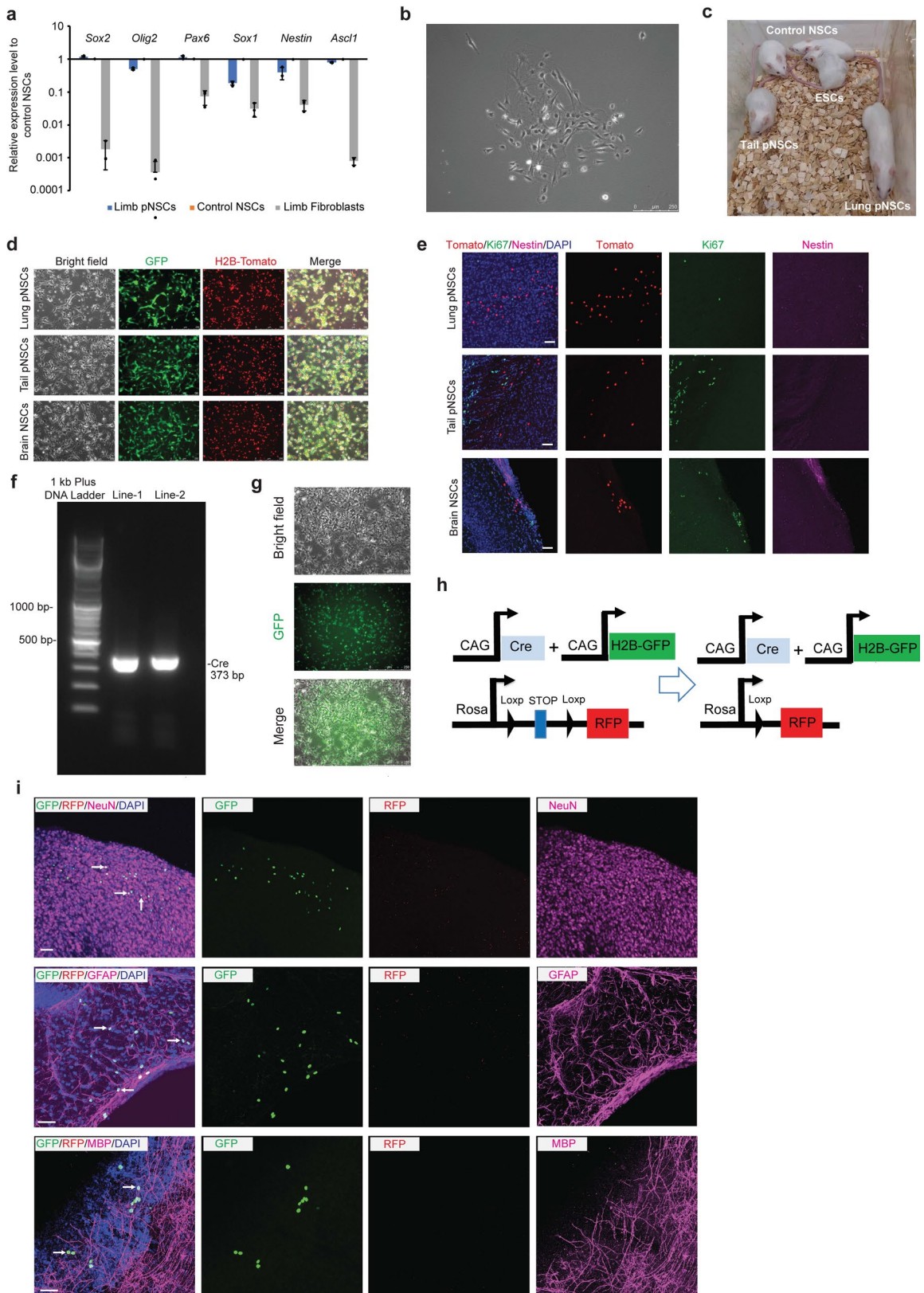

Extended Data Fig. 2 | See next page for caption.

**Extended Data Fig. 2 | Characterization of pNSCs derived without low-pH treatment. a**, Expression levels of NSC marker genes in different samples quantified by RT-qPCR. All data are calibrated to brain NSCs, whose expression is considered to be 1 for all genes. The data represent means ± s.d. ($n$ = 3 biological replicates). **b**, Representative morphology of Nes-GFP⁺ cells after culture, as assessed by bright-field microscopy. **c**, Lung pNSCs, tail pNSCs, brain control NSCs and mouse ESCs were injected into SCID mice. Only ESCs, but not lung pNSCs, tail pNSCs and brain NSCs, formed teratomas 4 weeks after injection. **d**, Nes-GFP⁺ lung pNSCs, tail pNSCs and brain NSCs were labelled with CAG-H2B-Tomato. Data represent 3 biological replicates. **e**, Immunohistological analysis of the transplanted cells 6 weeks after transplantation using antibodies against Tomato, Ki67 and Nestin. Data represent 3 biological replicates. Scale bar, 50 µm. **f**, Genotyping of Cre recombinase by polymerase chain reaction (PCR) in tail pNSCs from WT mouse. **g**, Postnatal WT mouse tail pNSCs were labelled with CAG-H2B-GFP. Data represent 3 biological replicates. **h**, The genetic approach to trace cell fusion between transplanted cells and endogenous cells. **i**, The transplanted tail pNSCs labelled with CAG-H2B-GFP were RFP⁻ and differentiated into mature neurons (NeuN⁺/GFP⁺), astrocytes (GFAP⁺/GFP⁺), and oligodendrocytes (MBP⁺/GFP⁺). White arrows indicate the differentiated neurons, astrocytes and oligodendrocytes. Data represent 3 biological replicates. Scale bar, 50 µm.

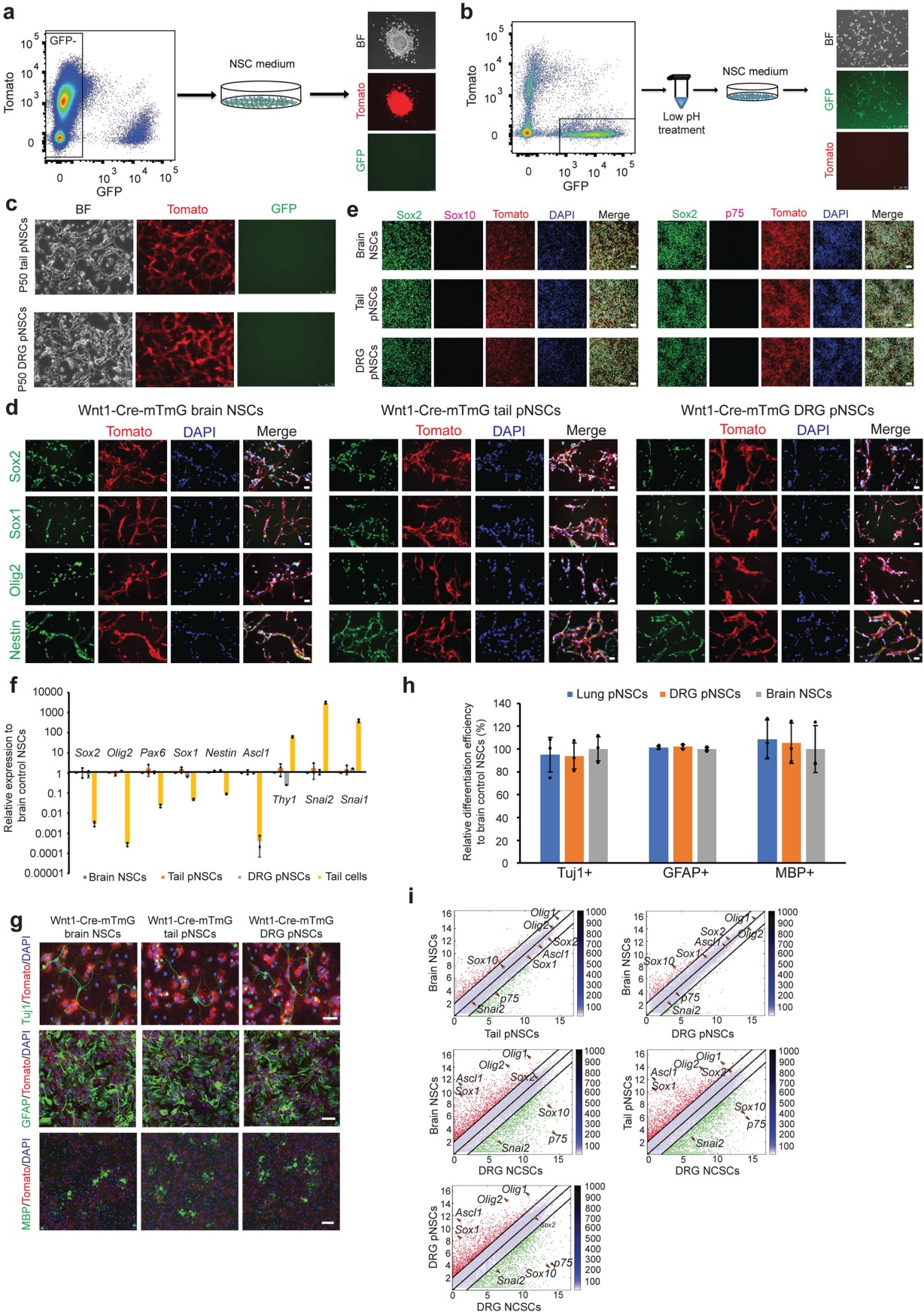

Extended Data Fig. 3 | See next page for caption.

**Extended Data Fig. 3 | pNSCs do not originate from NCCs and characterization of Wnt1-Cre-mTmG pNSCs. a**, FACS strategy for postnatal tail GFP⁻ cells from Wnt1-Cre-mTmG mouse, and morphology of a primary tail pNSC cluster from GFP⁻ cells, as assessed by bright-field (BF) microscopy, Tomato and GFP signals. **b**, FACS strategy for postnatal DRG GFP⁺ cells from Wnt1-Cre-mTmG mouse, and morphology of GFP⁺ cells after culture in NSC medium, as assessed by bright-field (BF) microscopy, Tomato and GFP signals. **c**, Postnatal tail and DRG pNSCs could be stably maintained for more than 50 passages in monolayer. **d**, Immunofluorescence microscopy images of brain NSCs, tail pNSCs, and DRG pNSCs from Wnt1-Cre-mTmG mouse using antibodies against Sox2, Sox1, Olig2 and Nestin. Scale bar, 50 μm. **e**, Immunofluorescence microscopy images of brain NSCs, tail pNSCs, and DRG pNSCs from Wnt1-Cre-mTmG mouse using antibodies against Sox2, Sox10 and p75. Scale bar, 50 μm. **f**, Expression levels of marker genes in different samples quantified by RT-qPCR. All data are calibrated to brain NSCs, whose expression is considered to be 1 for all genes. The data represent means ± s.d. ($n$ = 3 biological replicates). **g**, Brain NSCs, postnatal tail and DRG pNSCs differentiated into neurons, astrocytes, and oligodendrocytes, as determined by immunocytochemistry with antibodies against Tuj1, GFAP and MBP. Scale bar, 50 μm. **h**, The in vitro differentiation efficiencies of pNSCs and brain NSCs from Wnt1-Cre-mTmG mouse into neurons, astrocytes, and oligodendrocytes were quantified and compared via immunostaining with Tuj1, GFAP, and MBP, respectively. Brain control NSCs were used as a positive control for determining the relative differentiation efficiency. The data represent means ± s.d. ($n$ = 3 biological replicates). **i**, Pairwise scatter plots of the global gene expression RNA-seq profiles of different samples. $n$ = 3 biological replicates, the depicted results are an integration of data derived from all different biological samples.

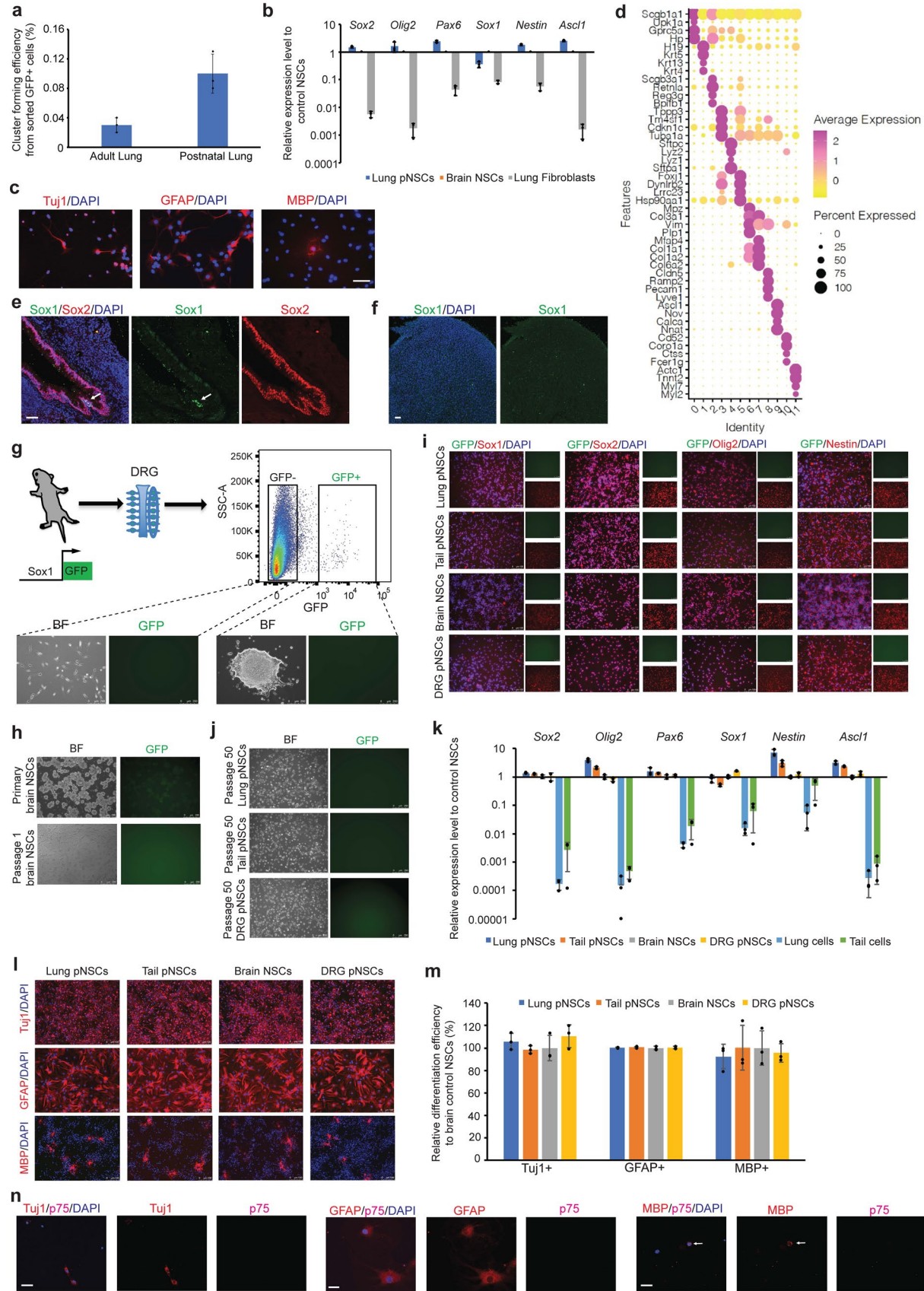

**Extended Data Fig. 4 | See next page for caption.**

**Extended Data Fig. 4 | Characterization of Sox2-GFP⁺ and Sox1-GFP⁺ pNSCs.**
**a**, NSC-like cluster forming efficiencies from sorted adult and postnatal lung Sox2-GFP⁺ cells. The data represent means ± s.d. (*n* = 3 biological replicates).
**b**, Expression levels of NSC marker genes in different samples quantified by RT-qPCR. All data are calibrated to brain NSCs, whose expression is considered to be 1 for all genes. The data represent means ± s.d. (*n* = 3 biological replicates).
**c**, Lung pNSCs from postnatal Sox2-GFP mouse could differentiate into neurons (Tuj1⁺), astrocytes (GFAP⁺) and oligodendrocytes (MBP⁺) in vitro. Scale bar, 50 μm. **d**, Dot plot showing the top 4 markers for each cluster in postnatal Sox2-GFP⁺ lung cells. Dot size represents percentage of cells where the gene is detected, colour indicates average expression level of the gene in each cluster. *n* = 1 biological replicate. **e**, Immunohistological analysis of lung tissue from adult WT mouse using antibody against Sox1, Sox2. Data represent 3 biological replicates. Scale bar, 50 μm. **f**, Immunohistological analysis of heart tissue from postnatal WT mouse using antibody against Sox1. Data represent 3 biological replicates. Scale bar, 50 μm. **g**, FACS strategy for postnatal DRG GFP⁺ and GFP⁻ cells from Sox1-GFP mouse, and morphology of a primary DRG pNSC cluster and GFP⁻ cells after culture, as assessed by bright-field (BF) microscopy and GFP signal. **h**, Primary and passage 1 brain NSCs from postnatal Sox1-GFP mouse,

as assessed by bright-field (BF) microscopy and GFP signal. Data represent 3 biological replicates. **i**, Immunofluorescence microscopy images of lung pNSCs, tail pNSCs and brain NSCs from postnatal Sox1-GFP mouse using antibodies against Sox1, Sox2, Olig2 and Nestin. **j**, Lung and DRG pNSCs, and tail pNSCs from postnatal Sox1-GFP mice could be stably maintained for more than 50 passages in monolayer. Data represent 3 biological replicates. **k**, Expression levels of NSC marker genes in different samples quantified by RT-qPCR. All data are calibrated to brain NSCs, whose expression is considered to be 1 for all genes. The data represent means ± s.d. (*n* = 3 biological replicates). **l**, Lung, tail and DRG pNSCs and brain NSCs from postnatal Sox1-GFP mouse could differentiate into neurons (Tuj1⁺), astrocytes (GFAP⁺) and oligodendrocytes (MBP⁺) in vitro. **m**, The in vitro differentiation efficiencies of pNSCs and brain NSCs from Sox1-GFP mouse into neurons, astrocytes, and oligodendrocytes were quantified and compared via immunostaining with Tuj1, GFAP, and MBP, respectively. Brain control NSCs were used as a positive control for determining the relative differentiation efficiency. The data represent means ± s.d. (*n* = 3 biological replicates). **n**, Sox1-GFP⁺ cells without expansion with growth factors EGF and FGF-2 could differentiate into neurons, astrocytes and oligodendrocytes in vitro. White arrows indicate the MBP⁺ oligodendrocytes. Data represent 3 biological replicates. Scale bar, 50 μm.

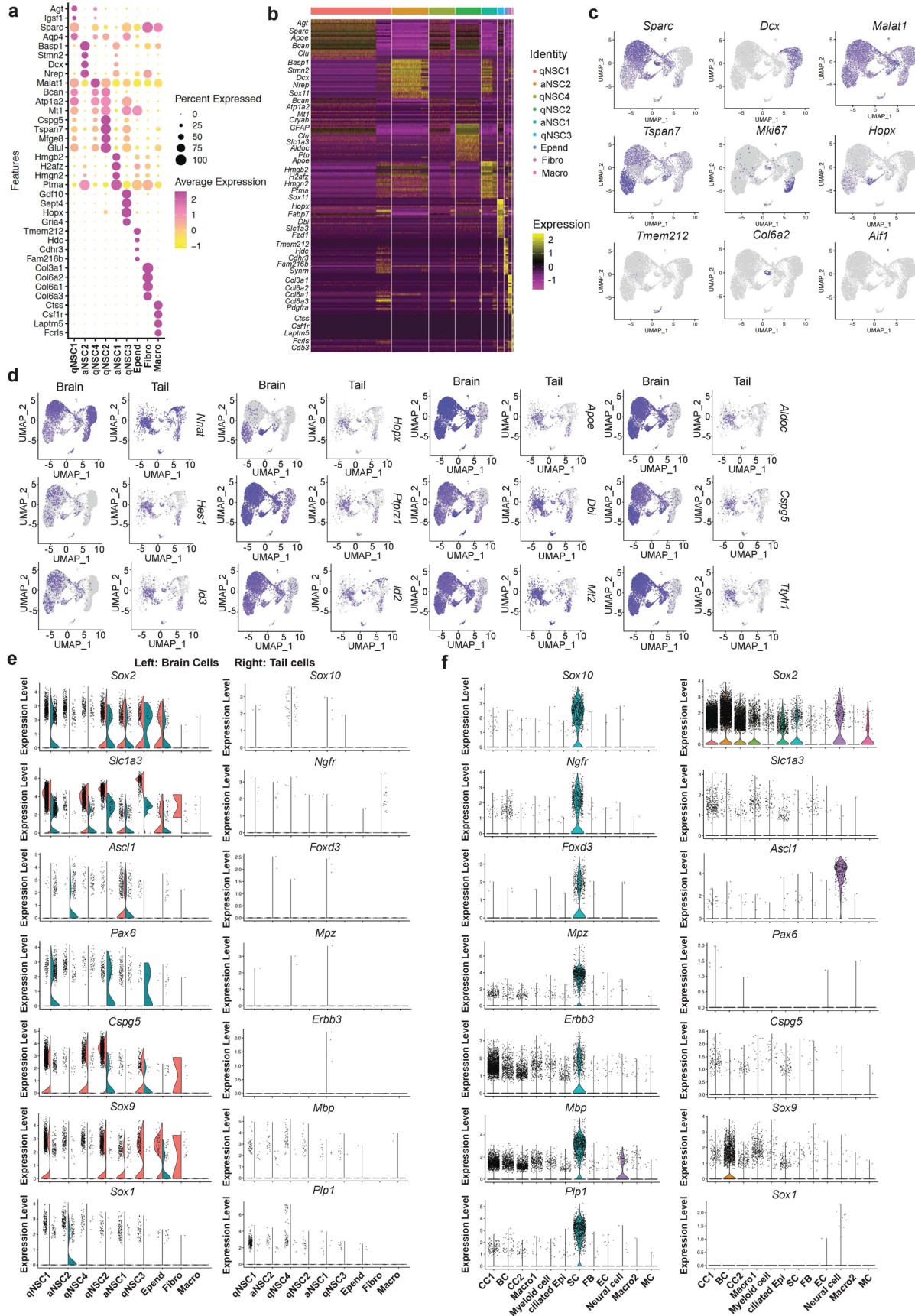

**Extended Data Fig. 5 | See next page for caption.**

**Extended Data Fig. 5 | scRNA-seq analysis of Sox1-GFP⁺ pNSCs. a**, Dot plot showing the top 4 markers for each cluster in all Sox1-GFP⁺ cells. **b**, Heatmap of the expression of the top 50 marker genes in different Sox1⁺ cell populations. Each column represents a cell and each row represents a gene. **c**, UMAP showing the expression of representative markers for each population in Sox1-GFP⁺ cells.

**d**, UMAP showing the expression of NSC markers in brain and tail Sox1-GFP⁺ cells. **e,f**, Violin plots showing the expression of NSC and Schwann cell/NC-derived cell marker genes in different populations of brain, tail Sox1-GFP⁺ cells **(e)** and Sox2-GFP⁺ lung Schwann cells **(f)**. *n* = 1 biological replicate for all the scRNA-seq samples.

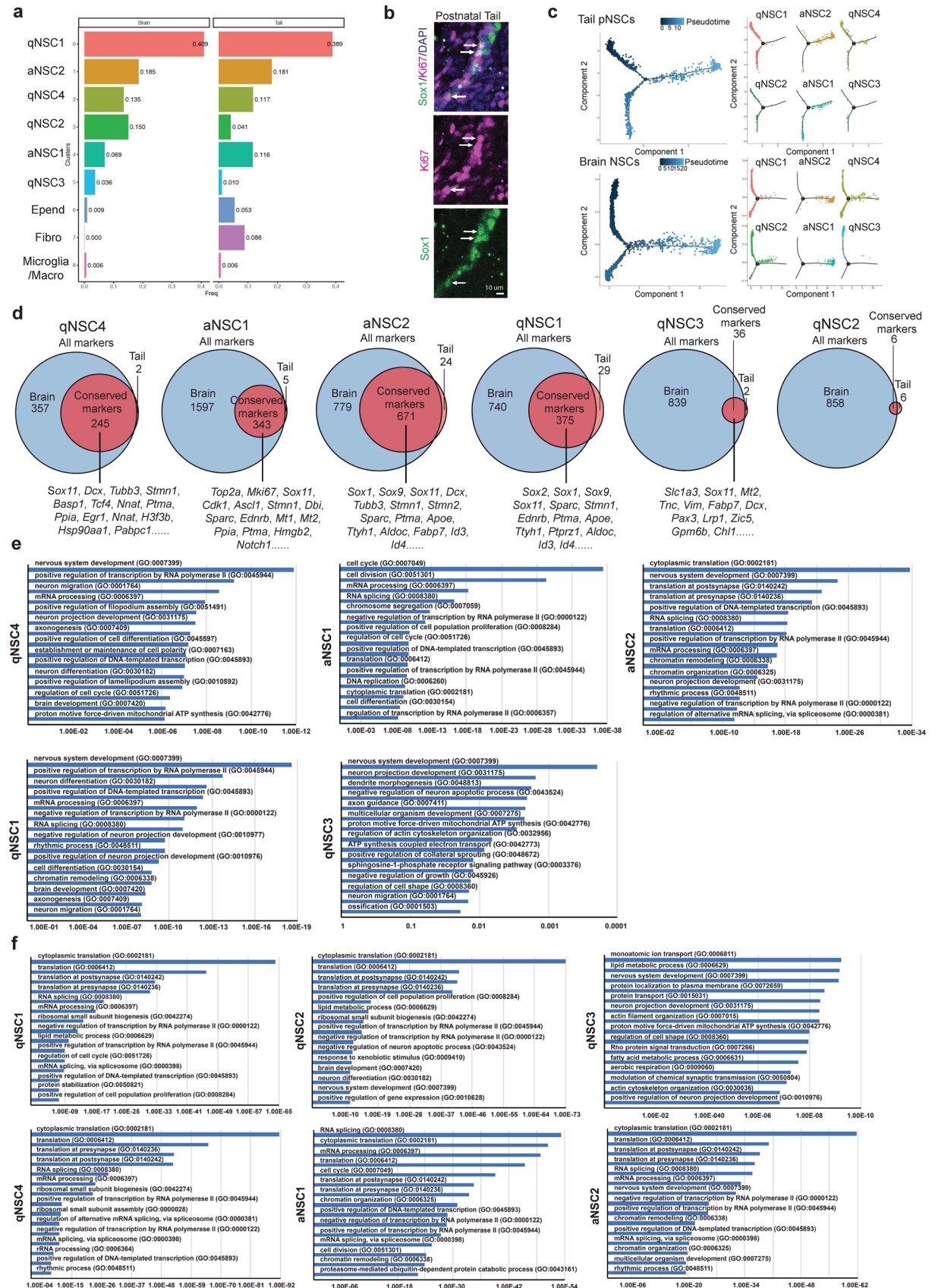

**Extended Data Fig. 6 | Comparison of Sox1-GFP⁺ pNSCs with brain NSCs. a**, Bar chart indicating percentage of each cluster cells. **b**, Immunohistological analysis of postnatal WT mouse tail tissue using antibodies against Sox1 and Ki67. Scale bar, 10 µm. White arrows indicate double-positive cells. **c**, Pseudotemporal ordering of brain and tail qNSC, aNSC cells. **d**, Venn diagram of conserved markers in different clusters between brain NSCs and tail pNSCs. **e**, Top 15 GO pathways (biological processes) of conserved markers between pNSCs and brain NSCs across each cluster. **f**, Top 15 GO pathways (biological processes) of brain-only gene markers. *n* = 1 biological replicate for all the scRNA-seq samples.

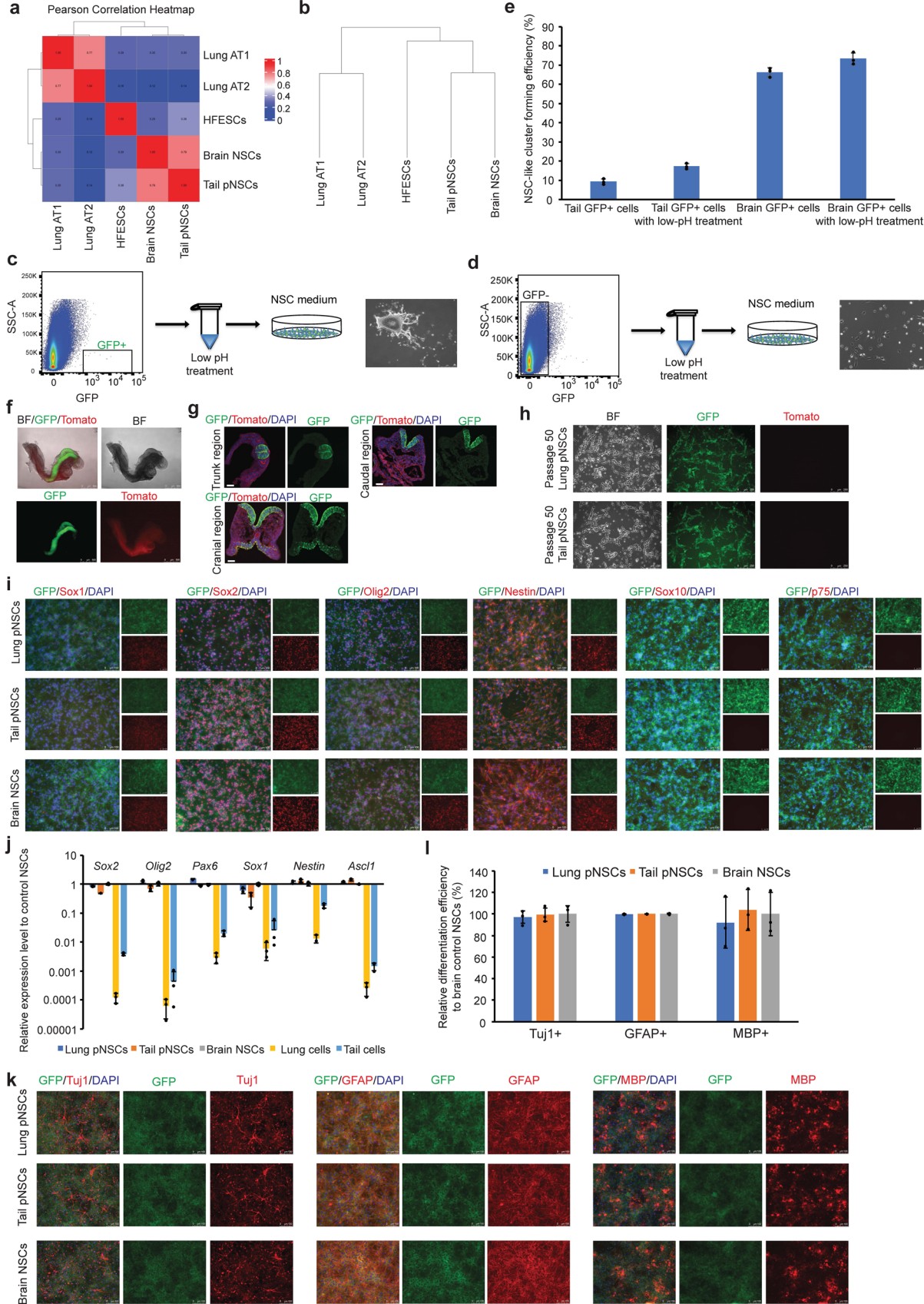

**Extended Data Fig. 7 | See next page for caption.**

**Extended Data Fig. 7 | Comparison of pNSCs with other tissue stem cells and the origin of ldNSCs and pNSCs. a**,**b**, Pearson correlation (**a**) and hierarchical clustering (**b**) of tail pNSCs with brain NSCs, lung AT1 and AT2 stem cells, and hair follicle epithelium stem cells. $n = 1$ biological replicate. **c,d**, FACS strategy for postnatal lung Sox1-GFP$^+$ (**c**) and Sox1-GFP$^-$ (**d**) cells, and representative morphologies of pNSC cluster and Sox1-GFP$^-$ cells after low-pH treatment and culture, as assessed by bright-field (BF) microscopy. **e**, NSC-like cluster forming efficiency from the postnatal tail and brain Sox1-GFP$^+$ cells with or without low-pH treatment. The data represent means ± s.d. ($n = 3$ biological replicates). **f**, In Sox1-Cre-mTmG embryos, GFP expression was detected in the neuroepithelial cells at E8.5, as assessed by bright-field (BF) microscopy, GFP and Tomato signals. Data represent 3 biological replicates. **g**, Immunohistological analysis of E8.5 Sox1-Cre-mTmG embryos using antibody against GFP. Data represent 3 biological replicates. Scale bar, 50 μm. **h**, Lung pNSCs and tail pNSCs from postnatal Sox1-Cre-mTmG mouse could be stably maintained for more than 50 passages in monolayer, as assessed by bright-field (BF) microscopy, GFP and Tomato signals. **i**, Immunofluorescence microscopy images of lung pNSCs, tail pNSCs and brain NSCs from postnatal Sox1-Cre-mTmG mouse using antibodies against Sox1, Sox2, Olig2, Nestin, Sox10 and p75. **j**, Expression levels of NSC marker genes in different samples quantified by RT-qPCR, normalized to brain NSCs. The data represent means ± s.d. ($n = 3$ biological replicates). **k**, Lung pNSCs, tail pNSCs and brain NSCs from postnatal Sox1-Cre-mTmG mouse could differentiate into neurons (Tuj1$^+$), astrocytes (GFAP$^+$) and oligodendrocytes (MBP$^+$) in vitro. **l**, The in vitro differentiation efficiencies of pNSCs and brain NSCs from Sox1-Cre-mTmG mouse into neurons, astrocytes, and oligodendrocytes were quantified and compared via immunostaining with Tuj1, GFAP, and MBP, respectively. Brain control NSCs were used as a positive control for determining the relative differentiation efficiency. The data represent means ± s.d. ($n = 3$ biological replicates).

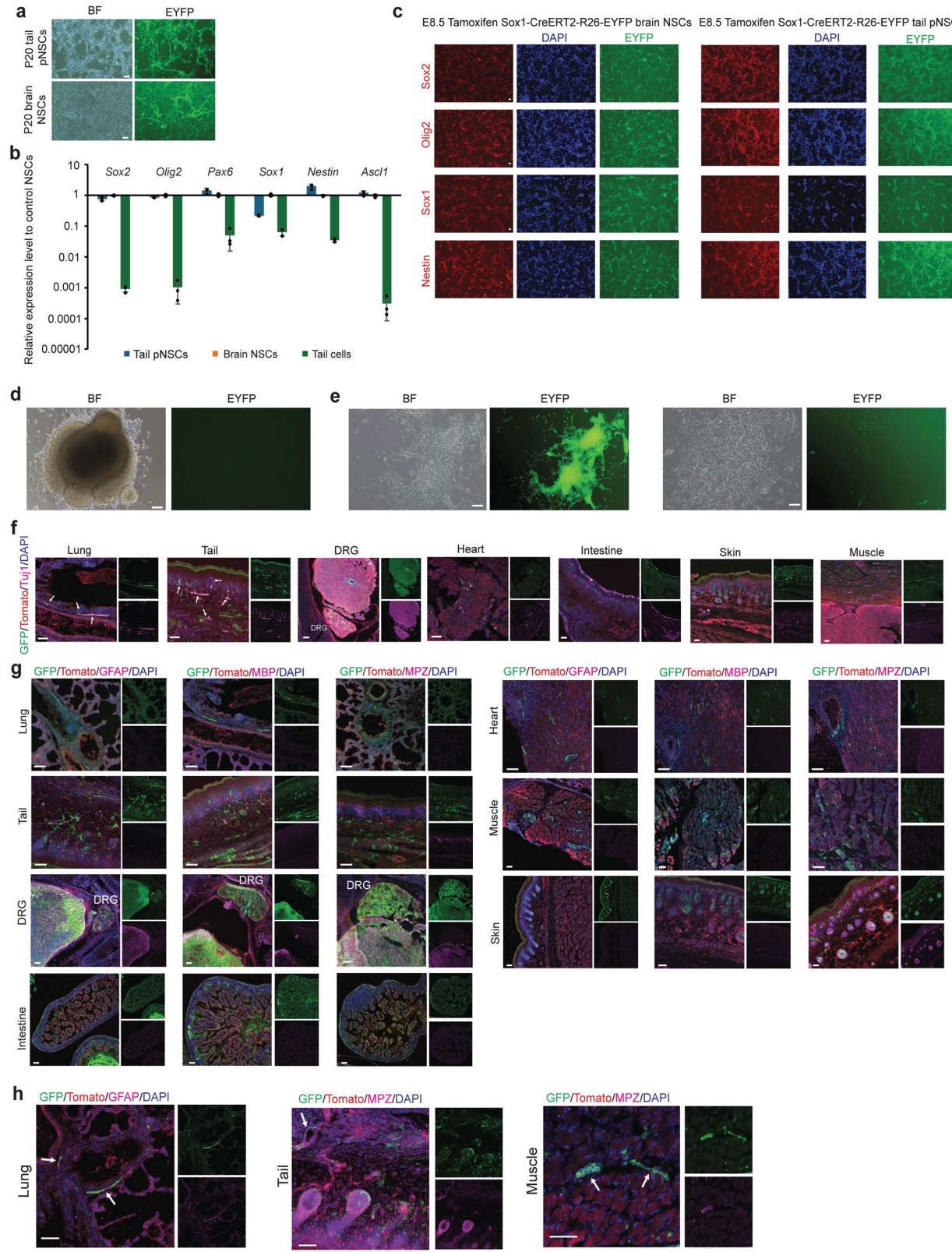

**Extended Data Fig. 8 | See next page for caption.**

**Extended Data Fig. 8 | pNSCs originate from early NECs and differentiate into neural derivatives during embryonic development. a**, EYFP⁺ tail pNSCs from Sox1-CreERT2-R26-EYFP mouse with tamoxifen injection only once at E8.5 could be stably maintained for more than 20 passages in monolayer. **b**, Expression levels of NSC marker genes in different samples quantified by RT-qPCR, normalized to brain NSCs. The data represent means ± s.d. ($n$ = 3 biological replicates). **c**, Immunofluorescence microscopy images of brain NSCs and tail pNSCs from Sox1-CreERT2-R26-EYFP mouse with tamoxifen injection only once at E8.5 using antibodies against Sox2, Olig2, Sox1 and Nestin. Scale bar, 50 μm. **d**, Representative morphologies of primary tail EYFP⁻ pNSC cluster from Sox1-CreERT2-R26-EYFP mouse with tamoxifen injection only once at E8.5, as assessed by bright-field (BF) microscopy, and EYFP signal. Data represent 3 biological

replicates. **e**, Representative morphologies of primary brain EYFP⁺ and EYFP⁻ NSC clusters from Sox1-CreERT2-R26-EYFP mouse with tamoxifen injection only once at E8.5, as assessed by bright-field (BF) microscopy, and EYFP signal. Data represent 3 biological replicates. **f**, Immunohistological analysis of different tissues from postnatal Sox1-Cre-mTmG mouse using antibody against GFP and Tuj1. Data represent 3 biological replicates. Scale bar, 50 μm. **g**, Immunohistological analysis of different tissues from postnatal Sox1-Cre-mTmG mouse using antibody against GFP, GFAP and MPZ. Data represent 3 biological replicates. Scale bar, 50 μm. **h**, Immunohistological analysis of different tissues from postnatal Sox1-Cre-mTmG mouse using antibody against GFP, GFAP, MBP and MPZ. Data represent 3 biological replicates. Scale bar, 50 μm.

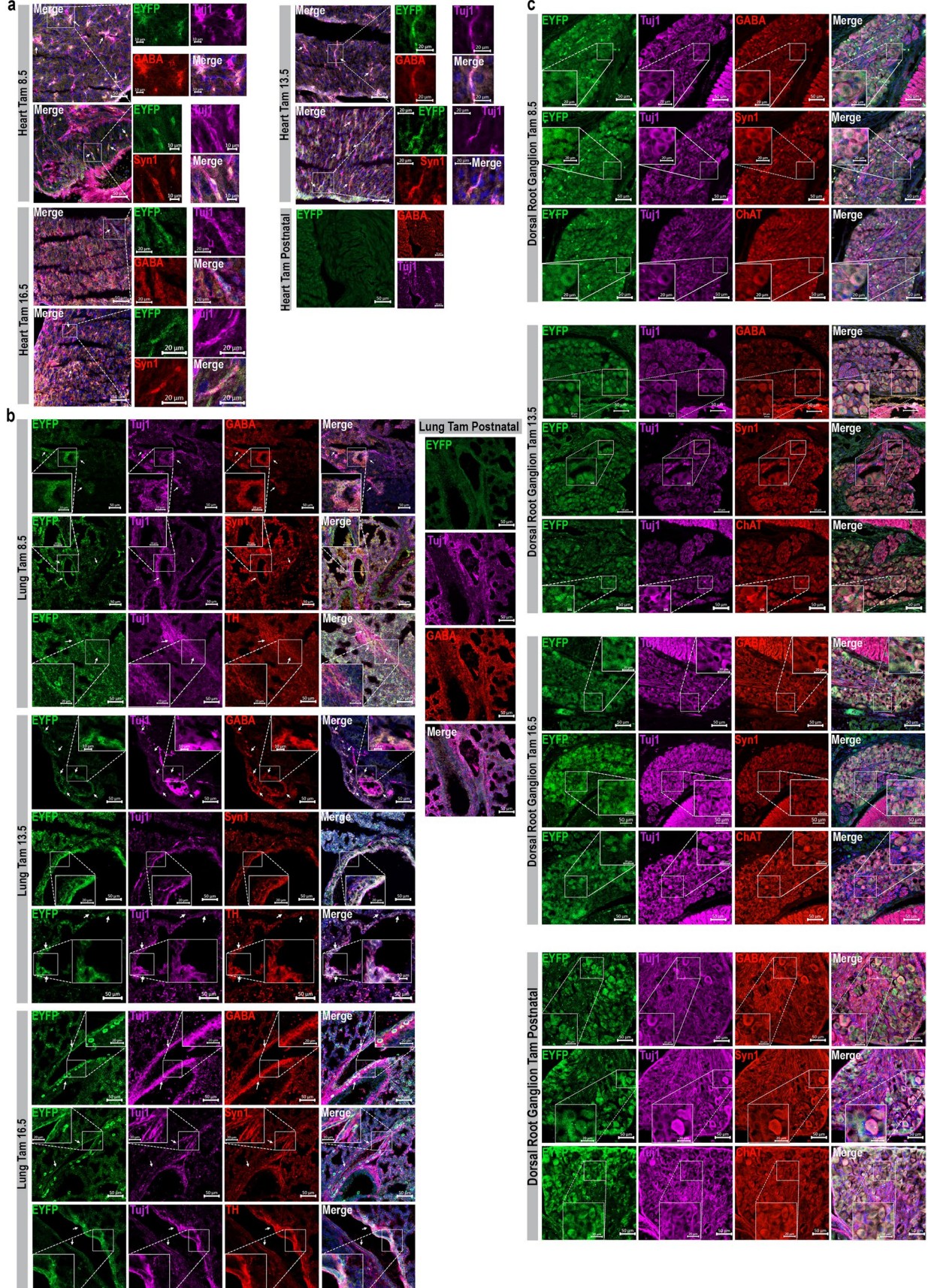

**Extended Data Fig. 9 | Temporal profiling of pNSCs' contribution to mature neurons in the heart, lung and DRG. a,b,c,** Immunohistological analysis of heart (**a**), lung (**b**) and DRG (**c**) from Sox1-CreERT2-R26-EYFP mouse with tamoxifen injection using antibodies against EYFP, Tuj1, GABA, Syn1 and TH. Data represent 3 biological replicates. TH, tyrosine hydroxylase.

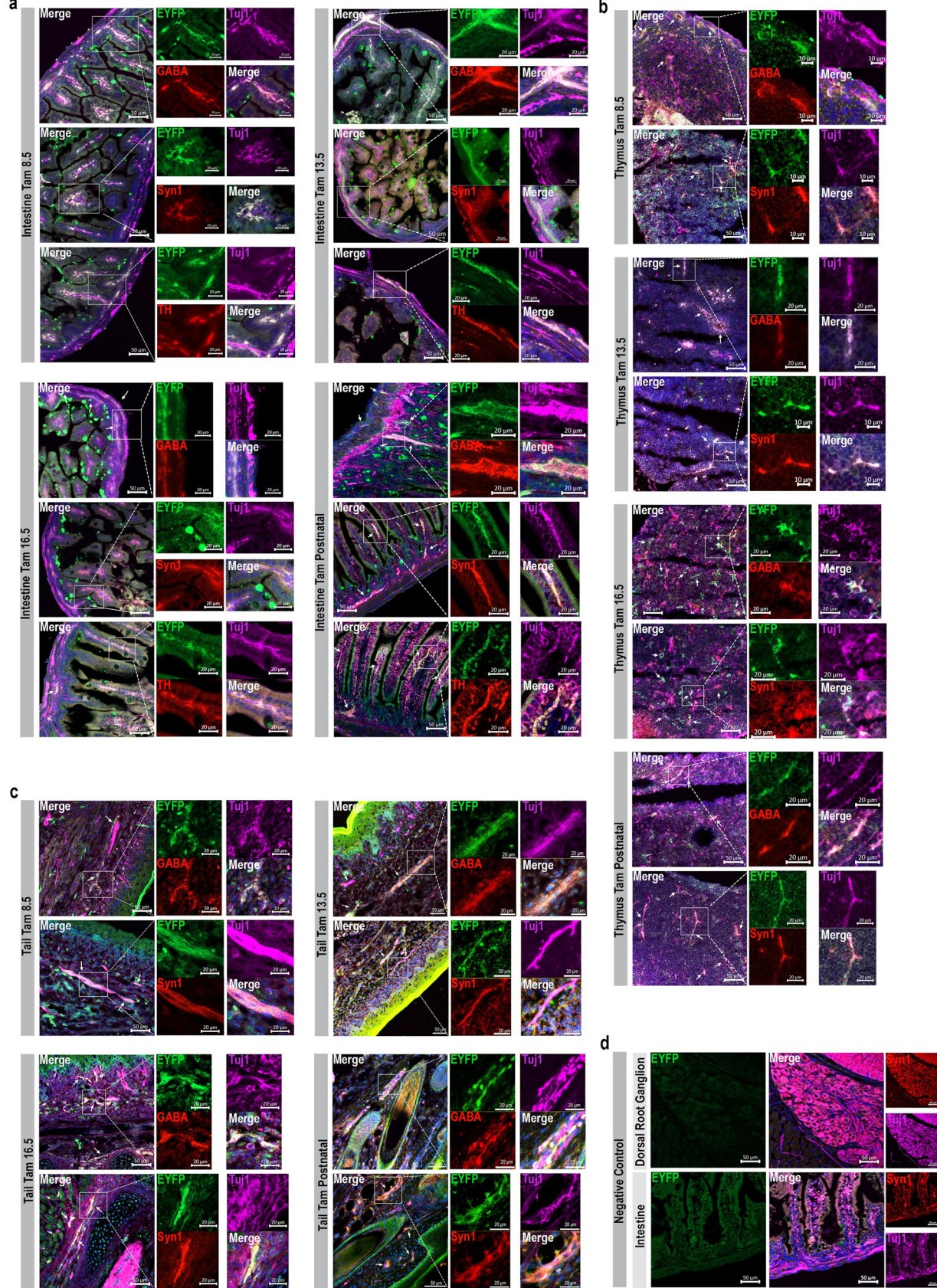

**Extended Data Fig. 10 | Temporal profiling of pNSCs' contribution to mature neurons in the intestine, thymus and tail. a,b,c** Immunohistological analysis of intestine (**a**), thymus (**b**) and tail (**c**) from Sox1-CreERT2-R26-EYFP mouse using antibodies against EYFP, Tuj1, ChAT, GABA, Syn1 and TH. Data represent 3 biological replicates. **d**, Immunohistological analysis of intestine and DRG tissues in Sox1-CreERT2-R26-EYFP mouse without tamoxifen injection. Data represent 3 biological replicates.

# nature research

# Reporting Summary

Nature Research wishes to improve the reproducibility of the work that we publish. This form provides structure for consistency and transparency in reporting. For further information on Nature Research policies, see our Editorial Policies and the Editorial Policy Checklist.

## Statistics

For all statistical analyses, confirm that the following items are present in the figure legend, table legend, main text, or Methods section.

| n/a | Confirmed | |
|---|---|---|
| ☐ | ☒ | The exact sample size (*n*) for each experimental group/condition, given as a discrete number and unit of measurement |
| ☐ | ☒ | A statement on whether measurements were taken from distinct samples or whether the same sample was measured repeatedly |
| ☒ | ☐ | The statistical test(s) used AND whether they are one- or two-sided<br>*Only common tests should be described solely by name; describe more complex techniques in the Methods section.* |
| ☒ | ☐ | A description of all covariates tested |
| ☒ | ☐ | A description of any assumptions or corrections, such as tests of normality and adjustment for multiple comparisons |
| ☐ | ☒ | A full description of the statistical parameters including central tendency (e.g. means) or other basic estimates (e.g. regression coefficient) AND variation (e.g. standard deviation) or associated estimates of uncertainty (e.g. confidence intervals) |
| ☐ | ☒ | For null hypothesis testing, the test statistic (e.g. *F*, *t*, *r*) with confidence intervals, effect sizes, degrees of freedom and *P* value noted<br>*Give P values as exact values whenever suitable.* |
| ☒ | ☐ | For Bayesian analysis, information on the choice of priors and Markov chain Monte Carlo settings |
| ☒ | ☐ | For hierarchical and complex designs, identification of the appropriate level for tests and full reporting of outcomes |
| ☒ | ☐ | Estimates of effect sizes (e.g. Cohen's *d*, Pearson's *r*), indicating how they were calculated |

*Our web collection on statistics for biologists contains articles on many of the points above.*

## Software and code

Policy information about availability of computer code

| Data collection | Applied Biosystems 7500; Leica DMI6000B inverted fluorescence microscope equipped with a Hamamatsu Orca-R2 charge-coupled device camerafor capturing immunoflourescence images; Zeiss LSM 710 confocal microscope; Zeiss LSM 780 confocal microscope; FACS Canto (BD Biosciences) or FACS Aria II (BD Biosciences); Chromium (10X Genomics); BD Rhapsody Express system (BD); Illumina NextSeq500. |
|---|---|
| Data analysis | iScan (v 1.3), BeadStudio (v 3.2), LAS AF (v.3.2.0), ZEN (3.9) software, Adobe Photoshop, ImageJ (2.1.0/1.53j), Microsoft Excel, UMI-tools (version1.0.1), STAR (version 2.7.1a), Subread featureCounts (version 1.6.4), Seurat (version 3.1.5), BD Rhapsody WTA Analysis pipeline (version 1.11), Monocle (version 2.14), PANTHER (17.0), eBD FACS Diva (8.0.1), FlowJo software (10.6.1), Molecular Signatures Database (MSigDB) (3.0), MATLAB® (MathWorks®) software (R2020b) , BeadStudio (3.2) (Illumina), QUMA, PatchMaster (2.4) software, Patcher's Power Tool routine for IgorPro (WaveMetrics, Lake Oswego, OR, USA) and SciDAVis program (http://scidavis.sourceforge.net/index.html), Molecular Signatures Database (MSigDB) (3.0),eBD FACS Diva (8.01), HISAT2 (v2.2.1), Cufflinks (v.2.2.1), HTSeq (htseq-count v1.99.2), lumiExpresso (2.24.0), R-Bioconductor (3.0). |

For manuscripts utilizing custom algorithms or software that are central to the research but not yet described in published literature, software must be made available to editors and reviewers. We strongly encourage code deposition in a community repository (e.g. GitHub). See the Nature Research guidelines for submitting code & software for further information.

## Data

Policy information about availability of data

All manuscripts must include a data availability statement. This statement should provide the following information, where applicable:

- Accession codes, unique identifiers, or web links for publicly available datasets
- A list of figures that have associated raw data
- A description of any restrictions on data availability

The accession numbers for the microarray, bulk RNA-seq, and single-cell RNA-seq data in this study are available from Gene Expression Omnibus under accession number GEO: GSE151649, GSE213158, GSE213133. The publicly available datasets used in this study are available from Gene Expression Omnibus under accession number GEO: GSE30500, GEO3781644 and GEO6499593. The mouse reference genome for RNA-seq analysis is GRCm38 (https://cloud.biohpc.swmed.edu/index.php/s/grcm38_tran/download). Source data are provided with this study. All other data supporting the findings of this study are available from the corresponding author on reasonable request.

# Field-specific reporting

Please select the one below that is the best fit for your research. If you are not sure, read the appropriate sections before making your selection.

☒ Life sciences  ☐ Behavioural & social sciences  ☐ Ecological, evolutionary & environmental sciences

For a reference copy of the document with all sections, see nature.com/documents/nr-reporting-summary-flat.pdf

# Life sciences study design

All studies must disclose on these points even when the disclosure is negative.

| | |
|---|---|
| Sample size | No statistical methods were used to predetermine sample size. For each experiment, sample size was chosen based on the common methods in literatures and our previous experiences. |
| Data exclusions | No data were excluded from the analysis. |
| Replication | The majority of experiments was independently repeated at least three times with successful replication, with the exception of single cell RNA-sequencing which was performed once. The number of independent experiments was provided in the figures legends or in the method. |
| Randomization | No randomization was used in this study. Randomization is not applicable to cell line studies. Animal studies were observational studies, identifying cell populations in its endogenous state without any external manipulations. Positive controls were taken from the same mice. in most cases, brain NSCs from the same mice served as control for assessing pNSCs from the lung, tail, or other organs. For negative control for the tamoxifen experiments, mice were age matched with the same genotype, except without tamoxifen injection. |
| Blinding | Blinding was not performed in the experiments as most experiments were conducted by the same few investigators. For experiments comprising imaging analysis, laser intensity and channel parameters were uniform across all groups in the same set of experiment. All experiments were repeated independently multiple times to validate the results. |

# Reporting for specific materials, systems and methods

We require information from authors about some types of materials, experimental systems and methods used in many studies. Here, indicate whether each material, system or method listed is relevant to your study. If you are not sure if a list item applies to your research, read the appropriate section before selecting a response.

### Materials & experimental systems

| n/a | Involved in the study |
|---|---|
| ☐ | ☒ Antibodies |
| ☐ | ☒ Eukaryotic cell lines |
| ☒ | ☐ Palaeontology and archaeology |
| ☐ | ☒ Animals and other organisms |
| ☒ | ☐ Human research participants |
| ☒ | ☐ Clinical data |
| ☒ | ☐ Dual use research of concern |

### Methods

| n/a | Involved in the study |
|---|---|
| ☒ | ☐ ChIP-seq |
| ☐ | ☒ Flow cytometry |
| ☒ | ☐ MRI-based neuroimaging |

## Antibodies

| | |
|---|---|
| Antibodies used | Primary antibody for immunocytochemistry experiments: mouse anti-Nestin (Millipore, MAB353C3, 1:200), rabbit anti-Sox2 (Cell Signaling technology, #23064, 1:1000), goat anti-Sox2 (Santa Cruz, sc-17320, 1:500), rabbit anti-Olig2 (Millipore, AB9610, 1:1000), goat anti-Sox1 (R&D, AF3369, 1:200), mouse anti-Tuj1 (Sigma, T8660, 1:1000), rabbit-anti Tuj1 (Biolegend, 802001, 1:1000), rabbit |

anti-MAP2 (Santa Cruz, SC-20172, 1:1000), rabbit anti-GFAP (Millipore, AB5804, 1:500), rabbit anti-S100B (Abcam, ab41548, 1:200), mouse anti-O4 (Millipore, MAB345, 1:100), rat anti-Myelin Basic Protein (MBP) (Abcam, ab7349, 1:500), rabbit anti-glutamate (Sigma, AB5018, 1:2000), rabbit anti-GABA (Millipore, ABN131, 1:500), rabbit anti-ChAT (Millipore, AB143, 1:500), mouse anti-vesicular glutamate transporter 1 (vGluT1) (Millipore, MAB5502, 1:100), rabbit anti-p75 (Sigma, AB1554, 1:500), and rabbit anti-Sox10 (Abcam, ab155279, 1:500).

Primary antibody for immunohistochemical experiments: rabbit anti-S100B (Abcam, ab41548, 1:200), chicken anti-glial fibrillary acidic protein (Millipore, AB5541, 1:1000 or 1:500), rabbit anti-Oligo2 (Millipore, AB9610, 1:400), and mouse anti-neuronal nuclei (NEUN) (Millipore, MAB377, 1:400), rat anti-Myelin Basic Protein (MBP) (Abcam, ab7349, 1:400), rabbit anti-GFAP (Thermo Fisher Scientific, RB087A1, 1:400), goat anti-tdTomato (SICGEN, AB8181-200, 1:400), chicken anti-GFP (AVES LABS, AB_2307313, 1:400), rabbit anti-RFP (Biomol, 600-401-379, 1:400), rabbit anti-Ki67 (Abcam, ab15580, 1:400) and mouse anti-Nestin (Millipore, MAB353C3, 1:200), Rabbit anti-Sox2 (Cell Signaling, #23064, 1:400), goat anti-Sox1 (R&D, AF3369, 1:400), mouse anti-Tuj1 (Sigma, T8660, 1:400), rabbit-anti Tuj1 (Biolegend, 802001, 1:400), rabbit-anti MPZ (Thermo Fisher Scientific, #PA5-37179, 1:400), rabbit anti-Synapsin 1 (Sigma, S193, 1:300), rabbit anti-GABA (Millipore, ABN131, 1:250), rabbit anti-ChAT (Millipore, AB143, 1:250), rabbit anti-TH (Santa Cruz, sc-14007, 1:250).

Alexa fluorophore-conjugated secondary antibodies (all from Invitrogen, 1:500) and DAPI/Hoechst 33342 (Invitrogen, 1:1000) were used to visualize primary antibodies and nuclei, respectively.

| Validation | |
|---|---|

All primary antibodies used in this study are commercially obtained and were validated by commercial suppliers and confirmed by specific labeling of target molecules or cell types. Secondary antibodies have been tested in our experimental conditions to rule out unspecific reactivity.

Primary antibodies were validated by the supplier, with the relevant data page listed below:

Primary antibody for immunocytochemistry experiments:

mouse anti-Nestin (Millipore, MAB353C3, 1:200):
https://www.merckmillipore.com/IE/en/product/Anti-Nestin-Antibody-clone-rat-401-Cy3-conjugate,MM_NF-MAB353C3?ReferrerURL=https%3A%2F%2Fwww.google.com%2F#anchor_COA
(see CoA)

rabbit anti-Sox2 (Cell Signaling technology, #23064, 1:1000):
https://www.cellsignal.com/products/primary-antibodies/sox2-d9b8n-rabbit-mab/23064?srsltid=AfmBOorrh3BXPX5KMBOTjupSVFNlUoYdPyEChLdmwIuK7G2ow9jDi3Ra

goat anti-Sox2 (Santa Cruz, sc-17320, 1:500):
https://www.scbt.com/p/sox-2-antibody-y-17?srsltid=AfmBOoopHDzPmqmchc6xYhfigkXM5lrnTIPov4YRG_UcRg687-oUbvEo

rabbit anti-Olig2 (Millipore, AB9610, 1:1000):
https://www.merckmillipore.com/IE/en/product/Anti-Olig-2-Antibody,MM_NF-AB9610?ReferrerURL=https%3A%2F%2Fwww.google.com%2F#anchor_COA
(see CoA)

goat anti-Sox1 (R&D, AF3369, 1:200):
https://www.rndsystems.com/cn/products/human-mouse-rat-sox1-antibody_af3369#product-datasheets

mouse anti-Tuj1 (Sigma, T8660, 1:1000):
https://www.sigmaaldrich.com/HK/en/product/sigma/t8660

rabbit-anti Tuj1 (Biolegend, 802001, 1:1000):
https://www.biolegend.com/fr-ch/products/purified-anti-tubulin-beta-3-tubb3-antibody-11579?GroupID=GROUP686

rabbit anti-MAP2 (Santa Cruz, SC-20172, 1:1000):
https://www.scbt.com/p/map-2-antibody-h-300?srsltid=AfmBOop_vL9PnH5FO8Q7gkSwBeOwYLbie58_6Rl8PSy5Gbq264GCvos8

rabbit anti-GFAP (Millipore, AB5804, 1:500):
https://www.merckmillipore.com/IE/en/product/Anti-Glial-Fibrillary-Acidic-Protein-GFAP-Antibody,MM_NF-AB5804?ReferrerURL=https%3A%2F%2Fwww.google.com%2F#anchor_COA
(see CoA)

rabbit anti-S100B (Abcam, ab41548, 1:200):
https://www.abcam.com/en-us/products/primary-antibodies/s100-beta-antibody-ab41548?srsltid=AfmBOoqBQ_MvCnHNkj0Y7hJp1ttJqQaU6vKYFR8hFQszG2gnRPaLiO5n#tab=images

mouse anti-O4 (Millipore, MAB345, 1:100):
https://www.emdmillipore.com/US/en/product/Anti-O4-Antibody-clone-81,MM_NF-MAB345#anchor_COA
(see CoA)

rat anti-Myelin Basic Protein (MBP) (Abeam, ab7349, 1:500):
https://www.abcam.com/en-us/products/primary-antibodies/myelin-basic-protein-antibody-12-ab7349?srsltid=AfmBOoqqYD_Lixbv-FPAYluB7ztPpJnRMCImiRuCm_MfKUIC-V4SvfHr#tab=images

rabbit anti-glutamate (Sigma, AB5018, 1:2000):
https://www.merckmillipore.com/IE/en/product/Anti-Glutamate-No-Glutaraldehyde-Antibody,MM_NF-AB5018?ReferrerURL=https%3A%2F%2Fwww.google.com%2F#anchor_COA
(see CoA)

rabbit anti-GABA (Millipore, ABN131, 1:500):
https://www.merckmillipore.com/IE/en/product/Anti-GABA-Antibody,MM_NF-ABN131?ReferrerURL=https%3A%2F%2Fwww.google.com%2F#anchor_COA

rabbit anti-ChAT (Millipore, AB143, 1:500):
https://www.merckmillipore.com/IE/en/product/Anti-Choline-Acetyltransferase-ChAT-Antibody,MM_NF-AB143?ReferrerURL=https%3A%2F%2Fwww.google.com%2F#anchor_COA
(see CoA)

mouse anti-vesicular glutamate transporter 1 (vGluTl) (Millipore, MAB5502, 1:100):
https://www.merckmillipore.com/IE/en/product/Anti-Vesicular-Glutamate-Transporter-1-Antibody,MM_NF-MAB5502?ReferrerURL=https%3A%2F%2Fwww.google.com%2F#anchor_COA
(see CoA)

rabbit anti-p75 (Sigma, AB1554, 1:500):
https://www.merckmillipore.com/IE/en/product/Anti-Nerve-Growth-Factor-Receptor-Antibody-p75,MM_NF-AB1554?ReferrerURL=https%3A%2F%2Fwww.google.com%2F#anchor_COA
(see CoA)

rabbit anti-Sox10 (Abcam, ab155279, 1:500):

https://www.abcam.com/en-us/products/primary-antibodies/sox10-antibody-epr4007-ab155279?
srsltid=AfmBOorLqqbXSOx4lsDqPsIaLvBHpVThBIugp7os60EhzmnhAIUbT_-6#tab=images

Primary antibody for immunohistochemical experiments:
rabbit anti-S100B (Abeam, ab41548, 1:200):
https://www.abcam.com/en-us/products/primary-antibodies/s100-beta-antibody-ab41548?
srsltid=AfmBOoqBQ_MvCnHNkj0Y7hJp1ttJqQaU6vKYFR8hFQszG2gnRPaLiO5n#tab=images
chicken anti-glial fibrillary acidic protein (Millipore, AB5541, 1:1000 or 1:500):
https://www.merckmillipore.com/IE/en/product/Anti-Glial-Fibrillary-Acidic-Protein-Antibody,MM_NF-AB5541?ReferrerURL=https%
3A%2F%2Fwww.google.com%2F#anchor_COA
(see CoA)
rabbit anti-Oligo2 (Millipore, AB9610, 1:400):
https://www.merckmillipore.com/IE/en/product/Anti-Olig-2-Antibody,MM_NF-AB9610?ReferrerURL=https%3A%2F%
2Fwww.google.com%2F#anchor_COA
(see CoA)
mouse anti-neuronal nuclei (NEUN) (Millipore, MAB377, 1:400):
https://www.merckmillipore.com/IE/en/product/Anti-NeuN-Antibody-clone-A60,MM_NF-MAB377?ReferrerURL=https%3A%2F%
2Fwww.google.com%2F#anchor_COA
(see CoA)
rat anti-Myelin Basic Protein (MBP) (Abcam, ab7349, 1:400):
https://www.abcam.com/en-us/products/primary-antibodies/myelin-basic-protein-antibody-12-ab7349?srsltid=AfmBOoqqYD_Lixbv-
FPAYluB7ztPpJnRMCImiRuCm_MfKUIC-V4SvfHr#tab=images
rabbit anti-GFAP (Thermo Fisher Scientific, RB087A, 1:400):
https://www.fishersci.com/shop/products/gfap-glial-fibrillary-acidic-protein-rabbit-polyclonal-antibody-epredia/RB087A
goat anti-tdTomato (SICGEN, AB8181-200, 1:400):
https://store.sicgen.pt/catalog/product/AB8181
chicken anti-GFP (AVES LABS, AB_2307313, 1:400):
https://www.antibodiesinc.com/products/anti-green-fluorescent-protein-antibody-gfp
rabbit anti-RFP (Biomol, 600-401-379, 1:400):
https://www.biomol.com/products/antibodies/primary-antibodies/epitope-tag/anti-red-fluorescent-protein-rfp-600-401-379
rabbit anti-Ki67 (Abcam, ab15580, 1:400):
https://www.abcam.com/en-us/products/primary-antibodies/ki67-antibody-ab15580?
srsltid=AfmBOooWUA2BkT7tFHYgXg24rGmeCDt8F6Ez9ZL0EjzgipQgi6UiZ2ym#tab=images
mouse anti-Nestin (Millipore, MAB353C3, 1:200):
https://www.merckmillipore.com/IE/en/product/Anti-Nestin-Antibody-clone-rat-401-Cy3-conjugate,MM_NF-MAB353C3?
ReferrerURL=https%3A%2F%2Fwww.google.com%2F#anchor_COA
Rabbit anti-Sox2 (Cell Signaling, #23064, 1:400):
https://www.cellsignal.com/products/primary-antibodies/sox2-d9b8n-rabbit-mab/23064?
srsltid=AfmBOorrh3BXPX5KMBOTjupSVFNlUoYdPyEChLdmwIuK7G2ow9jDi3Ra
goat anti-Sox1 (R&D, AF3369, 1:400):
https://www.rndsystems.com/cn/products/human-mouse-rat-sox1-antibody_af3369#product-datasheets
mouse anti-Tuj1 (Sigma, T8660, 1:400):
https://www.sigmaaldrich.com/HK/en/product/sigma/t8660
rabbit-anti Tuj1 (Biolegend, 802001, 1:400):
https://www.biolegend.com/fr-ch/products/purified-anti-tubulin-beta-3-tubb3-antibody-11579?GroupID=GROUP686
rabbit-anti MPZ (Thermo Fisher Scientific, #PA5-37179, 1:400):
https://www.thermofisher.com/antibody/product/MPZ-Antibody-Polyclonal/PA5-37179
 rabbit anti-Synapsin 1 (Sigma S193, 1:300):
https://www.sigmaaldrich.com/HK/en/product/sigma/s193#product-documentation
rabbit anti-GABA (Millipore, ABN131, 1:250):
https://www.merckmillipore.com/IE/en/product/Anti-GABA-Antibody,MM_NF-ABN131?ReferrerURL=https%3A%2F%
2Fwww.google.com%2F#anchor_COA
(see CoA)
rabbit anti-ChAT (Millipore, AB143, 1:250):
https://www.merckmillipore.com/IE/en/product/Anti-Choline-Acetyltransferase-ChAT-Antibody,MM_NF-AB143?ReferrerURL=https%
3A%2F%2Fwww.google.com%2F#anchor_COA
rabbit anti-TH (Santa Cruz, sc-14007, 1:250):
https://www.scbt.com/p/th-antibody-h-196?srsltid=AfmBOop6h8X-31yUILg0ocXkxZHuHZo50MM2vwp1gCk-euzHyWl80tx4

# Eukaryotic cell lines

Policy information about cell lines

| Cell line source(s) | Mouse embryonic limb NSCs, adult lung NSCs, brain NSCs from WT mice were derived in house. Mouse adult lung, postnatal lung, tail, DRG peripheral NSCs, brain NSCs from Nestin-GFP, Sox2-GFP, Wnt1-Cre-mTmG, Sox1-GFP, Sox1-Cre-mTmG, Sox1-CreERT2-EYFP mice were derived in house. All the cell lines were derived from mice of both sex randomly. HEK293 cells were obtained from American Type Culture Collection (ATCC). |
|---|---|
| Authentication | HEK293 cells were authenticated by STR profiling from the providers. Other cell lines were authenticated in house using various methods including RT-qPCR, immunofluorescence, genotyping, karyotyping and bulk RNA-seq. |

| Mycoplasma contamination | HEK293 cells were tested as negative. Other cell lines were not tested for mycoplasma contamination. |
|---|---|
| Commonly misidentified lines (See ICLAC register) | No commonly misidentified cell lines were used. |

# Animals and other organisms

Policy information about studies involving animals; ARRIVE guidelines recommended for reporting animal research

| Laboratory animals | All mice used were bred and housed at the mouse facility Max Planck Institute (MPI) in Muenster and the Centre for Comparative Medicine Research (CCMR) at The University of Hong Kong (HKU), and animal handling was in accordance with MPI or HKU CCMR animal protection guidelines. The protocols for animal handling and maintenance for this study were approved by the Landesamt für Natur, Umwelt und Verbraucherschutz Nordrhein-Westfalen under the supervision of a certified veterinarian in charge of the MPI animal facility (protocols: Az 81-02.05.50.17.014, Az 84-02.04.2016.A525, Az 81-02.04.2017.A376 and Az 84-02.05.2016.A494) and by the Government of the Hong Kong Special Administrative Region Department of Health (protocol: (23-208) in DH/HT&A/8/2/3 Pt.55). C57BL/6, CD1, B6C3F1 mice were bred in house. The background of WT mice used in this study was CD1 mice or mixtures between CD1 and C3H. Nestin-GFP mouse (Stock No: 029671), Wnt1-Cre mouse (Stock No: 007807), R26-mT/mG mouse (Stock No: 007676) were from the Jackson Laboratory and bred in house with WT mice. Sox1-GFP mice were kindly provided by Professor Stavros Malas (University of Nicosia Medical School). R26-RFP mice were kindly provided by Professor Ralf Admas (Max Placnk Institute for molecular biomedicine). Sox2-GFP mice were generated in house. Sox1-Cre sperm were got from RIKEN BRC and Sox1-Cre mice were generated in house. Sox1-CreERT2-R26-EYFP mice were kindly provided by Professor Vasso Episkopou and Professor Robin Lovell-Badge. All transgenic mice were bred with WT mice in house for experiments. Adult WT or transgenic mice with age around 8-12 weeks were used for breeding. Adult WT or transgenic mice with age around 4-6 weeks were used for experiments. Postnatal mice with age day 1-5 were used for experiments. Embryonic limb cells were derived from embryos at embryonic day 13.5. For the teratoma assay, SCID mice (2-3 months old) were used. For the in vivo transplantation experiments, NOD. Cg-Prkdcscid Il2rgtm1Wjl/SzJ mice (8 weeks) or C57BL/6 (postnatal day 1) mice were used. Mice were maintained in the animal facility with a controlled temperature of 22 °C, 40–60% humidity, a 14:10 h light:dark photoperiod, and free access to water and food. |
|---|---|
| Wild animals | This study did not involve wild animals. |
| Field-collected samples | This study did not involve field-collected samples. |
| Ethics oversight | Animal experiments and husbandry were performed according to the German Animal Welfare guidelines and approved by the Landesamt für Natur, Umwelt und Verbraucherschutz Nordrhein-Westfalen (State Agency for Nature, Environment and Consumer Protection of North Rhine-Westphalia). Animal experiments and husbandry were approved by the Government of the Hong Kong Special Administrative Region Department of Health and performed according to HKU CCMR regulations. |

Note that full information on the approval of the study protocol must also be provided in the manuscript.

# Flow Cytometry

## Plots

Confirm that:

☒ The axis labels state the marker and fluorochrome used (e.g. CD4-FITC).

☒ The axis scales are clearly visible. Include numbers along axes only for bottom left plot of group (a 'group' is an analysis of identical markers).

☒ All plots are contour plots with outliers or pseudocolor plots.

☒ A numerical value for number of cells or percentage (with statistics) is provided.

## Methodology

| Sample preparation | Adult or postnatal Nestin-GFP mice, Wnt1-Cre-mTmG mice, Sox2-GFP mice, Sox1-GFP mice, Sox1-Cre-mTmG mice, Sox1-CreERT2-R26-EYFP mice were sacrificed, lung, brain, tail or DRG tissues were cut and minced using a pair of scissors, and incubated with 0.25% trypsin (Invitrogen) at 37°C for 10 min. After trypsinization, 3 times volume amount of MEF medium was added, and the entire suspension was pipetted up and down to dissociate the tissues. The dissociated tissues were passed through a 40-μm cell strainer (Beckon Dickinson, BD), centrifuged and resuspended in FACS medium (D-PBS without calcium and magnesium (Sigma) supplemented with 0.3% BSA (Sigma)). |
|---|---|
| Instrument | Flow cytometry was performed using either FACS Canto (BD Biosciences) or FACS Aria II (BD Biosciences). |
| Software | The data were analyzed using either eBD FACSDiva 8.01 (BD Biosciences) or FlowJo software 10.6.1 (Tree Star, Inc.). |
| Cell population abundance | After the sorting the cell population identity and abundance was verified by fluorescent microscope. |
| Gating strategy | Single viable cells were first selected based on forward scatter area/side scatter area and forward scatter width/forward scatter area gating to select for live cells and then sorted for GFP. For GFP+ cell sorting, brain GFP+ cells was used as positive control to select strong GFP+ cells. |

☒ Tick this box to confirm that a figure exemplifying the gating strategy is provided in the Supplementary Information.

