## [Peer Review File · Nature Cell Biology]

Multipotent neural stem cells originating from neuroepithelium exist outside of the mouse central nervous system

Corresponding Author: Dr Hans SCHÖLER

Version 0:

Decision Letter:

Dear Hans,

Your manuscript, "Multipotent neural stem cells with unique molecular feature exist outside of the mouse central nervous system", has now been seen by 3 referees, who are experts in neurogenesis, neuronal stem cells (referee 1); neural development (referee 2); and CNS development, scRNAseq (referee 3), and whose comments are pasted below. In light of their advice, we regret that we cannot offer to publish the study in Nature Cell Biology.

As you will see, although the reviewers find this work interesting, they raise serious concerns that question the strength of the data and of the novel conclusions that can be drawn at this stage.

We would be open to the possibility of considering a revised manuscript that would fully address the referee concerns (as an appeal). However, any decision to re-review such a revised study would depend on the strength of the revisions and the published literature at the time of resubmission.

We are very sorry that we could not be more positive on this occasion, but we thank you for the opportunity to consider this work.

With kind regards,

Christine.

Christine Weber, PhD
Senior Editor
Nature Cell Biology
E-mail: christine.weber@nature.com
Phone: +44 (0)207 843 4924

Reviewers' comments:

Reviewer #1 (Remarks to the Author):

In this manuscript the authors propose the isolation of CNS NSCs from embryonic limb or adult lung tissue. They base this on the expression of Nestin as shown by in vivo fate mapping, which comprise all the neurosphere-forming cells. However, Nestin has been shown to be expressed also in other cell types in the lung (e.g. endothelial cells, Bhagwani et al., 2019) and the authors did not show self-renewal and multipotent differentiation for these sorted cells. The authors further demonstrate that neurospheres emerging from adult lung tissue (without sorting for nestin-Cre/GFP) can self-renew for many passages and differentiate into neurons, GFAP+ and S100b+ cells and MBP/Olig2+ cells in vitro as well as after transplantation into the adult brain in vivo. However, they do not control for cell fusion nor show how frequent these respective cell types are in vivo. Finally, the authors present scRNA-seq data from adult lung tissue and identify an interesting cluster expressing some genes known to be also expressed in neural development and oligodendrocytes. However, several of these genes are also expressed in non-myelinating Schwann cells (e.g. Gfra3, Thai et al., 2019). Indeed, the manuscript lacks rigorous comparison to neural crest (stem) cells and Schwann cells, e.g. in Figure 1F, Figure 2-4. While the cells isolated and amplified in vitro in this manuscript may represent an interesting population of stem cells, the

claim that these are CNS NSCs and derive from neuroectoderm is rather premature and not yet fully supported by the data.

Necessary data to support conclusions:

- 1) Include RNA-seq data from neural crest-derived stem and progenitor cells in Figure 1F.
- 2) Analyse several exclusive Schwann cell markers in Figure 2B.
- 3) Control for cell fusion by transplanting cells expressing Cre into a reporter mouse line.
- 4) Provide quantification of the proportion of NeuN+, S100b and MBP+ cells amongst all mCherry+ cells from at least 3 animals.
- 5) Provide 2) and 4) from neurosphere cells derived from Nestin-Cre fate mapped cells.
- 6) Ideally use a more specific marker to fate-map neuro-ectoderm-derived cells as nestin is also expressed in many cell types.
- 7) Figure 4: compare expression of markers shown in e and g to neural crest derived cells and examine Schwann cell markers in this population.

Reviewer #2 (Remarks to the Author):

In this manuscript, Han et al. reported derivation of cells with CNS neural stem cell (NSC) properties in vitro from different peripheral tissues, including mouse embryonic limb, postnatal tail tip, and adult lung. They presented evidence that upon culturing these cells with growth factors can be passage for over 50 times, exhibit transcriptomes similar to CNS NSCs, and can give rise to three lineages in the CNS both in vitro and after transplantation. They further showed that these NSCs are likely derived from Nestin-expressing cells. They concluded that multipotent neural stem cells with unique molecular features exist outside of the mouse central nervous system. In addition to continuous neurogenesis from endogenous quiescent neural stem cells in the discrete regions of the adult mouse brain, continuous neurogenesis also occurs in the peripheral nervous system, such as olfactory epithelium and gut. Therefore, neural stem cells are not limited to CNS. Previous studies have already shown that neural progenitor cells derived from the peripheral tissue, once cultured as neurospheres for multiple passages, can give rise to CNS cell types in vitro and after transplantation (e.g. PMID: 21525278). The current study does not provide any evidence on the properties of putative neural stem cells in the peripheral tissue in vivo. Are these putative stem cells quiescent normally? Can these putative stem cells be activated (e.g. after injury) and exhibit self-renewal and differentiation in vivo? How do their transcriptomes compare to quiescent neural stem cells in the brain and neural crest stem cells in the PNS? The authors have the tool of nestin-cre to perform lineage-tracing and other analyses of these cells under in vivo conditions in mice. Without any insight into the endogenous properties of these putative stem cells in vivo, the observed effect could be due to in vitro culture reprogramming and there is little novelty from the current study for publication in Nature Cell Biology.

Specific comments:

In addition to analysis of properties of these putative neural stem cells in vivo as listed above, the authors need to show whether these cells isolated from in vivo (e.g. by Nestin-GFP) can give rise to CNS cell types in vitro and after transplantation in vivo without expansion with growth factors EGF and FGF-2 to reveal their intrinsic properties. Can these cell also differentiate into neural crest lineages?

Single-cell RNA-seq of Nestin-GFP cells showed expression of some markers of quiescent NSCs and neural crest stem cells, but not others. There is a need of a systematic comparison with neural crest stem cells and brain quiescent NSCs at the single cell levels. Lack of expression of some markers is not a good indicator, given the limited depth of single-cell RNA-seq.

The authors should compare the transcriptome of acutely isolated nestin-GFP cells (via single-cell RNA-seq) with those after culturing in the presence of growth factors and passaging to see whether there are any substantial changes in gene expression, suggesting reprogramming.

The authors used nestin-cre fate mapping to argue that these putative stem cells have a CNS origin. However, the nestin-Cre tool is not specific enough, as evident in their single-cell RNA-seq analysis of nestin-GFP cells. Therefore, the origin of these putative stem cells is still a question.

The authors showed that transplanting IdNSCs into adult mouse cortex leads to generation of cells of neuronal and glial lineages. However, this result is not quantified. Is the propensity to yield terminally differentiated neuron/glial cell comparable from IdNSCs comparable to "control" NSCs?

Other comments:

The authors need to explain control NSC in more detail, instead of citing a previous paper.

The authors showed electrophysiological evidence of neuronal excitability, but the action potential is rather immature and no repetitive firing. The authors did not examine evidence of morphological and functional synapse formation of neural progeny.

The authors did not study netrin-GFP sorted cells from the limbs.

Reviewer #3 (Remarks to the Author):

Han, Schöler and co-workers describe isolation of a specific cell type, which shares some features with neuronal stem cells (NSCs). These cells were isolated from the embryonic limb, postnatal tail tip, and adult lung tissue after treatment in low pH before placement in NSC culture medium. Slowly proliferating cells emerged and in differentiating culture conditions, and after transplantation into mouse cortex, cells formed glia and neurons. The authors suggest that these "low-pH-derived neural stem cells" (or IdNSCs), are multipotent neural stem cells derived from embryonic neuroepithelial cells. It is claimed that the cells do not originate from neural crest (NC) cells like other peripheral neural cells including peripheral neurons, glia and Schwann cells. As the authors point out, this is contrary to the dogma that neural stem cells exist outside of the central nervous system. However, as pointed out in the specific comments below, the claim that NSCs exist in peripheral tissues remains unproven and extensive additional experiments would be required to make such a far-reaching claim. Also, it would be essential to demonstrate the existence of such cells in their natural environment and preferably understand if – and under what conditions – they can generate neurons and glia *in vivo*.

Specific comments:

1. It is not possible to conclude that a specific cell type has certain properties unless the studied cells are clonally derived from a single cell. It remains unclear if the authors have performed any such single cell clonal analyses in the characterization of IdNSCs. Indeed, different types of cells with proliferative potential are likely present in the analyzed peripheral tissues (some with neural properties; see below). Thus, cultures defined as IdNSCs could be a heterogeneous mixture of different types of stem-/progenitor cells with different potential to generate e.g. neurons or glia. This is also relevant when analyzing cells after transplantation. As presented, it remains unclear if a clonally derived stem cell from peripheral tissues has potential to form both glia and neurons.
2. Already from images in Fig. 1a-d and Ext Fig 1c,d it is concluded that the cultured cells have properties of NSCs. It is unclear why NSC-properties are presumed from these data alone. Separate channels should be shown in Fig. 1d.
3. In Fig 1, IdNSCs were compared to i) TF-induced NSCs from a previous study; ii) control adult brain NSCs, and iii) fibroblasts. However, the comparison to two types of NSCs and fibroblasts by array hybridization provides rather limited information on cell type identity. This experiment only shows that IdNSCs are more similar to NSCs as compared to fibroblasts, and it remains unclear why the cells are referred to as NSCs based on these data (see further comments below).
4. There is no quantification presented in relation to the data in Fig 2a-d. What is the relative proportion of different cell types after differentiation? What proportion of all plated cells differentiates into differentiated neural cell types (neurons, astrocytes, oligos)?
5. It is stated that allNSCs, after transplantation to cortex, "survived in the circuits of an *in vivo* neural network (Fig. 2g)". From the presented data it is not possible to conclude that cells are integrated into the cortical neural network.
6. The data in Fig. 2h is very difficult to evaluate since it shows high-magnifications of individual cells without any quantification or overview that indicates how common these cells are in the recipient cortex. How many of the total transplanted mCherry+ cells form differentiated neural cells *in vivo*? What is the fate of other non-neural mCherry+ cells? Are there evidence for proliferation? Also, how were cells labelled with mCherry (appears to be missing in Methods).
7. Figure 3 is absolutely central to the conclusions of this paper and evidence needs to be rock solid for the far-reaching claim that is made in this study. The experiments are designed to show that pNSCs exist in peripheral adult tissues and that they are derived from embryonic neuroepithelial cells. The data in Fig. 3a-d shows that "pNSCs" can be derived without low-pH treatment, but unfortunately they do not provide evidence showing that such cells exist *in vivo*. Importantly, transgenic reporters based on Nestin-regulatory sequences can be leaky and may also be affected by dissection/dissociation procedures. Thus, it remains possible that they are programmed into proliferating neural cells as a consequence of dissociation/plating, or they may be derived from peripheral NC-derived glial cells expressing the transgenic reporter (but not high levels of endogenous Nestin). Importantly, only low levels of leaky expression would be sufficient to give rise to GFP+ cells. In addition, concluding that IdNSCs are derived from endogenous pNSCs based on the data in Ext Fig 3e,f, is based on negative data (IdNSCs cannot be derived from GFP- cells). Also this is not very convincing.
8. There are additional problems with the conclusions from the lineage tracing experiment in Fig. 3e-g. The authors have already shown that Nes-GFP+ cells can be derived from the adult lungs. Thus, using Nes-Cre to show that cells originate from the neural epithelium is not possible since Nes-Cre expression (just like Nes-GFP in the adult lung) would lead to recombination of the reporter. Showing the origin of these cells would require a much more extensive lineage tracing strategy with inclusion of appropriate controls. A strategy must demonstrate that recombination can only occur in the embryonic neuroepithelium e.g. by using a suitable CreER mouse line (e.g. NE-NesCreER), allowing for tamoxifen-induced recombination during embryogenesis. Using viral transfection of neuroepithelium *in utero* or intersectional genetic approaches to make the targeting more specific could also be useful. It would also be essential to follow labelled cells during development and careful controls need to assure that NC cells (or other non-neuroepithelial cells) are not lineage marked. Showing the absence of just one marker (HNK1) in *in vitro* cultured cells is far from sufficient for the conclusion that pNSC of non-NC origin exists in peripheral adult tissues (Fig 3g).

9. The scRNA-seq data in Fig. 4 identifies clusters derived from GFP sorted cells from Nes-GFP mouse brain and lung. Several clusters in Fig 4a,c correspond to non-neural cell types that do not normally express Nestin including EC, SMC, Lym, Epi etc. What is the explanation for this? It seems to indicate that the Nes-GFP is leaky, or that the sorting is rather non-selective, a finding that has implications also for the conclusions in Fig. 3 (see above). How is Nestin mRNA expressed in the various clusters including the small ppNSC cluster? Showing expression in the clusters or in e.g. violin plots would be useful.

10. Some NC markers were shown to be expressed in the ppNSC cluster and several of the "oligodendroglial" markers are normally expressed also in NC-derived Schwann cells and enteric glia (see e.g. mousebrain.org). In addition, Olig1 and Olig 2 were not expressed in this cluster. Does this not open up for the possibility that the peripheral "pNSCs" are of NC origin?

11. It would be more informative to plot all cells from both brain and lung together to see how ppNSCs cluster in relation to NSCs and other cell types found in the brain. In fact, markers found for the ppNSCs appear to match more closely to glia than NSCs (see above). For example, most markers that authors claim are for NSCs (lines 222-223) appear to be strongly expressed in enteric glia and other NC-derived cells according to mousebrain.org. In addition to mousebrain.org many other scRNA-seq data sets are now available and an integrated bioinformatic approach can go a long way to provide further information on the identity of ppNSCs.

**Although we cannot offer to publish your paper in Nature Cell Biology, the work may be appropriate for another journal in the Nature Research portfolio. If you wish to explore suitable journals and transfer your manuscript to a journal of your choice, please use our manuscript transfer portal. If you transfer to Nature-branded journals or to the Communications journals, you will not have to re-supply manuscript metadata and files. This link can only be used once and remains active until used.

All Nature Research journals are editorially independent, and the decision to consider your manuscript will be taken by their own editorial staff. For more information, please see our manuscript transfer FAQ page.

Note that any decision to opt in to In Review at the original journal is not sent to the receiving journal on transfer. You can opt in to In Review at receiving journals that support this service by choosing to modify your manuscript on transfer. In Review is available for primary research manuscript types only.

**For Nature Research general information and news for authors, see <http://npg.nature.com/authors>.

Version 1:

Decision Letter:

Dear Dr SCHÖLER,

Your manuscript "Multipotent neural stem cells originating from neuroepithelium exist outside of the mouse central nervous system", has now been seen by 2 of the original referees, whose comments are pasted below. We were unfortunately unable to obtain the report by referee #1 and we therefore asked referee #3 to cross-comment on your rebuttal to the original comments by referee #1 (see below the report provided by referee #3). In light of their advice, we regret that we cannot offer to publish the study in Nature Cell Biology.

While the referees acknowledge the additional experiments performed, they continue to raise concerns about the strength of the conclusions that can be drawn. We feel that these reservations are sufficiently important to preclude publication of this study in Nature Cell Biology.

We are very sorry that we could not be more positive on this occasion, but we thank you for the opportunity to consider this work.

With kind regards,
Stelios

Stylianos Lefkopoulos, PhD
He/him/his
Associate Editor
Nature Cell Biology
Springer Nature
Heidelberger Platz 3, 14197 Berlin, Germany

E-mail: stylianos.lefkopoulos@springernature.com

Twitter: @s_lefkopoulos

Reviewers' comments:

Reviewer #2 (Remarks to the Author):

In the current resubmission of the manuscript, Han et al., have performed additional experiments to support their conclusion that neuroepithelium derived multipotent stem cells can be enriched from various parts of the murine peripheral (non-CNS) body. The revised manuscript now provides evidence for the following findings. First, low-pH derived NSCs (IdNSCs) can be obtained from embryonic limb/ adult lung with NSC-like properties. In vitro, they have the proliferative and differentiation potential similar to brain derived NSCs. These cells are multipotent, survive transplantation without resulting in tumors, differentiating into neuron, astrocytes and oligodendrocytes at a degree similar to brain derived NSCs. Hierarchical clustering and microarray data presented do show the similarity between brain derived NSCs and IdNSCs. Second, from peripheral tissues (lung, limb, postnatal tail) the authors do isolate multipotent peripheral NSCs (pNSCs) which can be maintained in culture and have differentiation similar to brain derived NSCs. The in vivo transplantation of pNSCs, obtained without low pH treatment, expressing Cre, H2B-GFP into R26-loxp-stop-loxp-RFP does provide evidence to exclude cell fusion. Third, the pNSCs do not express NC markers in vitro and pNSCs isolated from postnatal lung from Sox1-GFP KI animals (and identified as Sox1-GFP+ cells) behave as control brain derived NSCs. Despite these new results, the concerns listed below need to be addressed.

Concerns

1. In Figure 7 the authors performed scRNA-seq of Sox1-GFP cells from the brain and the postnatal tail and argue that NSCs from both tissues overlap. This is a significant concern, not just for the current study but an inference which can be drawn is that cellular quiescence is similar irrespective of the niche and microenvironment, which is not the case- even for qNSCs from different brain regions. The authors attribute other cells populations to cytometry noise. For comparison between cells from the brain and the postnatal tail, these cells have to be excluded from Figure 7C. Their presence in the analysis, especially immune cells, will lead to a false sense of clustering (The "NSCs" do not really overlap either). Furthermore, immune cells in the tail would not be microglia; macrophages, at best. To succinctly show similarities, the authors need to show an UMAP just with the "NSCs". Next the authors need to perform far more detailed pair wise comparisons, along with hypergeometric tests, between the "NSC" populations to show greater than expected overlap (please present venn diagrams). It is not clear from the methods how identities were attributed to cell clusters, specially the cells from the tail. In the same figure, 7d panel is problematic. Sox2, even at the transcript level, is well published to be downregulated in brain neuroblasts. However, there is no observable difference between the brain clusters. This problem is far more exaggerated for Sox1. Given that the cells are obtained from Sox1-GFP knockin line, one would expect far more NSCs, rather than neuroblasts to be Sox1 positive.
2. The authors elude to a full-fledged neurogenic niche within the postnatal tail (with qNSCs, aNSCs and NBs) (Figure 7f). From the numbers, ~450 NBs (18%) were sequenced. While the authors show some very intriguing Sox1+Ki67+ cells within the tail, it is pertinent to perform staining for other NB markers (DCX, Calretinin, PSA-NCAM). Given the magnitude of the claim, a sufficient explanation has to be provided regarding the final fate of these pNSC derived NBs. It is recommended to follow this niche into adulthood.
3. An alternate explanation for pNSCs is de/trans-differentiation (a case can be made for either phenomena). The authors clearly aim to address this in Figure 8. Using constitutive Sox1 driven Cre does explain the observed GFP (mTmG) detection in the brain during embryonic development. However, an inducible version of Sox1-Cre had to be employed to rule out a late onset of Sox1 expression in the peripheral tissue, explaining GFP+/Tomato- cells in peripheral tissue, which negates a neuroepithelial origin. It is advised to do a far more stringent time course, if the current line is used.
4. Quiescence is not a singular state, but graded. The authors interpret that low recovery of colonies from pNSCs is indicative of their quiescence. While the authors perform low-pH treatment from Sox1-GFP+/- cells, it would be pertinent to check if the neurosphere/cluster formation from GFP+ fractions from various tissues increases with the treatment, and compare this treating control brain NSCs with low-pH (which was not done in any experiments). This experiment would also try to shed light on why so few pNSCs, at least from the postnatal tail, form colonies.
5. The observation that DRG houses pNSCs is intriguing. The authors do not show the FACS plot of DRG isolated cells for Tomato/GFP levels. A closer look at the image in Figure 5e shows some GFP+ staining. What is the proportion of Tomato+/GFP- cells? The expectation is that GFP+ cells would outnumber the GFP- cells. What fraction of the Tomato+/GFP- cells form clusters? If GFP+ cells can be recovered from stringent Tomato+ gating, does this mean that a minority of the population cells start to express Wnt1 (and p75)? Can low-pH treatment of the GFP+ cells from DRG induce p75- "pNSC-like" cultures? If they don't, that could argue against trans/de-differentiation. Can you recover pNSCs from the Sox1-GFP mice's DRGs?
6. In many experiments, the authors report that that sorted GFP+ cells were cultured on monolayers in NSC medium and clusters were picked. How does one isolate and pick adhered monolayer cells from the surrounding cells? (Specifically refer line 191-192; methods line 611-612) Please explain this more clearly in the methods.
7. The supplementary table with the raw results is useful. It is advised that the authors provide the raw numbers from the transplantation experiments employed in the manuscript.
8. Please provide references for p75 (line 255-256)
9. In the experiment detailed in lines 123-127, the authors checked the methylation status for 2 genes- Colla1 and Nestin. To say that IdNSCs and brain derived NSCs are similar at the epigenetic level is a huge overreach.

Reviewer #3 (Remarks to the Author):

The revised manuscript by Han et al. has been extensively revised and improved. Most of my previous technical concerns have been addressed, and the conclusion that multipotent cells with NSC-properties can be isolated from the lung and tail is supported by the data.

An important critique in my previous review was that the authors did not accurately distinguish the tail and lung NSCs from neural crest derived stem cells (NCSCs). One of the approaches was to use a Wnt-Cre transgenic mouse line to mark cells of neural crest origin. However, the data is based on lack of Wnt-Cre-induced floxing in neuroepithelial-derived NSCs, but not in neural crest-derived cells. However, it is likely that not 100% of NC-derived cells are accurately floxed and that a small proportion NC-derived cells can escape floxing and give rise to the misleading data. This is a common weakness in many lineage-tracing studies, e.g. emphasized in many of the strongly criticized astrocyte-to-neuron reprogramming studies that have been published in recent years. It is important that the authors explain the potential problems in interpretation of their lineage-tracing data. However, the conclusion that cultured lung and tail stem cells have properties distinct of NCSCs is now convincingly supported, not least by the inclusion of more solid scRNAseq data.

Also, the new scRNAseq data derived from Sox1-GFP+ cells from the tail and compared to Sox1-GFP+ cells from the embryonic brain is an important new addition that strengthens the manuscript.

The authors also conclude that Sox1+ cells in the embryonic epithelium gave rise to the peripheral NSCs cultured in vitro and the cells with NSC-properties observed in tissue sections. However, as eluded to above, lineage tracing approaches are often leading to incorrect conclusions. The data in Fig. 8e-g can be explained either by Cre-expression in peripheral tissues leading to Cre-induced recombination outside of the neuroepithelium. Such expression could be a reflection of "real" Sox1 expression or it could be a result of weak expression of the transgenic KI-construct leading to sufficient Cre for recombination even in cells where Sox1 is not detected. Also, since this is not a tamoxifen-controlled Cre, expression can lead to recombination at post-embryonic stages, e.g in the adult tissues. Thus, the evidence supporting the existence of neuroepithelial-derived quiescent NSCs in the periphery is rather weak. More importantly, what is the significance of such cells and are the neurons derived from them functional and of physiological importance?

Comments on authors responses to referee 1:

Comments from referee 1 resembles many of my own concerns (referee 3). The authors have responded to all comments from referee 1 and they have also added new data that strengthens the claims.

Some comments:

- Point 2. The authors refer to the Wnt1-Cre tracing experiment which provides support for the conclusion that the analyzed cells were not derived from NCSCs. It should be noted, and pointed out in the ms, that this is a conclusion that is based on negative data. Thus, Wnt1-Cre labels NCCs and the lack of labeling is used to conclude that that cells are non-NCCs (see comment also to referee 3).
- Point 5. Another lineage tracing approach used Sox1-Cre to label cells originating from the neuroepithelium. See comment from referee 3 concerning this approach. It is important to consider the many problematic studies using reprogramming approaches to make far-reaching claims that astrocytes and other cell types can be reprogrammed into therapeutically desired cell types in vivo. Many of these papers have not been possible to repeat and major problems were due to lineage tracing approaches that are not solid. In this manuscript, additional supporting data (e.g. scRNA-seq) provides independent strength to the conclusions, but cautionary comments in the ms on the tracing strategies would be important.
- The problem pointed out above is partly addressed by the scRNA-seq data from Sox1-Cre labeled cells. These data provide a strong addition addressing problems with the original conclusions.
- I believe that most other specific points raised by Referee 1 have been addressed by the addition of new experimental data.

****Although we cannot publish your paper, it may be appropriate for another journal in the Nature Portfolio. If you wish to explore the journals and transfer your manuscript please use our manuscript transfer portal. You will not have to re-supply manuscript metadata and files, but please note that this link can only be used once and remains active until used. For more information, please see our [manuscript transfer FAQ](http://www.nature.com/authors/author_resources/transfer_manuscripts.html?WT.mc_id=EMI_NPG_1511_AUTHORTRANSF&WT.ec_id=AUTHOR) page.**

Note that any decision to opt in to In Review at the original journal is not sent to the receiving journal on transfer. You can opt in to [In Review](https://www.nature.com/nature-research/for-authors/in-review) at receiving journals that support this service by choosing to modify your manuscript on transfer. In Review is available for primary research manuscript types only.

**For Nature Portfolio general information and news for authors, see <http://npg.nature.com/authors>.

Version 2:

Decision Letter:

*Please delete the link to your author homepage if you wish to forward this email to co-authors.

Dear Hans,

As previously communicated via e-mail, we had initially sent the manuscript back to the original referees #2 and #3, but referee #3 did not manage to submit their report this time. We therefore had to seek advice from a new expert (referee #4), who was asked to assess your response to the comments by the original referee #3. Both referees #2 and referee #4 responded, as you know, and we have now assessed their reports, but also your response to them, which you have recently shared with me via e-mail (thank you very much for that). The referees' comments (attached below) indicate that the work is of interest, and has improved upon revision, but still raise some points, which we believe you should address the way you have proposed in the point-by-point response that you shared with me via e-mail, before we can consider publication of the study.

We are therefore inviting you to revise your manuscript per the plan explained and new analyses performed and explained in your point-by-point response and resubmit your manuscript within 4 weeks. This is also an opportunity for you to revise your point-by-point response to the reviewers (should you find it necessary) before you resubmit your files.

Please pay close attention to our guidelines on statistical and methodological reporting (listed below) as failure to do so may delay the reconsideration of the revised manuscript. In particular please provide:

We therefore invite you to take these points into account when revising the manuscript. In addition, when preparing the revision please:

- ensure that it conforms to our format instructions and publication policies (see below and <https://www.nature.com/nature/for-authors>).

- provide a point-by-point rebuttal to the full referee reports verbatim, as provided at the end of this letter.

- provide the completed Reporting Summary (found here <https://www.nature.com/documents/nr-reporting-summary.pdf>). This is essential for reconsideration of the manuscript and will be available to editors and referees in the event of peer review. For more information see <http://www.nature.com/authors/policies/availability.html> or contact me.

Nature Cell Biology is committed to improving transparency in authorship. As part of our efforts in this direction, we are now requesting that all authors identified as 'corresponding author' on published papers create and link their Open Researcher and Contributor Identifier (ORCID) with their account on the Manuscript Tracking System (MTS), prior to acceptance. ORCID helps the scientific community achieve unambiguous attribution of all scholarly contributions. You can create and link your ORCID from the home page of the MTS by clicking on 'Modify my Springer Nature account'. For more information please visit please visit www.springernature.com/orcid.

This journal strongly supports public availability of data. Please place the data used in your paper into a public data repository, or alternatively, present the data as Supplementary Information. If data can only be shared on request, please explain why in your Data Availability Statement, and also in the correspondence with your editor. Please note that for some data types, deposition in a public repository is mandatory - more information on our data deposition policies and available repositories appears below.

Link Redacted

I am reminding you that we would like to receive the revision within four weeks. If submitted within this time period, reconsideration of the revised manuscript will not be affected by related studies published elsewhere, or accepted for publication in Nature Cell Biology in the meantime. We would be happy to consider a revision even after this timeframe, but in that case we will consider the published literature at the time of resubmission when assessing the file.

We hope that you will find our referees' comments, and editorial guidance helpful. Please do not hesitate to contact me if there is anything you would like to discuss.

Best wishes,

Stelios

Stylios Lefkopoulos, PhD
He/him/his
Senior Editor, Nature Cell Biology
Springer Nature
Heidelberger Platz 3, 14197 Berlin, Germany

E-mail: stylios.lefkopoulos@springernature.com
Twitter: [@s_lefkopoulos](https://twitter.com/s_lefkopoulos)
LinkedIn: [linkedin.com/in/stylios-lefkopoulos-81b007a0](https://www.linkedin.com/in/stylios-lefkopoulos-81b007a0)

Reviewers' Comments:

Reviewer #2 (Remarks to the Author):

In the revised manuscript the authors have performed extensive experiments to address the previous concerns. The presented work shows that Sox1-expressing cells exist outside the CNS with properties similar to brain derived NSCs- in terms of colony formation, differentiation potential and marker expression. These cells, resident in many tissues, at the very least, seem to retain Sox1 expression and seemingly be able to differentiate into neurons (at different developmental speeds). The authors do provide compelling evidence using a single Tamoxifen injection strategy that the "pNSCs" identified are most likely from the neural tube early in the brain development. While the manuscript is improved, there are several remaining some points to be addressed:

1. A overall caveat is that these pNSCs, although possess stem like properties, may represent biological "noise"- a few cells escaping canonical developmental steps, ending up in the periphery. These cells do have the remarkable ability to generate terminally differentiated cells, which express makers which might suggest a neuron-like function. These cells have similar expression profile to brain NSCs. But it has to be recognized that the brain NSCs are an aggregate of NSCs from both the SGZ and SVZ. Hence the similarity could be driven, in part from the fact these are stem-like cells. To overcome this concern, can the authors can use two approaches- i. compare the expression profiles of tail Sox1+ cells to bona fide stem cells of the resident/other tissues, and repeat the venn diagram. Furthermore, regress the cell cycle genes from the expression profiles and regenerate the correlation map; ii. Post the low pH treatment expose the Sox1+ cells to medium favoring the differentiation into cell types of different origin. If other cell types are generated, an alternate hypothesis emerges- Sox1+ pNSCs are more akin to resident stem cells with a large degree of differentiation potential. The authors might appreciate that this hypothesis is not ruled out, as all experiments, including the transplantation heavily, if not exclusively, drive the "pNSCs" to a neuronal fate.
2. What intriguing is that the venn diagram presented shows that the tail pNSCs have a very small proportion of exclusively expressed genes. Is the overlap purely cell cycle genes? What pathways do brain NSCs, specially the quiescent ones, express which are not "tuned on" in the pNSCs? Can the authors generate GO analyses of DEGs enriched in brain specific NSCs?
3. The authors show that neurons are generated, survive and express mature makers and the authors claim that they are functional. Please tone down the claim of functionality, as the authors do not show innervation or functional properties in

vivo.

4. Can the authors provide control images for Figure 8H?

5. Can the authors change the colour scheme in Figure 4d-f? It is very hard to separate red and magenta.

Reviewer #4 (Remarks to the Author):

In this manuscript, Han et al. have presented interesting results that the peripheral neural stem cells (pNSCs) exist which are found from the process of validating the stimulus-triggered acquisition of pluripotency (STAP) based on low-pH treatment of somatic cells. The authors provide the following evidence to prove the existence and the characteristics of pNSCs. First, the pNSCs can be obtained from the limbs and adult lung tissues by low-pH treatment which have the properties of CNS NSC. Second, the pNSCs can be directly isolated from tissue outside of CNS in NSC culture medium without low-pH treatment. Third, the evidence of distinct properties of pNSCs between neural crest cells (NCCs) and the non-NCCs derivation of pNSCs which are proved by lineage tracing method with multiple Cre mice strains are presented. Fourth, the results of scRNA-seq are also give the convincing evidence of the similarity between CNS NSCs and pNSCs.

Comments on authors responses to referee 3:

The most questions raised by referee 3 and referee 1 have been addressed by the latest version of manuscript, and the lineage tracing data with Sox1-CreERT2 is convincing. However, the lineage tracing by Wnt1-Cre is still the doubtful point. Can author provide any evidences to prove the NCCs and NECs are mutually exclusive? In other words, can any evidence prove that Cre+ cells in Wnt1-Cre mice are Sox1- cells at embryonic or postnatal stages? Nevertheless, I support the publication of the paper.

GUIDELINES FOR SUBMISSION OF NATURE CELL BIOLOGY ARTICLES

ARTICLE FORMAT

ABSTRACT – should not exceed 150 words and should be unreferenced. This paragraph is the most visible part of the paper and should briefly outline the background and rationale for the work, and accurately summarize the main results and conclusions. Key genes, proteins and organisms should be specified to ensure discoverability of the paper in online searches.

TEXT – the main text consists of the Introduction, Results, and Discussion sections and must not exceed 3500 words including the abstract. The Introduction should expand on the background relating to the work. The Results should be divided in subsections with subheadings, and should provide a concise and accurate description of the experimental findings. The Discussion should expand on the findings and their implications. All relevant primary literature should be cited, in particular when discussing the background and specific findings.

REFERENCES – are limited to a total of 70 in the main text and Methods combined,. They must be numbered sequentially as they appear in the main text, tables and figure legends and Methods and must follow the precise style of Nature Cell Biology references. References only cited in the Methods should be numbered consecutively following the last reference cited in the main text. References only associated with Supplementary Information (e.g. in supplementary legends) do not count toward the total reference limit and do not need to be cited in numerical continuity with references in the main text. Only published papers can be cited, and each publication cited should be included in the numbered reference list, which should include the manuscript titles. Footnotes are not permitted.

Methods should be written concisely, but should contain all elements necessary to allow interpretation and replication of the results. As a guideline, Methods sections typically do not exceed 3,000 words. The Methods should be divided into subsections listing reagents and techniques. When citing previous methods, accurate references should be provided and any alterations should be noted. Information must be provided about: antibody dilutions, company names, catalogue numbers and clone numbers for monoclonal antibodies; sequences of RNAi and cDNA probes/primers or company names and catalogue numbers if reagents are commercial; cell line names, sources and information on cell line identity and authentication. Animal studies and experiments involving human subjects must be reported in detail, identifying the committees approving the protocols. For studies involving human subjects/samples, a statement must be included confirming that informed consent was obtained. Statistical analyses and information on the reproducibility of experimental results should be provided in a section titled "Statistics and Reproducibility".

All Nature Cell Biology manuscripts submitted on or after March 21 2016, must include a Data availability statement as a separate section after Methods but before references, under the heading "Data Availability". For Springer Nature policies on data availability see <http://www.nature.com/authors/policies/availability.html>; for more information on this particular policy see <http://www.nature.com/authors/policies/data/data-availability-statements-data-citations.pdf>. The Data availability statement should include:

- Accession codes for primary datasets (generated during the study under consideration and designated as "primary accessions") and secondary datasets (published datasets reanalysed during the study under consideration, designated as "referenced accessions"). For primary accessions data should be made public to coincide with publication of the manuscript. A list of data types for which submission to community-endorsed public repositories is mandated (including sequence, structure, microarray, deep sequencing data) can be found here <http://www.nature.com/authors/policies/availability.html#data>.
- Unique identifiers (accession codes, DOIs or other unique persistent identifier) and hyperlinks for datasets deposited in an approved repository, but for which data deposition is not mandated (see here for details <http://www.nature.com/sdata/data-policies/repositories>).
- At a minimum, please include a statement confirming that all relevant data are available from the authors, and/or are included with the manuscript (e.g. as source data or supplementary information), listing which data are included (e.g. by figure panels and data types) and mentioning any restrictions on availability.
- If a dataset has a Digital Object Identifier (DOI) as its unique identifier, we strongly encourage including this in the Reference list and citing the dataset in the Methods.

We recommend that you upload the step-by-step protocols used in this manuscript to [protocols.io](http://www.protocols.io). More details can found at <https://www.protocols.io/help/publish-articles>.

DISPLAY ITEMS – main display items are limited to 6-8 main figures and/or main tables. For Supplementary Information see below.

FIGURES – Colour figure publication costs \$395 per colour figure. All panels of a multi-panel figure must be logically connected and arranged as they would appear in the final version. Unnecessary figures and figure panels should be avoided (e.g. data presented in small tables could be stated briefly in the text instead).

All imaging data should be accompanied by scale bars, which should be defined in the legend.

Cropped images of gels/blots are acceptable, but need to be accompanied by size markers, and to retain visible background signal within the linear range (i.e. should not be saturated). The boundaries of panels with low background have to be demarked with black lines. Splicing of panels should only be considered if unavoidable, and must be clearly marked on the figure, and noted in the legend with a statement on whether the samples were obtained and processed simultaneously. Quantitative comparisons between samples on different gels/blots are discouraged; if this is unavoidable, it has to be performed for samples derived from the same experiment with gels/blots were processed in parallel, which needs to be stated in the legend.

Regardless of format, all figures must be vector graphic compatible files, not supplied in a flattened raster/bitmap graphics format, but should be fully editable, allowing us to highlight/copy/paste all text and move individual parts of the figures (i.e. arrows, lines, x and y axes, graphs, tick marks, scale bars etc). The only parts of the figure that should be in pixel raster/bitmap format are photographic images or 3D rendered graphics/complex technical illustrations.

Unprocessed scans of all key data generated through electrophoretic separation techniques need to be presented in a supplementary figure that should be labeled and numbered as the final supplementary figure, and should be mentioned in every relevant figure legend. This figure does not count towards the total number of figures and is the only figure that can be displayed over multiple pages, but should be provided as a single file, in PDF or TIFF format. Data in this figure can be

displayed in a relatively informal style, but size markers and the figures panels corresponding to the presented data must be indicated.

The total number of Supplementary Figures (not including the “unprocessed scans” Supplementary Figure) should not exceed the number of main display items (figures and/or tables (see our Guide to Authors and March 2012 editorial <http://www.nature.com/ncb/authors/submit/index.html#suppinfo>; <http://www.nature.com/ncb/journal/v14/n3/index.html#ed>). No restrictions apply to Supplementary Tables or Videos, but we advise authors to be selective in including supplemental data.

GUIDELINES FOR EXPERIMENTAL AND STATISTICAL REPORTING

REPORTING REQUIREMENTS – We ask authors to complete a Reporting Summary that collects information on experimental design and reagents. We hope this will aid in your evaluation of the paper. The Reporting Summary can be found here <https://www.nature.com/documents/nr-reporting-summary.pdf> Please note that these forms are dynamic ‘smart pdfs’ and must therefore be downloaded and completed in Adobe Reader. We will then flatten them for ease of use. If you would like to reference the guidance text as you complete the template, please access these flattened versions at <http://www.nature.com/authors/policies/availability.html>.

We strongly recommend the presentation of source data for graphical and statistical analyses as a separate Supplementary Table, and request that source data for all independent repeats are provided when representative experiments of multiple independent repeats, or averages of two independent experiments are presented. This supplementary table should be in Excel format, with data for different figures provided as different sheets within a single Excel file. It should be labelled and numbered as one of the supplementary tables, titled “Statistics Source Data”, and mentioned in all relevant figure legends.

Version 3:

Decision Letter:

25th November 2024

Dear Hans,

Thank you for submitting your revised manuscript "Multipotent neural stem cells originating from neuroepithelium exist outside of the mouse central nervous system" (NCB-S44254C). It has now been seen by the original referees #2 and #4 and their comments are below. The reviewers find that the paper has improved in revision, and therefore we'll be happy in principle to publish it in Nature Cell Biology, pending minor revisions to comply with our editorial and formatting guidelines.

If the current version of your manuscript is in a PDF format, please email us a copy of the file in an editable format (Microsoft Word or LaTeX)– we cannot proceed with PDFs at this stage.

Thank you again for your interest in Nature Cell Biology. Please do not hesitate to contact me if you have any questions.

Best wishes,
Stelios

Stylianos Lefkopoulos, PhD
He/him/his
Senior Editor, Nature Cell Biology
Springer Nature
Heidelberger Platz 3, 14197 Berlin, Germany

E-mail: stylianos.lefkopoulos@springernature.com
Twitter: [@s_lefkopoulos](https://twitter.com/@s_lefkopoulos)
LinkedIn: [linkedin.com/in/stylianos-lefkopoulos-81b007a0](https://www.linkedin.com/in/stylianos-lefkopoulos-81b007a0)

Reviewer #2 (Remarks to the Author):

In the current revision of the manuscript, the authors provide further analyses to compare the pNSCs to tissue resident stem cells, along with brain derived NSCs, to show a greater degree of similarity to CNS NSCs. Furthermore, they show where the brain derived NSCs defer from pNSCs. These results, along with the lineage tracing data presented, address the issues raised in the previous round of revision. The authors further posit an argument on why these pNSCs do not represent biological noise, whether these arguments are valid requires work beyond the scope of the current manuscript. The arguments and results warrant publication. The remaining questions, as to how a certain population of NECs "escape" to form the peripheral NSC pool, and how this ratio can be modulated does fall outside the scope of this study.

Reviewer #4 (Remarks to the Author):

In this manuscript, Han et al., have cited some references to discuss whether the Cre+ cells in Wnt1-Cre mice are Sox1-cells at embryonic or postnatal stages. Just as the authors elaborated, the pNSCs cannot be labeled by Wnt1-Cre which means pNSCs are not derived from NCC and midbrain which can be labeled by Wnt1-Cre. Same as last time, I support the publication of this paper.

Version 4:

Decision Letter:

Dear Hans,

I am pleased to inform you that your manuscript, "Multipotent neural stem cells originating from neuroepithelium exist outside of the mouse central nervous system", has now been accepted for publication in Nature Cell Biology. Congratulations to you and the whole team!

Once your paper has been scheduled for online publication, the Nature press office will be in touch to confirm the details. An

online order form for reprints of your paper is available at <https://www.nature.com/reprints/author-reprints.html>. All co-authors, authors' institutions and authors' funding agencies can order reprints using the form appropriate to their geographical region.

Publication is conditional on the manuscript not being published elsewhere and on there being no announcement of this work to any media outlet until the online publication date in *Nature Cell Biology*.

Please note that *Nature Cell Biology* is a Transformative Journal (TJ). Authors may publish their research with us through the traditional subscription access route or make their paper immediately open access through payment of an article-processing charge (APC). Authors will not be required to make a final decision about access to their article until it has been accepted. [Find out more about Transformative Journals](https://www.springernature.com/gp/open-research/transformative-journals)

If you have not already done so, we strongly recommend that you upload the step-by-step protocols used in this manuscript to protocols.io (<https://protocols.io>), an open online resource that allows researchers to share their detailed experimental know-how. All uploaded protocols are made freely available and are assigned DOIs for ease of citation. Protocols and Nature Portfolio journal papers in which they are used can be linked to one another, and this link is clearly and prominently visible in the online versions of both. Authors who performed the specific experiments can act as primary authors for the Protocol as they will be best placed to share the methodology details, but the Corresponding Author of the present research paper should be included as one of the authors. By uploading your Protocols onto protocols.io, you are enabling researchers to more readily reproduce or adapt the methodology you use, as well as increasing the visibility of your protocols and papers. You can also establish a dedicated workspace to collect your lab Protocols. Further information can be found at <https://www.protocols.io/help/publish-articles>.

Nature Cell Biology encourages authors presenting evidence for cell, biological, molecular, and genetic interactions to consider communicating these findings using Biofactoid (<https://biofactoid.org/>). This tool helps users share a searchable representation of interactions (e.g. binding, gene expression, post-translational modification) between genes, gene products, or chemicals. Information added to Biofactoid, with author attribution, is shared on social media and public databases, such as Pathway Commons, where it can be discovered and analyzed in the context of a large and growing corpus of knowledge.

With kind regards,
Stelios

Stylianos Lefkopoulou, PhD
He/him/his
Senior Editor, Nature Cell Biology
Springer Nature

Heidelberger Platz 3, 14197 Berlin, Germany

E-mail: stylianos.lefkopoulos@springernature.com

Twitter: @s_lefkopoulos

LinkedIn: [linkedin.com/in/stylianos-lefkopoulos-81b007a0](https://www.linkedin.com/in/stylianos-lefkopoulos-81b007a0)

** Visit the Springer Nature Editorial and Publishing website at http://editorial-jobs.springernature.com?utm_source=ejp_NCB_email&utm_medium=ejp_NCB_email&utm_campaign=ejp_NCB for more information about our career opportunities. If you have any questions please click [here](mailto:editorial.publishing.jobs@springernature.com).**

Response to Reviewers

We sincerely thank the Reviewers for their constructive comments and insightful suggestions. We feel that the new datasets that were requested greatly contribute to the paper and further strengthen our conclusions. We have added a large amount of new data and made extensive changes to the manuscript.

Please find below a point-to-point response to the Reviewers' comments.

Reviewers' comments:

Reviewer #1 (Remarks to the Author):

In this manuscript the authors propose the isolation of CNS NSCs from embryonic limb or adult lung tissue. They base this on the expression of Nestin as shown by in vivo fate mapping, which comprise all the neurosphere-forming cells. However, Nestin has been shown to be expressed also in other cell types in the lung (e.g. endothelial cells, Bhagwani et al., 2019) and the authors did not show self-renewal and multipotent differentiation for these sorted cells. The authors further demonstrate that neurospheres emerging from adult lung tissue (without sorting for nestin-Cre/GFP) can self-renew for many passages and differentiate into neurons, GFAP+ and S100b+ cells and MBP/Olig2+ cells in vitro as well as after transplantation into the adult brain in vivo. However, they do not control for cell fusion nor show how frequent these respective cell types are in vivo. Finally, the authors present scRNA-seq data from adult lung tissue and identify an interesting cluster expressing some genes known to be also expressed in neural development and oligodendrocytes. However, several of these genes are also expressed in non-myelinating Schwann cells (e.g. Gfra3, Thai et al., 2019). Indeed, the manuscript lacks rigorous comparison to neural crest (stem) cells and Schwann cells, e.g. in Figure 1F, Figure 2-4. While the cells isolated and amplified in vitro in this manuscript may represent an interesting population of stem cells, the claim that these are CNS NSCs and derive from neuroectoderm is rather premature and not yet fully supported by the data.

We agree with the Reviewer that Nes-GFP transgene is not specific for labelling NSCs in the lung tissue. In Nes-GFP mice, GFP is driven by the promoter and second intron enhancer of Nestin, which should be specifically expressed in neural stem and progenitor cells (Yu et al., 2008). However, our single cell-RNA-seq data showed that the Nes-GFP reporter was also

expressed in other cell types in the lung (new Fig. 6b; new Extended Data Fig. 6e and 6f), indicating that Nes-GFP is actually not specific for labelling NSCs in the lung tissue. We clearly mentioned this in the new main text (page 10, line 311). Instead of using the neurosphere-forming method, we've used the monolayer culture system to derive peripheral NSCs (pNSCs) in all our experiments, as primary neurospheres comprise a mixture of different type of cells. By using the colony formation assay in the monolayer culture condition, we found that only few Nes-GFP⁺ cells could form NSC-like colonies (new Fig. 3a-3f). All these results demonstrate that pNSCs exist in Nes-GFP⁺ cells, but most Nes-GFP⁺ are not pNSCs. In our revised manuscript, by using new transgenic mouse lines and scRNA-seq analysis, we eventually found the specific NSC marker, Sox1, for labelling and tracing pNSCs, which is addressed in detail below in a point-to-point response.

To demonstrate the self-renewal capacity and NSC properties of these sorted cells, we comprehensively investigated the self-renewal, NSC marker expression, genome-wide transcriptional profile, and epigenetic features of pNSCs derived from Nes-GFP⁺ adult lung and postnatal tail cells without low-pH treatment. We found that pNSCs showed similar properties to brain control NSCs in all these aspects (new Fig. 3g-3k; new Extended Data Fig. 3f-3i). Furthermore, we also showed the self-renewal capacity and NSC marker expression of pNSCs derived from sorted Wnt1-Cre-mTmG Tomato⁺/GFP⁻ cells (new Extended Data Fig. 5b-5e), Sox2-GFP⁺ cells (new Extended Data Fig. 8c-8g), Sox1-GFP⁺ cells (new Extended Data Fig. 9d-9f), and Sox1-Cre-mTmG GFP⁺/Tomato⁻ cells (new Extended Data Fig. 11c-11e).

To demonstrate the multipotent differentiation of the sorted cells, we generated clonal adult lung and postnatal tail pNSCs lines from Nes-GFP⁺ sorted cells without low-pH treatment (new Fig. 4a) and investigated their multipotent differentiation *in vitro* and *in vivo*. We found that pNSCs could differentiate into neurons, astrocytes, and oligodendrocytes both *in vitro* and *in vivo* (new Fig. 4b, 4d-4f; new Extended Data Fig. 4a and 4b). We quantified the *in vitro* and *in vivo* differentiation efficiencies of pNSCs into the three neural lineages and found that they were similar to those of brain control NSCs (new Fig. 4c and 4g). We also ruled out the possibility of cell fusion between the transplanted cells and endogenous cells *in vivo* (new Extended Data Fig. 4c-4f). We also showed the *in vitro* multipotent differentiation of pNSCs derived from sorted Wnt1-Cre-mTmG Tomato⁺/GFP⁻ cells (new Extended Data Fig. 5f and 5g), Sox1-GFP⁺ cells (new Extended Data Fig. 9g and 9h), and Sox1-Cre-mTmG GFP⁺/Tomato⁻ cells (new Extended Data Fig. 11f and 11g). The differentiation efficiencies of these pNSCs

into the three neural lineages were similar to those of brain control NSCs. All these data demonstrate that the differentiation ability of pNSCs is similar to that of brain control NSCs. This question is addressed in detail below in point-to-point responses 3 and 4.

Given that neural crest stem cells (NCSCs) can be derived outside of the CNS, we sought to assess whether the derived pNSCs are NCSCs. To this end, we carefully compared pNSCs and NCSCs on marker expression and differentiation. Our immunostaining results demonstrated that pNSCs were different from NCSCs (new Fig. 5a and 5b). To further genetically confirm that pNSCs are not NCSCs and do not come from neural crest cells (NCCs), we used lineage tracing method by crossing *Wnt1-Cre* mouse (Danielian et al., 1998) with R26-mTmG mouse (Muzumdar et al., 2007) to label NCCs and their progeny cells, such as NCSCs and Schwann cells (new Fig. 5c). pNSCs derived from postnatal *Wnt1-Cre-mTmG* mouse tail and dorsal root ganglia (DRG) were Tomato⁺/GFP⁻ (new Fig. 5d and 5e), indicating that pNSCs do not originate from NCCs and are not NCSCs or Schwann cells. Importantly, bulk RNA-seq analysis showed that the whole-genome transcription pattern of pNSCs was similar to that of brain control NSCs, but clearly different from that of DRG NCSCs (new Fig. 5f and g; new Extended Data Fig. 5h). Our immunostaining, RNA-seq, and lineage tracing analysis allowed us to rigorously compare pNSCs to NCSCs, and to demonstrate that pNSCs do not originate from NCCs and are distinct from NCSCs. This question is addressed in detail below in point-to-point responses 1 and 2.

We thank the Reviewer for pointing out that several genes expressed in the putative pNSCs from *Nes-GFP*⁺ lung scRNA-seq data are also expressed in non-myelinating Schwann cells, e.g. *Gfra3*. *Gfra3* is actually a NC-derived cell marker. We corrected this mistake in the main text and changed the *Gfra3* gene to *Nestin* (Page 10, line 312). In the revised manuscript, we realized that several NSC markers expressed in the *Nes-GFP* putative pNSC cluster are also expressed in NC-derived cells, such as *Apoe*, *Fabp7*, and *Id3*. Many Schwann cell markers were also observed in this cluster cells, indicating that the putative pNSC cluster in *Nes-GFP*⁺ lung cells actually comprised a mixture of different type of cells, including pNSCs and Schwann cells. We corrected this result in the main text and renamed this cluster cells as neural cells (page 10, line 319-324) (new Fig. 6b; new Extended Data Fig. 7a-7e). In our revised manuscript, we used *Sox1-GFP* to trace pNSCs and systematically analyzed new scRNA-seq data on *Sox1-GFP*⁺ tail and brain cells. We found that *Sox1-GFP*⁺ pNSCs expressed NSC-specific markers and showed molecular features similar to brain NSCs (new Fig. 7; new Extended Data Fig. 10d).

Furthermore, we also showed that pNSCs highly expressed NSC-specific markers but not Schwann cell/NC-derived cell markers. In contrast, Schwann cells highly expressed Schwann cell/NC-derived cell markers but not NSC-specific markers, indicating that pNSCs are clearly different from Schwann cells or NC-derived cells (new Extended Data Fig. 10e and 10f). This question is addressed in detail below in point-to-point response 7.

Finally, we performed new lineage tracing analysis using Sox1-Cre-mTmG transgenic mouse and clearly showed that pNSCs originate from neuroepithelium (new Fig. 8a-8d; Extended Data Fig. 11a and 11b). This is addressed in detail in the following point-to-point response 6.

Taking all these results together, we have provided evidence that the derived pNSCs express multiple NSC-specific markers and exhibit cell morphology, self-renewing capacity, genome-wide transcriptional profile, epigenetic features, and differentiation ability similar to those of brain control NSCs. pNSCs are distinct from NCSCs and do not originate from NCCs.

Actually, pNSCs are from Sox1⁺ cells and originate from neuroepithelial cells. We show that pNSCs distribute within lung and tail tissues. Our new scRNA-seq data identifies that pNSCs *in situ* show molecular features similar to brain NSCs but different from Schwann cells.

Furthermore, many pNSCs that migrate out of the neural tube can differentiate into neurons and limited numbers of glia cells during embryonic development. With our extensive new data, we hope that the Reviewer now finds that our claim that pNSCs with CNS NSC properties exist in peripheral tissues and are derived from neuroectoderm is well supported.

Necessary data to support conclusions:

1) Include RNA-seq data from neural crest-derived stem and progenitor cells in Figure 1F.

We have followed the Reviewer's suggestion and compared the RNA-seq data between pNSCs, brain NSCs, and neural crest-derived stem cells. The global genome heatmap (new Fig. 5f), the hierarchical clustering analyses (new Fig. 5g), and pairwise scatter plots (new Extended Data Fig. 5h) demonstrated that the genome-wide gene expression pattern of pNSCs was similar to that of brain control NSCs, but clearly different from that of NCSCs.

2) Analyse several exclusive Schwann cell markers in Figure 2B.

Given that NCSCs can be derived outside of the CNS and to assess whether the derived pNSCs are indeed NCSCs, we stained Nes-GFP⁺ adult lung and postnatal tail pNSCs with

NSC- and neural crest (NC)-specific/Schwann cell markers. Passage-3 Nes-GFP⁺ lung and tail pNSCs were found to express Sox2 and the NSC-specific marker Olig2, but they did not express the NC-specific/Schwann cell markers Sox10 and p75. In contrast, passage-3 NCSCs from adult WT mouse dorsal root ganglia (DRG) expressed Sox2, Sox10, and p75, but did not express Olig2 (new Fig. 5a). On differentiation, pNSCs efficiently differentiated into Tuj1⁺ neurons, GFAP⁺ astrocytes, and MBP⁺ oligodendrocytes, which all did not express p75 (new Fig. 5b). In contrast, it was very difficult to observe Tuj1⁺ and MBP⁺ cells in the differentiation of adult DRG NCSCs. Furthermore, Tuj1⁺, GFAP⁺, and MBP⁺ cells from DRG NCSCs differentiation expressed the NC-specific marker p75 (new Fig. 5b). Furthermore, Wnt1-Cre-mTmG Tomato⁺/GFP⁻ pNSCs (new Extended Data Fig. 5d), Sox2-GFP⁺ pNSCs (new Extended Data Fig. 8d), and Sox1-Cre-mTmG pNSCs (new Extended Data Fig. 11e) expressed Olig2 but not NC-specific markers. These results demonstrate that pNSCs are different from NCSCs.

By using Wnt1-Cre-mTmG new lineage tracing analysis, we further genetically confirmed that pNSCs were not NCSCs and did not originate from NCCs (new Fig. 5c-5e; new Extended Data Fig. 5a-5g). Bulk RNA-seq data analysis further showed that pNSCs were different from NCSCs (new Fig. 5f and 5g; new Extended Data Fig. 5h). From scRNA-seq analysis, we also showed that pNSCs highly expressed NSC-specific markers but not Schwann cell/NC-derived cell markers. In contrast, Schwann cells highly expressed Schwann cell/NC-derived cell markers but not NSC-specific markers, indicating that pNSCs are clearly different from Schwann cells or NC-derived cells (new Extended Data Fig. 10e and 10f). Taken together, these results demonstrate that pNSCs do not originate from NCCs/NC-derived cells and are distinct from NCSCs or Schwann cells.

3) Control for cell fusion by transplanting cells expressing Cre into a reporter mouse line.

To exclude the cell fusion between transplanted cells and endogenous cells, we transplanted 3×10^5 wild-type (WT) mouse postnatal tail pNSCs, expressing the CAG promoter-driven Cre recombinase and the CAG promoter-driven H2B-GFP (new Extended Data Fig. 4c and 4d), into the brain of postnatal day 1 R26-loxp-Stop-loxp-RFP mouse (Luche et al., 2007). If the transplanted GFP⁺ cells had fused with endogenous cells, the fused cells would have become GFP⁺/RFP⁺ after Cre excision of the loxP-flanked stop sequences (new Extended Data Fig. 4e). Immunohistological staining results showed that transplanted tail pNSCs differentiated into GFP⁺/GFAP⁺ astrocytes, GFP⁺/NeuN⁺ neurons, and GFP⁺/MBP⁺ oligodendrocytes, and

we did not observe any GFP⁺/RFP⁺ cells (new Extended Data Fig. 4f), indicating that there was no cell fusion between the transplanted cells and the endogenous cells.

4) Provide quantification of the proportion of NeuN⁺, S100b and MBP⁺ cells amongst all mCherry⁺ cells from at least 3 animals.

We have followed the Reviewer's suggestion and repeated the transplantation experiments. To assess the ability of pNSCs derived without low-pH treatment to survive and differentiate *in vivo*, instead of IdNSCs we transplanted 3 x 10⁵ clonal Nes-GFP⁺ adult lung and postnatal tail pNSCs, labelled with CAG promoter-driven H2B-Tomato (new Extended Data Fig. 4a), into the cortex of postnatal day 1 (P1) C57Bl/6 mice brain. Six weeks after transplantation, all mice had survived the transplantation procedure and there was no indication of a tumor. Immunohistological staining results revealed that transplanted lung and tail pNSCs and brain NSCs did not express the NSC marker Nestin and cell-cycle marker Ki67, indicating that the pNSCs had lost their stem cell identity after transplantation and therefore can be used safely for cell transplantation approaches (new Extended Data Fig. 4b). Importantly, transplanted lung, tail pNSCs, and brain NSCs were found to differentiate into neurons, astrocytes, and oligodendrocytes *in vivo*, as evidenced by the presence of Tomato⁺/NeuN⁺, Tomato⁺/GFAP⁺, Tomato⁺/MBP⁺ cells, respectively (new Fig. 4d-f). For quantification, multiple sections (five or six) from one animal and three independent animals were examined for each group. The efficiency of differentiation into neurons, astrocytes, and oligodendrocytes from brain control NSCs and pNSCs was calculated by the average number of NeuN⁺, GFAP⁺, and MBP⁺ cells among H2B-Tomato cell number in each animal. The relative differentiation efficiency was normalized to brain control NSC injections. We have added this description into the Methods (page 25, line 816-page 26, line 862). We observed that the differentiation efficiencies of pNSCs into the three neural lineages *in vivo* were comparable to those of brain control NSCs (new Fig. 4g). These results demonstrate that pNSCs exhibit characteristics similar to brain control NSCs when transplanted into the mouse brain.

5) Provide 2) and 4) from neurosphere cells derived from Nestin-Cre fate mapped cells.

We found that the Nes-GFP transgene was expressed in many cell types and was not specific for labelling pNSCs. Similarly, Nestin-Cre should also not be specific for labelling neuroepithelial cells (NECs). Therefore, we excluded the data from the Nestin-Cre fate mapped cells.

In our revised manuscript, we used Sox1-Cre-mTmG transgenic mouse line to label neuroepithelial cells. We discussed this question in detail below in point-to-point response 6.

6) Ideally use a more specific marker to fate-map neuro-ectoderm-derived cells as nestin is also expressed in many cell types.

We thank the Reviewer for raising this critical point. As Nestin is also expressed in many cell types, Nestin-Cre should not be specific for labelling neuroepithelial cells. In our revised manuscript, we used Sox1-Cre mouse crossing with R26-mTmG mouse to label the neuro-ectoderm-derived cells (Takashima et al., 2007) (new Fig. 8a). In the Sox1-Cre-mTmG E8.5 embryos, GFP expression was detected exclusively in neuroepithelial cells (NECs) in the trunk and caudal regions. In the cranial region, GFP expression was detected almost exclusively in NECs, except for very few GFP⁺ cells that had started to migrate out of the neural fold (new Fig. 8b; new Extended Data Fig. 11a and 11b). These results confirm that the NECs were efficiently and specifically labelled by GFP signal in Sox1-Cre-mTmG mouse. Notably, pNSCs from Sox1-Cre-mTmG postnatal lung and tail tissues were GFP⁺/Tomato⁻ and could be derived only from GFP⁺ cells, indicating that pNSCs indeed originate from NECs (new Fig. 8c and 8d). GFP⁺/Tomato⁻ pNSCs derived from Sox1-Cre-mTmG postnatal lung and tail tissues could be maintained for more than 50 passages, indicating their self-renewal capacity (new Extended Data Fig. 11c). Immunostaining and RT-qPCR results showed that these cells expressed NSC markers (new Extended Data Fig. 11d and 11e). Sox1-Cre-mTmG pNSCs could differentiate into neurons, astrocytes, and oligodendrocytes *in vitro*, indicating multipotency (new Extended Data Fig. 11f). The differentiation efficiencies of pNSCs into the three neural lineages were comparable to those of brain control NSCs (new Extended Data Fig. 11g). Sox1-Cre-mTmG pNSCs did not express the NC-specific markers Sox10, and p75, indicating that these pNSCs are different from NCSCs or Schwann cells (new Extended Data Fig. 11e). These data demonstrate the NSC properties of Sox1-Cre-mTmG pNSCs. Taken together, our results obtained using Sox1-Cre-mTmG lineage tracing analysis provided evidence that pNSCs originate from Sox1⁺ NECs (new Fig. 8e).

7) Figure 4: compare expression of markers shown in e and g to neural crest derived cells and examine Schwann cell markers in this population.

In the revised manuscript, we realized that several NSC markers expressed in the Nes-GFP putative pNSC cluster are also expressed in NC-derived cells, such as *ApoE*, *Fabp7*, and *Id3*. Many Schwann cell markers were also observed in this cluster cells (new Extended Data Fig. 7e), indicating that the putative pNSC cluster in Nes-GFP⁺ lung cells actually comprised a mixture of different type of cells, including pNSCs and Schwann cells. We corrected this result in the main text and renamed this cluster as neural cells (page 10, line 319-324) (new Fig. 6b; new Extended data Fig. 7).

By using another transgenic mouse line, we provided evidence that pNSCs were actually from Sox1⁺ cells (new Fig. 6d-6h; new Extended Data Fig. 9d-dh). Then we performed scRNA-seq experiments on postnatal tail and adult brain Sox1-GFP⁺ cells and systematically analyzed the new scRNA-seq data. We found that tail pNSCs showed molecular features similar to brain NSCs, indicating the NSC properties of pNSCs (new Fig. 7; new Extended Data Fig. 10d). We did not observe Schwann cells or a NC-derived cells cluster from Sox1-GFP⁺ scRNA-seq dataset (new Fig. 7a). This makes sense, as Schwann cell/NC-derived cells should not express Sox1. Fortunately, we did observe a Schwann cell cluster from the Sox2-GFP⁺ scRNA-seq dataset (new Fig. 6c; new Extended Data Fig. 8i; Supplementary Table3). We followed the Reviewer's suggestion and then assessed the marker expression in Sox1-GFP⁺ tail pNSCs and Sox2-GFP⁺ Schwann cells. We found that pNSCs highly expressed NSC-specific markers but not Schwann cell/NC-derived cell markers. In contrast, the Schwann cell cluster highly expressed Schwann cell/NC-derived cell markers but not NSC-specific markers, suggesting that pNSCs are different from Schwann cells or NC-derived cells (new Extended Data Fig. 10e and 10f). Furthermore, as we discussed above, we rigorously compared pNSCs to NCSCs and demonstrated that pNSCs do not originate from NCCs/NC-derived cells and are distinct from NCSCs (point-to-point response 2). Taken together, these results provided evidence that pNSCs are not NCSCs, Schwann cells, or NC-derived cells.

Reviewer #2 (Remarks to the Author):

In this manuscript, Han et al. reported derivation of cells with CNS neural stem cell (NSC) properties in vitro from different peripheral tissues, including mouse embryonic limb, postnatal tail tip, and adult lung. They presented evidence that upon culturing these cells with growth factors can be passage for over 50 times, exhibit transcriptomes similar to CNS NSCs, and can give rise to three lineages in the CNS both in vitro and after transplantation. They further showed that these NSCs are likely derived from Nestin-expressing cells. They concluded that multipotent neural stem cells with unique molecular features exist outside of the mouse central nervous system. In addition to continuous neurogenesis from endogenous quiescent neural stem cells in the discrete regions of the adult mouse brain, continuous neurogenesis also occurs in the peripheral nervous system, such as olfactory epithelium and gut. Therefore, neural stem cells are not limited to CNS. Previous studies have already shown that neural progenitor cells derived from the peripheral tissue, once cultured as neurospheres for multiple passages, can give rise to CNS cell types in vitro and after transplantation (e.g. PMID: 21525278). The current study does not provide any evidence on the properties of putative neural stem cells in the peripheral tissue in vivo. Are these putative stem cells quiescent normally? Can these putative stem cells be activated (e.g. after injury) and exhibit self-renewal and differentiation in vivo? How do their transcriptomes compare to quiescent neural stem cells in the brain and neural crest stem cells in the PNS? The authors have the tool of nestin-cre to perform lineage-tracing and other analyses of these cells under in vivo conditions in mice. Without any insight into the endogenous properties of these putative stem cells in vivo, the observed effect could be due to in vitro culture reprogramming and there is little novelty from the current study for publication in Nature Cell Biology.

We agree with the Reviewer that continuous neurogenesis occurs in the peripheral nervous system, such as olfactory epithelium and gut in the adult animal. However, the new neurons generated in these two regions are from neural crest stem cells (Suzuki et al., 2013; Bixby et al., 2002; Kruger et al., 2002). NCSCs, with limited self-renewal ability, are different from NSCs in gene expression and differentiation potential (Binder et al., 2011; Vidal et al., 2014; Nagoshi et al., 2008). It may be more appropriate to describe the NCSCs as progenitor cells rather than true stem cells (Achilleos and Trainor, 2012). Therefore, to our opinion, NSCs are only found in the CNS until now.

As mentioned by the Reviewer, previous studies have already shown that neural progenitor cells derived from the peripheral tissue-dorsal root ganglia (DRG), once cultured as

neurospheres for multiple passages, can give rise to CNS cell types *in vitro* and after transplantation (Binder et al., 2011; Weber et al., 2015). Our new immunostaining, lineage tracing, and bulk RNA-seq results showed that pNSCs are distinct from NCSCs and do not originate from NCCs or their daughter cells (new Fig. 5; new Extended Data Fig. 5). These data exclude the possibility that pNSCs have been reprogrammed from NCC-derived cells. Notably, we found that pNSCs existed in DRG tissue (new Fig. 5e). Previous studies derived NCSCs *in vitro* by using the neurosphere formation method (Binder et al., 2011; Weber et al., 2015). The primary neurospheres could easily be a mixture of different types of cells, including pNSCs. It is likely that pNSCs are dominantly expanded after culture and create a false reprogramming impression (Binder et al., 2011; Weber et al., 2015). Furthermore, pNSCs could be derived only from Sox1⁺ cells (new Fig. 6g and 6h). Sox1 is the earliest known specific marker of NSCs in the mouse embryo (Wood and Episkopou, 1999; Aubert et al., 2003). This data suggests that pNSCs are from endogenous stem cells and excludes the possibility that pNSCs are reprogrammed from other somatic cells, such as fibroblasts. Importantly, our scRNA-seq data identified that pNSCs *in situ* showed molecular features similar to brain control NSCs but different from Schwann cells/NC-derived cells (new Fig. 7; new Extended Data Fig. 10d-10f), not only confirming the NSC identity of the endogenous pNSCs but also further excluding the possibility that pNSCs have been reprogrammed from NC-derived cells. All these pieces of evidence lead us to conclude that the derivation of pNSCs is conceptually different from the process of reprogramming.

It was a longstanding process for us to identify the specific marker to trace pNSCs. Our scRNA-seq data from lung Sox2-GFP⁺ cells demonstrated that some cells coexpressed *Sox1* and *Sox2* in the neural cell cluster (new Fig. 6d). It is well known that NSCs and many other stem/progenitor cells express Sox2 (Ellis et al., 2004; Suh et al., 2007; Arnold et al., 2011), and Sox1 is the earliest known specific marker of NSCs in the mouse embryo (Wood and Episkopou, 1999; Aubert et al., 2003). These results prompted us to investigate the cells that coexpressed Sox1 and Sox2 in lung and tail tissues. Immunohistological staining results showed that Sox1⁺/Sox2⁺ cells indeed exist in lung and tail tissues. In postnatal mouse lung tissue, Sox1⁺/Sox2⁺ cells mainly distributed in the big bronchi epithelial cell wall (new Fig. 6e). This was also confirmed in adult mouse lung tissue (new Extended Data Fig. 9a). In postnatal mouse tail tissue, Sox1⁺/Sox2⁺ cells were found to be distributed under the skin as a short tube structure approximately in the middle of the tail in longitudinal axis (new Fig. 6f). By using a Sox1-GFP KI transgenic

reporter mouse line (Aubert et al., 2003), we confirmed that pNSCs were indeed from Sox1⁺ cells (new Fig. 6g and 6h). Our scRNA-seq data of tail Sox1-GFP⁺ cells demonstrated that pNSCs *in situ* showed molecular features similar to brain NSCs and that pNSCs were different from Schwann cells (new Fig. 7; new Extended Data Fig. 10d-10f), indicating the NSC identity of endogenous pNSCs. Lineage tracing analysis with Sox1-Cre-mTmG mouse lines demonstrated that pNSCs originate from neuroepithelial cells (new Fig. 8a-8e; new Extended Data Fig. 11). Furthermore, we observed that many pNSCs that migrate out of the neural tube could differentiate into neurons and limited numbers of glia cells during embryo development (new Fig. 8f and 8g; new Extended Data Fig. 12). All these new data demonstrate what the pNSCs are, where the pNSCs are distributed, what the molecular features of pNSCs *in situ* are, where the pNSCs originate, and how pNSCs differentiate during embryonic development. Thus, we have provided much evidence regarding the properties of pNSCs in peripheral tissues *in vivo*.

Our scRNA-seq data on Sox1-GFP⁺ cells showed that the majority of tail Sox1⁺ cells overlapped with brain qNSCs, indicating that the quiescent pNSCs are in a similar quiescent state as brain qNSCs (new Fig. 7b, 7c and 7f). The scRNA-seq data showed that 11.6% of tail Sox1-GFP⁺ cells overlapped with brain aNSCs (new Fig. 7f). Immunohistological staining showed that roughly 13.1% of postnatal tail Sox1⁺ pNSCs (n=520 cells) expressed the cell-cycle marker Ki67 (Fig. 7g). These data indicate the intrinsic self-renewal of certain pNSCs. Pseudotemporal ordering analysis revealed the developmental trajectory of qNSC-aNSC-NB in both tail pNSCs and brain NSCs, indicating the intrinsic differentiation ability of pNSCs *in vivo* (new Fig. 7h). Sox1-Cre-mTmG lineage tracing analysis further showed the differentiation ability of pNSCs *in vivo* during embryo development (new Fig. 8f and 8g; new Extended Data Fig. 12). All these data demonstrate the self-renewal and differentiation of pNSCs *in vivo*. We agree with the Reviewer that it is very interesting to investigate whether the quiescent pNSCs can be activated by injury or some other stimulus and participate in tissue regeneration. However, we felt that it would go beyond the scope of the current manuscript and will definitely be investigated further in the future.

In our revised manuscript, we compared pNSCs with DRG-derived NCSCs and found that pNSCs are distinct from NCSCs (new Fig. 5a and 5b). Bulk RNA-seq analysis also clearly showed that pNSCs were similar to brain NSCs but different from NCSCs (new Fig. 5f and 5g; new Extended Data Fig. 5h). Lineage tracing analysis using Wnt1-Cre-mTmG mouse further genetically confirmed that pNSCs do not originate from NCCs or their progeny cells

(new Fig. 5; new Extended Data Fig. 5). To compare the pNSCs with Schwann cells at the single-cell level, we assessed the marker expression between Sox1-GFP⁺ pNSCs and Schwann cells from the Sox2-GFP⁺ scRNA-seq dataset. pNSCs were found to highly express NSC-specific marker genes but not Schwann cell/NC-derived cell marker genes. In contrast, Schwann cells expressed Schwann cell/NC-derived cell marker genes but not NSC marker genes (new Extended Data Fig. 10e and 10f). Taken together, these data demonstrate that pNSCs do not originate from NCCs and are different from NCSCs and Schwann cells.

Taking all our data together, we have excluded the possibility that pNSCs have been reprogrammed from NCSCs *in vitro*, and provided much evidence regarding the NSC properties of endogenous pNSCs *in vivo*.

Specific comments:

1. In addition to analysis of properties of these putative neural stem cells *in vivo* as listed above, the authors need to show whether these cells isolated from *in vivo* (e.g. by Nestin-GFP) can give rise to CNS cell types *in vitro* and after transplantation *in vivo* without expansion with growth factors EGF and FGF-2 to reveal their intrinsic properties. Can these cell also differentiate into neural crest lineages?

As the Nes-GFP transgene is not specific for labelling pNSCs and is expressed in many cell types, we could not use Nes-GFP to trace the pNSCs. In the revised manuscript, we found that pNSCs were from Sox1⁺ cells (new Fig. 6d-6h; new Extended Data Fig. 9). Therefore, we used Sox1-GFP to trace pNSCs. To reveal the intrinsic properties of pNSCs without expansion with growth factors EGF and FGF-2 *in vitro*, we isolated Sox1-GFP⁺ cells from lung and tail tissues and cultured them directly in differentiation medium. We observed that Sox1-GFP⁺ cells without expansion could differentiate into Tuj1⁺ neurons, GFAP⁺ astrocytes, and MBP⁺ oligodendrocytes, indicating the intrinsic NSC properties of pNSCs. All these differentiated cells did not express the NC-specific marker p75, indicating that Sox1-GFP⁺ pNSCs do not differentiate into neural crest lineage *in vitro* (new Extended Data Fig. 9i).

As the pNSC population is relatively rare, we could not obtain a sufficient number of Sox1⁺ pNSCs by FACS to perform *in vivo* transplantation experiments. However, the developmental trajectory analysis of scRNA-seq data from Sox1-GFP⁺ cells indicates the intrinsic differentiation ability of pNSCs *in vivo* (new Fig. 7h). Furthermore, Sox1-Cre-mTmG lineage tracing analysis showed that many pNSCs that migrated out of the neural tube could

differentiate into neurons and limited numbers of glia cells during embryo development (new Fig. 8f and 8g; Extended Data Fig. 12). All these results indicate the intrinsic NSC properties of pNSCs *in vivo*.

2. Single-cell RNA-seq of Nestin-GFP cells showed expression of some markers of quiescent NSCs and neural crest stem cells, but not others. There is a need of a systematic comparison with neural crest stem cells and brain quiescent NSCs at the single cell levels. Lack of expression of some markers is not a good indicator, given the limited depth of single-cell RNA-seq.

We thank the Reviewer for this constructive suggestion. We have followed the Reviewer's suggestion and systematically analyzed the scRNA-seq data from Sox1-GFP⁺ cells and compared pNSCs to brain NSCs or Schwann cells.

Notably, the majority of tail Sox1⁺ cells overlapped with brain qNSCs and aNSCs (new Fig. 7b and 7c). They expressed brain NSC markers, such as *Sox2*, *Sox1*, *Sox9*, *Slc1a3*, *Pax6*, *Fabp7*, *Nnat*, *Hes1*, *Id3*, *Id2*, *Hopx*, *Ptprz1*, *Apoe*, *Dbi*, *Mt2*, *Ttyh1*, *Aldoc*, and *Cspg5*, indicating the NSC property of these cells (new Fig. 7d; new Extended Data Fig. 10d). Examination of Gene Ontology (GO) enrichment of NSC cluster cells showed that nervous system development and neurogenesis appeared in the top 5 GO terms (new Fig. 7e), confirming the NSC identity of these clusters. To assess whether Sox1-GFP⁺ pNSCs are similar to Schwann cells, we examined the marker expression in pNSCs and Schwann cells from the Sox2-GFP⁺ scRNA-seq dataset. We found that both pNSCs and Schwann cells highly expressed *Sox2*. pNSCs highly expressed the NSC-specific markers *Slc1a3*, *Ascl1*, *Pax6*, *Cspg5*, *Sox9*, and *Sox1*, but not the Schwann cell/NC-derived cell markers *Sox10*, *Ngfr*, *Foxd3*, *MPZ*, *ErbB3*, *Mbp*, and *Plp1*. In contrast, Schwann cells highly expressed Schwann cell/NC-derived cell markers, but not NSC-specific markers (new Extended Data Fig. 10e and 10f), indicating that pNSCs are different from Schwann cells. Altogether, these results demonstrate that a large number of tail Sox1-GFP⁺ cells show molecular features similar to brain NSCs, indicating the NSC properties of these cells.

Among the tail Sox1⁺ cells, about 66.3% of cells overlapped with brain qNSCs, indicating that the majority of pNSCs are in a quiescent state. About 11.6% and 18.1% of cells overlapped with brain aNSCs and NBs, respectively (new Fig. 7f). Immunohistological staining showed that roughly 13.1% of postnatal tail Sox1⁺ pNSCs (n=520 cells) expressed

the cell-cycle marker Ki67 (new Fig. 7g), not only confirming our sc-RNA-seq data but also indicating the self-renewal of certain pNSCs. Pseudotemporal order analysis revealed the development trajectory from qNSCs to aNSCs to NBs in both tail pNSCs and brain NSCs (new Fig. 7h), indicating the intrinsic NSC property of pNSCs *in vivo* and the ongoing neurogenesis process from pNSCs at the postnatal stage in tail tissue.

3. The authors should compare the transcriptome of acutely isolated nestin-GFP cells (via single-cell RNA-seq) with those after culturing in the presence of growth factors and passaging to see whether there are any substantial changes in gene expression, suggesting reprogramming.

Based on our new immunostaining, lineage tracing, bulk RNA-seq, and scRNA-seq analysis results, we excluded the possibility that pNSCs have been reprogrammed from NC-derived cells or another type of somatic cell. We have discussed this in detail above (Response to Reviewer #2, second paragraph).

4. The authors used nestin-cre fate mapping to argue that these putative stem cells have a CNS origin. However, the nestin-Cre tool is not specific enough, as evident in their single-cell RNA-seq analysis of nestin-GFP cells. Therefore, the origin of these putative stem cells is still a question.

We thank the Reviewer for raising this important point. We agree with the Reviewer that Nestin-Cre fate mapping is not specific enough for labelling the neuroepithelial cells. In the revised manuscript, we used Sox1-Cre mice crossing with R26-mTmG mice to label the neuroepithelial cells (Takashima et al., 2007) (new Fig. 8a). We showed that the neuroepithelial cells were efficiently and specifically labelled by GFP signal in the Sox1-Cre-mTmG mouse (new Fig. 8b; new Extended Data Fig. 11a and 11b). Notably, pNSCs from Sox1-Cre-mTmG postnatal lung and tail tissues were GFP⁺/Tomato⁻ and could be derived from only GFP⁺ cells, indicating that pNSCs indeed originate from NECs (new Fig. 8c and 8d). Therefore, with the new lineage tracing analysis, we have clarified the origin of the pNSCs. We have also discussed this above (Response to Reviewer #1, point 6).

5. The authors showed that transplanting IdNSCs into adult mouse cortex leads to generation of cells of neuronal and glial lineages. However, this result is not quantified. Is the propensity to yield terminally differentiated neuron/glial cell comparable from IdNSCs comparable to “control” NSCs?

We have followed the Reviewer's suggestion and repeated the transplantation experiments. Instead of IdNSCs, we repeated the transplantation experiment with clonal adult lung and postnatal tail pNSCs derived from Nes-GFP⁺ cells without low-pH treatment. We observed that lung and tail pNSCs could differentiate into neurons, astrocytes, and oligodendrocytes *in vivo* (new Fig. 4d-f). The differentiation efficiencies of pNSCs into the three neural lineages were comparable to those of brain control NSCs (new Fig. 4g). The new transplantation results demonstrate that pNSCs exhibit characteristics similar to brain control NSCs when transplanted into the mouse brain. We have also discussed this above (Response to Reviewer #1, point 4).

Other comments:

1. The authors need to explain control NSC in more detail, instead of citing a previous paper.

We have followed the Reviewer's suggestion and added more details about how to get the brain control NSCs. We have added the paragraph in the Methods (page 19, line 614-621): "For generation of brain control NSCs, the whole brain tissues from adult or postnatal mouse were cut and minced using a pair of scissors, and incubated with 0.25% trypsin (Invitrogen) at 37°C for 10 min. After trypsinization, 3 times volume of MEF medium was added, and the entire suspension was pipetted up and down to dissociate the tissues. The dissociated tissues were centrifuged and resuspended in NSC medium, and the cells were plated onto 6-well plates coated with gelatin and cultured in a humidified incubator at 37°C under 5% CO₂. After several passages, pure brain NSCs could be established."

2. The authors showed electrophysiological evidence of neuronal excitability, but the action potential is rather immature and no repetitive firing. The authors did not examine evidence of morphological and functional synapse formation of neural progeny.

We agree with the Reviewer that the action potential of our *in vitro* differentiated neurons was rather immature and that there was no repetitive firing. As the repeating firing was also not observed from brain control NSC-derived neurons, we speculated that our *in vitro* neuron differentiation protocol might be not ideal for generating fully matured neurons. Importantly, we did not observe any electrophysiological difference between brain control NSC- and IdNSC-derived neurons. Our main goal of the electrophysiological experiments is to provide evidence that pNSCs (including IdNSCs) can differentiate into neurons rather than get fully differentiated mature neurons. We rewrote the main text and mentioned the immature neuron differentiation: "These data show that although neurons, derived from IdNSCs and brain

control NSCs, were not fully mature under our *in vitro* differentiation conditions, neurons derived from IdNSCs acquired the electrical properties of neurons derived from control NSCs.” (page 5, line 151-154). Likewise, we thank the Reviewer for the suggestion to investigate for evidence of morphological and functional synapse formation of the neural progeny, which we will explore in future experiments.

3. The authors did not study netrin-GFP sorted cells from the limbs.

We first observed pNSC generation from WT mouse embryonic limb tissue cells. Based on this observation, we used different transgenic reporter mouse lines to provide evidence that pNSCs exist in the postnatal lung, tail and DRG, and adult lung tissues. As we were able to derive pNSCs from different tissues, the conclusion of the manuscript will not be affected without studying Nestin-GFP–sorted cells from limbs.

Reviewer #3 (Remarks to the Author):

Han, Schöler and co-workers describe isolation of a specific cell type, which shares some features with neuronal stem cells (NSCs). These cells were isolated from the embryonic limb, postnatal tail tip, and adult lung tissue after treatment in low pH before placement in NSC culture medium. Slowly proliferating cells emerged and in differentiating culture conditions, and after transplantation into mouse cortex, cells formed glia and neurons. The authors suggest that these "low-pH-derived neural stem cells" (or ldNSCs), are multipotent neural stem cells derived from embryonic neuroepithelial cells. It is claimed that the cells do not originate from neural crest (NC) cells like other peripheral neural cells including peripheral neurons, glia and Schwann cells. As the authors point out, this is contrary to the dogma that neural stem cells exist outside of the central nervous system. However, as pointed out in the specific comments below, the claim that NSCs exist in peripheral tissues remains unproven and extensive additional experiments would be required to make such a far-reaching claim. Also, it would be essential to demonstrate the existence of such cells in their natural environment and preferably understand if – and under what conditions – they can generate neurons and glia *in vivo*.

We thank the Reviewer for recognizing the importance and conceptual advance of our study. With our extensive additional experiments, we have provided evidence of the existence of pNSCs in their natural environment and their NSC properties both *in vitro* and *in vivo*.

Specific comments:

1. It is not possible to conclude that a specific cell type has certain properties unless the studied cells are clonally derived from a single cell. It remains unclear if the authors have performed any such single cell clonal analyses in the characterization of ldNSCs. Indeed, different types of cells with proliferative potential are likely present in the analyzed peripheral tissues (some with neural properties; see below). Thus, cultures defined as ldNSCs could be a heterogenous mixture of different types of stem-/progenitor cells with different potential to generate e.g. neurons or glia. This is also relevant when analyzing cells after transplantation. As presented, it remains unclear if a clonally derived stem cell from peripheral tissues has potential to form both glia and neurons.

We thank the Reviewer for raising this point. We have followed the Reviewer's suggestion and generated clonal adult lung and postnatal tail pNSCs lines from Nes-GFP⁺ sorted cells without low-pH treatment (new Fig. 4a) and investigated their multipotent differentiation both *in vitro* and *in vivo*. We found that clonal pNSCs could differentiate into neurons, astrocytes, and oligodendrocytes both *in vitro* and *in vivo* (new Fig. 4b, 4d-4f; new Extended Data Fig. 4a and 4b). We quantified the *in vitro* and *in vivo* differentiation efficiencies of pNSCs and found that they were similar to those of brain control NSCs (new Fig. 4c and 4g). We also ruled out the possibility of cell fusion between the transplanted cells and endogenous cells *in vivo* (new Extended data Fig. 4c-4f). All these results provided evidence that clonally derived pNSCs from peripheral tissues has potential to form glia and neurons. We have also discussed this above (response to Reviewer #1, point 3 and 4; response to Reviewer #2, specific comments 5).

2. Already from images in Fig. 1a-d and Ext Fig 1c,d it is concluded that the cultured cells have properties of NCSs. It is unclear why NSC-properties are presumed from these data alone. Separate channels should be shown in Fig. 1d.

First, we cultured the cells in NSC medium with EGF and bFGF growth factors, which is highly advantageous for culturing NSCs. Second, in monolayer culture condition, the morphologies of primary NSC-like cluster cells from limb and lung tissues were of a network pattern, which are very similar to that of brain NSCs (Conti et al., 2005) or iNSCs (Han et al., 2012). We have added the following sentence into the main text to explain why we speculated that this NSC-like cluster could be putative NSCs: "Because the cells were cultured in NSC medium containing EGF and bFGF growth factors, which is highly advantageous for culturing NSCs, and the morphologies of primary NSC-like cell clusters from limb and lung tissues were of a network pattern that was very similar to that of brain NSCs¹⁴ (Conti et al., 2005) or transcription factors (TF)-induced NSCs (iNSCs)¹⁵ (Han et al., 2012), we tentatively named these cells as 'embryonic limb low-pH NSCs' (eINSCs) and 'adult lung low-pH NSCs' (allINSCs), respectively. We took each single NSC-like cluster and expanded the cells in NSC medium." (page 3, line 95-101). Third, these cell clusters could be isolated, expanded, and passaged for more than 90 passages in NSC medium, indicating the self-renewal capacity of these stem cells. Fourth, the expanded cells expressed many NSC-specific markers, such as Sox2, Olig2, and Nestin. Based on these primary results, we thought that these cells have

features in common with NSCs. With our further characterization addressed below, we provided evidence that these cells are indeed NSCs.

We have followed the Reviewer's suggestion and separated channels are shown in Fig. 1d.

3. In Fig 1, IdNSCs were compared to i) TF-induced NSCs from a previous study; ii) control adult brain NSCs, and iii) fibroblasts. However, the comparison to two types of NSCs and fibroblasts by array hybridization provides rather limited information on cell type identity. This experiment only shows that IdNSCs are more similar to NSCs as compared to fibroblasts, and it remains unclear why the cells are referred to as NSCs based on these data (see further comments below).

The hierarchical clustering analysis of the whole transcription genome only showed the relative relationships that IdNSCs were more similar to NSCs as compared to fibroblasts (new Fig. 1g). However, the heatmap and pairwise scatter plot analysis showed the very similar gene expression pattern between IdNSCs and brain NSCs (new Fig. 1f; new Extended Data Fig. 1i).

We agree with the Reviewer that array hybridization experiments alone did not fully address cell type identity. Besides the transcriptome analysis, we also did many other experiments to characterize the identity of IdNSCs, including self-renewal capacity, gene expression, epigenetic features (new Fig. 1; Fig. 2; new Extended Data Fig. 1; new Extended Data Fig. 2). IdNSCs showed properties similar to brain NSCs in all these aspects. Furthermore, we quantified the *in vitro* differentiation efficiencies of IdNSCs into the three neural lineages and found that the differentiation ability of pNSCs was similar to that of brain control NSCs *in vitro* (new Fig. 2d). All these results demonstrate that IdNSCs are similar to brain NSCs.

Furthermore, in our revised manuscript, we also extensively characterized the NSC properties of pNSCs derived without low-pH treatment. The self-renewing capacity, NSC marker expression, genome-wide transcriptional profile, epigenetic features, and *in vitro* and *in vivo* differentiation ability of pNSCs were all similar to those of brain control NSCs (Response to Reviewer #1, paragraph 2 and 3; Response to Reviewer #3, point 4 and 6 below).

4. There is no quantification presented in relation to the data in Fig 2a-d. What is the relative proportion of different cell types after differentiation? What proportion of all plated cells differentiates into differentiated neural cell types (neurons, astrocytes, oligos)?

We have followed the Reviewer's suggestion and quantified the *in vitro* differentiation efficiencies of lDNSCs and brain NSCs into the three neural lineages. We found that the differentiation efficiencies of lDNSCs into the three neural lineages were similar to those of brain control NSCs (new Fig. 2d). There were around 88.0% brain NSCs, 87.7% eINSCs, 88.6% allNSCs that had differentiated into astrocytes during *in vitro* astrocyte differentiation. There were 3.0% brain NSCs, 2.9% lung pNSCs, and 2.7% tail pNSCs that had differentiated into oligodendrocytes during *in vitro* oligodendrocyte differentiation. There were around 11.7% brain NSCs, 11.8% eINSCs, and 10.9% allNSCs that had differentiated into neurons during *in vitro* neuronal differentiation. (Supplementary Table6).

Furthermore, we also showed the *in vitro* multipotent differentiation of Nes-GFP⁺ pNSCs (new Fig. 4b and 4c), Wnt1-Cre-mTmG Tomato⁺/GFP⁻ pNSCs (new Extended Data Fig. 5f and 5g), Sox1-GFP⁺ pNSCs (new Extended Data Fig. 9g and 9h), and Sox1-Cre-mTmG GFP⁺/Tomato⁻ pNSCs (new Extended Data Fig. 11f and 11g). The differentiation efficiencies of these pNSCs were similar to those of brain control NSCs. All these data demonstrate that the differentiation abilities of pNSCs are similar to that of brain control NSCs *in vitro*. We have also discussed this above (Response to Reviewer #1, point 3 and 4).

5. It is stated that allNSCs, after transplantation to cortex, “survived in the circuits of an *in vivo* neural network (Fig. 2g)”. From the presented data it is not possible to conclude that cells are integrated into the cortical neural network.

We agree with the Reviewer that our data only showed that the transplanted cells survived in the brain after transplantation, and we could not conclude from the presented data that the transplanted cells had integrated into the cortical neural network (new Fig. 2h and 2i). We rewrote the main text and deleted the sentence describing that transplanted cells had intergraded into the neural network: “Six weeks after transplantation, an overview of the grafts revealed that allNSCs and control NSCs had survived (Fig. 2h).” (page 5, line 159-160).

6. The data in Fig. 2h is very difficult to evaluate since it shows high-magnifications of individual cells without any quantification or overview that indicates how common these cells are in the recipient cortex. How many of the total transplanted mCherry⁺ cells form differentiated neural cells *in vivo*? What is the fate of other non-neural mCherry⁺ cells? Are

there evidence for proliferation? Also, how were cells labelled with mCherry (appears to be missing in Methods).

We thank the Reviewer for raising this point. We have followed the Reviewer's suggestion and repeated the transplantation experiments of pNSCs. Instead of IdNSCs, we repeated the transplantation experiment with clonal adult lung and postnatal tail pNSCs derived from Nes-GFP⁺ cells without low-pH treatment. The Nes-GFP⁺ lung pNSCs, tail pNSCs, and brain NSCs were labelled with CAG promoter-driven H2B-Tomato (new Extended Data Fig. 4a). Immunohistological staining results revealed that transplanted lung, tail pNSCs, and brain NSCs did not express the NSC marker Nestin and the cell-cycle marker Ki67, indicating that the pNSCs had lost their stem cell identity after transplantation and therefore can be used safely for cell transplantation approaches (new Extended Data Fig. 4b). Importantly, we observed that lung, tail pNSCs, and brain NSCs could differentiate into neurons, astrocytes, and oligodendrocytes *in vivo* (new Fig. 4d-f). The differentiation efficiencies of pNSCs into the three neural lineages were comparable to those of brain control NSCs (new Fig. 4g). There were around 5.3% brain NSCs, 5.1% lung pNSCs, and 4.8% tail pNSCs that had differentiated into neurons; around 68.0% brain NSCs, 68.7% lung pNSCs, and 66.4% tail pNSCs that had differentiated into astrocytes; and 22.4% brain NSCs, 20.9% lung pNSCs, and 22.5% tail pNSCs that had differentiated into oligodendrocytes during *in vivo* differentiation (Supplementary Table6). The new transplantation results demonstrate that most of the transplanted cells differentiated into the three neural lineage cells and pNSCs exhibit characteristics similar to brain control NSCs when transplanted into the mouse brain. We have also discussed this above (Response to Reviewer #1, point 4; Response to Reviewer #2, specific comments 5).

We apologize for this oversight regarding the description of cell labelling in the Methods. We have now added how we labelled the cells in the Methods (page 23, line 762-page 24, line 782).

7. Figure 3 is absolutely central to the conclusions of this paper and evidence needs to be rock solid for the far-reaching claim that is made in this study. The experiments are designed to show that pNSCs exist in peripheral adult tissues and that they are derived from embryonic neuroepithelial cells. The data in Fig. 3a-d shows that "pNSCs" can be derived without low-pH treatment, but unfortunately they do not provide evidence showing that such cells exist *in vivo*. Importantly, transgenic reporters based on Nestin-regulatory sequences can be leaky and may also be affected by dissection/dissociation procedures. Thus, it remains possible that they

are programmed into proliferating neural cells as a consequence of dissociation/plating, or they may be derived from peripheral NC-derived glial cells expressing the transgenic reporter (but not high levels of endogenous Nestin). Importantly, only low levels of leaky expression would be sufficient to give rise to GFP⁺ cells. In addition, concluding that IdNSCs are derived from endogenous pNSCs based on the data in Ext Fig 3e,f, is based on negative data (IdNSCs cannot be derived from GFP⁻ cells). Also this is not very convincing.

We thank the Reviewer for raising this key point. At the beginning, we did not know which marker could be used to trace pNSCs, as the existence of pNSCs outside of the CNS is a new finding. It is common to use Nestin to label and trace NSCs in the brain. Therefore, we also used the Nes-GFP transgenic mouse line to trace the pNSCs. However, our scRNA-seq data from Nes-GFP⁺ cells showed that the Nes-GFP transgene is not specific for labelling NSCs in the lung tissue. Based on the results of our colony formation assay, we knew that pNSCs existed in the Nes-GFP⁺ cells, but most Nes-GFP⁺ cells were not pNSCs (new Fig. 3c and 3f). As the Nes-GFP transgene is not specific for labelling pNSCs, we next tried to use the Sox2-GFP KI transgene to label and trace pNSCs. Our Sox2-GFP⁺ scRNA-seq data revealed that most Sox2-GFP⁺ cells were epithelial cells in the lung tissue, such as basal cells and club cells (new Fig. 6c; Extended Data Fig. 8i). Interestingly, we found that some Sox2⁺ cells also expressed Sox1. It is well known that NSCs and many other stem/progenitor cells express Sox2 (Ellis et al., 2004; Suh et al., 2007; Arnold et al., 2011), and Sox1 is the earliest known specific marker of NSCs in the mouse embryo (Wood and Episkopou, 1999; Aubert et al., 2003). These results made it particularly interesting for us to explore the cells coexpressing Sox1 and Sox2 in lung and tail tissues. Immunohistological staining results showed that Sox1⁺/Sox2⁺ cells indeed exist in lung and tail tissues. In postnatal mouse lung tissue, the Sox1⁺/Sox2⁺ cells were distributed mainly in the big bronchi epithelial cell wall (new Fig. 6e). This was also confirmed in adult mouse lung tissue (new Extended data Fig. 9a). In postnatal mouse tail tissue, Sox1⁺/Sox2⁺ cells were distributed under the skin as a short tube structure around the middle of the tail in longitudinal axis (new Fig. 6f). By using the Sox1-GFP KI transgenic mouse line (Aubert et al., 2003), we confirmed that pNSCs arise from Sox1⁺ cells (new Fig. 6g and 6h). Furthermore, our new scRNA-seq data of Sox1-GFP⁺ cells demonstrated that pNSCs *in situ* showed molecular features similar to brain NSCs, indicating the NSC identity of the endogenous pNSCs. All these new results provide evidence showing that pNSCs exist *in vivo*.

We have also provided evidence that pNSCs are distinct from NCSCs and do not originate from NCCs or their progeny cells (new Fig. 5; new Extended Data Fig. 5), ruling out the possibility that pNSCs have been reprogrammed from NC-derived glial cells. pNSCs could be derived only from Sox1-GFP⁺ cells, indicating that pNSCs arise from endogenous stem cells (new Fig. 6g and 6h; new Extended Data Fig. 9d-9h). Our scRNA-seq data on Sox1-GFP⁺ cells demonstrated that pNSCs *in situ* showed molecular features similar to brain NSCs but different from NC-derived Schwann cells, confirming the NSC identity of the endogenous pNSCs (new Fig. 7; new Extended Data Fig. 10d-10f). All these results have led us to conclude that the derivation of pNSCs is conceptually different from the process of reprogramming. We have also discussed this above (Response to Reviewer #1, paragraph 4; Response to Reviewer #2, paragraph 2).

To test whether the IdNSCs are derived from endogenous pNSCs or converted from somatic cells, we treated Sox1-GFP⁻ postnatal lung cells, which should not contain endogenous pNSCs, and Sox1-GFP⁺ postnatal lung cells with low-pH medium for 30 min and then cultured them in NSC medium for 3 weeks. We could observe NSC-like cell clusters from only Sox1-GFP⁺ cells but not from Sox1-GFP⁻ cells (new Extended Data Fig. 10g), indicating that IdNSCs are derived from endogenous Sox1⁺ pNSCs.

8. There are additional problems with the conclusions from the lineage tracing experiment in Fig. 3e-g. The authors have already shown that Nes-GFP⁺ cells can be derived from the adult lungs. Thus, using Nes-Cre to show that cells originate from the neural epithelium is not possible since Nes-Cre expression (just like Nes-GFP in the adult lung) would lead to recombination of the reporter. Showing the origin of these cells would require a much more extensive lineage tracing strategy with inclusion of appropriate controls. A strategy must demonstrate that recombination can only occur in the embryonic neuroepithelium e.g. by using a suitable CreER mouse line (e.g. NE-NesCreER), allowing for tamoxifen-induced recombination during embryogenesis. Using viral transfection of neuroepithelium in utero or intersectional genetic approaches to make the targeting more specific could also be useful. It would also be essential to follow labelled cells during development and careful controls need to assure that NC cells (or other non-neuroepithelial cells) are not lineage marked. Showing the absence of just one marker (HNK1) in *in vitro* cultured cells is far from sufficient for the conclusion that pNSC of non-NC origin exists in peripheral adult tissues (Fig 3g).

We thank the Reviewer for raising this point. We agree that Nestin-Cre is not specific for lineage-tracing of neuroepithelial cells. In our revised manuscript, we used new lineage tracing analysis with the Sox1-Cre-mTmG mouse line. In the Sox1-Cre-mTmG E8.5 embryos, GFP expression was detected exclusively in NECs in the trunk and caudal regions. In the cranial region, GFP expression was detected almost exclusively in NECs, except for very few GFP⁺ cells that had started to migrate out of the neural fold (new Fig. 8b; new Extended Data Fig. 11a and b). These results confirm that the NECs were efficiently and specifically labelled by GFP signal in the Sox1-Cre-mTmG mouse. Notably, pNSCs from Sox1-Cre-mTmG postnatal lung and tail tissues were GFP⁺/Tomato⁻ and could be derived from only GFP⁺ cells, indicating that pNSCs indeed originate from NECs (new Fig. 8c and 8d). In addition, we have provided evidence that pNSCs are different from NCSCs and do not originate from neural crest cells and their progeny cells (new Fig. 5; new Extended Data, Fig. 5). The scRNA-seq data showed that pNSCs *in situ* were different from Schwann cells or NC-derived cells (new Extended Data Fig. 10e and 10f). Taken together, our results provide evidence that pNSCs originate from Sox1⁺ NECs (new Fig. 8e). We have also discussed this above (Response to Reviewer #1, point 6; Response to Reviewer #2, specific comments 4).

9. The scRNA-seq data in Fig. 4 identifies clusters derived from GFP sorted cells from Nes-GFP mouse brain and lung. Several clusters in Fig 4a,c correspond to non-neural cell types that do not normally express Nestin including EC, SMC, Lym, Epi etc. What is the explanation for this? It seems to indicate that the Nes-GFP is leaky, or that the sorting is rather non-selective, a finding that has implications also for the conclusions in Fig. 3 (see above). How is Nestin mRNA expressed in the various clusters including the small ppNSC cluster? Showing expression in the clusters or in e.g. violin plots would be useful.

In the brain tissue, the Nes-GFP transgene was specific for labelling NSCs, as the majority of the cells were qNSCs, aNSCs, and NBs (new Fig.6a; Extended Data Fig. 6a-6d). However, in the lung tissue, the Nes-GFP transgene was not specific for labelling NSCs at all. Most Nes-GFP⁺ cells were non-neural lineage cells (new Fig. 6b; new Extended Data Fig. 6e and 6f). We think this is because the Nes-GFP transgene is not specific for labelling and tracing NSCs in the lung tissue. UMAP and violin plots showed that Nestin indeed was highly expressed in many types of cells (new Extended Data Fig. 7a and 7b). pNSCs existed in Nes-GFP⁺ cells, but most Nes-GFP⁺ cells were not pNSCs, as shown in the NSC colony forming analysis (new

Fig. 3c and 3f). In the revised manuscript, we used the Sox1-GFP KI transgene to label and trace the pNSCs.

10. Some NC markers were shown to be expressed in the ppNSC cluster and several of the “oligodendroglial” markers are normally expressed also in NC-derived Schwann cells and enteric glia (see e.g. mousebrain.org). In addition, Olig1 and Olig 2 were not expressed in this cluster. Does this not open up for the possibility that the peripheral “pNSCs” are of NC origin?

We thank the Reviewer for pointing out that several genes expressed in the putative pNSCs from Nes-GFP⁺ lung scRNA-seq data are also expressed in NC-derived Schwann cells and enteric glia. In the revised manuscript, we found that several NSC markers expressed in these cluster cells are also expressed in NC-derived cells, such as *ApoE*, *Fabp7*, and *Id3*. Many Schwann cell markers were also observed in this cluster cells (Extended Data Fig. 7e), indicating that this cell cluster could be mixture of different types of cells, including pNSCs and Schwann cells. We corrected this result in the main text and renamed this cluster as neural cells (page 10, line 319-324) (new Fig. 6b; Extended data Fig. 7a-7e).

As discussed above, we have demonstrated that pNSCs are not derived from NCCs and are different from NCSCs, which rules out the possibility that pNSCs have been reprogrammed from NC-derived glial cells (new Fig. 5; new Extended Data Fig. 5). pNSCs could be derived from only Sox1-GFP⁺ cells, indicating that pNSCs originate from endogenous stem cells (new Fig. 6g and 6h; new Extended Data Fig. 9d-9h). Our scRNA-seq data on Sox1-GFP⁺ cells demonstrated that pNSC *in situ* showed molecular features similar to brain NSCs and were different from Schwann cells, confirming the NSC identity of the endogenous pNSCs (new Fig. 7; new Extended Data Fig. 10d-10f). Based on the new lineage tracing analysis using the Sox1-Cre-mTmG mouse line, we provided evidence that pNSCs originate from NECs (new Fig. 8a-8e; new Extended Fig. 11). Taken together, our results have allowed us to rule out the possibility that pNSCs originated from NCs. We have also discussed this above (Response to Reviewer #1, paragraph 4; Response to Reviewer #2, paragraph 2).

11. It would be more informative to plot all cells from both brain and lung together to see how ppNSCs cluster in relation to NSCs and other cell types found in the brain. In fact, markers found for the ppNSCs appear to match more closely to glia than NSCs (see above). For

example, most markers that authors claim are for NSCs (lines 222-223) appear to be strongly expressed in enteric glia and other NC-derived cells according to mousebrain.org. In addition to mousebrain.org many other scRNA-seq data sets are now available and an integrated bioinformatic approach can go a long way to provide further information on the identity of ppNSCs.

We thank the Reviewer for this suggestion. In our revised manuscript, we used Sox1-GFP to label and trace the pNSCs (new Fig. 6d-6h; new Extended Data Fig. 9).

We have followed the Reviewer's suggestion and systematically analyzed the scRNA-seq data of Sox1-GFP⁺ tail and brain cells and compared the pNSCs to brain NSCs or Schwann cells. Notably, the majority of tail Sox1-GFP⁺ cells overlapped with brain qNSCs and aNSCs (new Fig. 7b and c). They expressed many brain NSC markers, such as *Sox2*, *Sox1*, *Sox9*, *Slc1a3*, *Pax6*, *Fabp7*, *Nnat*, *Hes1*, *Id3*, *Id2*, *Hopx*, *Ptprz1*, *Apoe*, *Dbi*, *Mt2*, *Tyh1*, *Aldoc*, and *Cspg5*, indicating the NSC property of these cells (new Fig. 7d; new Extended Data Fig. 10d). Examination of Gene Ontology (GO) enrichment of NSC cluster of cells showed that nervous system development and neurogenesis appeared in the top 5 GO terms (new Fig. 7e), confirming the NSC identity of these clusters. To assess whether Sox1-GFP⁺ pNSCs are similar to Schwann cells, we examined the marker expression in pNSCs and Schwann cells from the Sox2-GFP⁺ scRNA-seq dataset. We found that both pNSCs and Schwann cells highly expressed *Sox2*. pNSCs highly expressed the NSC-specific markers *Slc1a3*, *Ascl1*, *Pax6*, *Cspg5*, *Sox9* and *Sox1*, but not the Schwann cell/NC-derived cell markers *Sox10*, *Ngfr*, *Foxd3*, *MPZ*, *ErbB3*, *Mbp*, and *Plp1*. In contrast, Schwann cells highly expressed Schwann cell/NC-derived cell markers, but not NSC-specific markers (new Extended Data Fig. 10e and f), indicating that pNSCs are different from Schwann cells. Taken together, these results demonstrate that a large number of tail Sox1-GFP⁺ cells show molecular features similar to brain NSCs, indicating the NSC properties of these cells.

References:

- Achilleos, A, Trainor, P.A. Neural crest stem cells: discovery, properties and potential for therapy. *Cell Res.* **22**, 288-304 (2012).
- Arnold, K. et al. Sox2(+) adult stem and progenitor cells are important for tissue regeneration and survival of mice. *Cell Stem Cell* **9**, 317-329 (2011).
- Aubert, J. et al. Screening for mammalian neural genes via fluorescence-activated cell sorter purification of neural precursors from Sox1-gfp knock-in mice. *Proc. Natl. Acad. Sci. USA* **100**, 11836-11841 (2003).
- Binder, E., Rukavina, M., Hassani, H., Weber, M., Nakatani, H., Reiff, T., Parras, C., Taylor, V., Rohrer, H. Peripheral nervous system progenitors can be reprogrammed to produce myelinating oligodendrocytes and repair brain lesions. *J. Neurosci.* **31**, 6379-6391 (2011).
- Bixby S, Kruger GM, Mosher JT, Joseph NM, Morrison SJ. Cell-intrinsic differences between stem cells from different regions of the peripheral nervous system regulate the generation of neural diversity. *Neuron* **35**, 643-656 (2002).
- Conti, L., Pollard, S.M., Gorba, T., Reitano, E., Toselli, M., Biella, G., Sun, Y., Sanzone, S., Ying, Q.L., Cattaneo, E., Smith, A. Niche-independent symmetrical self-renewal of a mammalian tissue stem cell. *PLoS Biol.* **3**, e283 (2005).
- Danielian PS, Muccino D, Rowitch DH, Michael SK, McMahon AP. Modification of gene activity in mouse embryos in utero by a tamoxifen-inducible form of Cre recombinase. *Curr. Biol.* **8**, 1323-1326 (1998).
- Ellis, P., Fagan, B. M., Magness, S. T., Hutton, S., Taranova, O., Hayashi, S., McMahon, A., Rao, M., Pevny, L. SOX2, a persistent marker for multipotential neural stem cells derived from embryonic stem cells, the embryo or the adult. *Dev. Neurosci.* **26**, 148-165 (2004).
- Han, D. W. et al. Direct reprogramming of fibroblasts into neural stem cells by defined factors. *Cell Stem Cell* **10**, 465-472 (2012).
- Kruger, G. M., Mosher, J. T., Bixby, S., Joseph, N., Iwashita, T., Morrison, S. J. Neural crest stem cells persist in the adult gut but undergo changes in self-renewal, neuronal subtype potential, and factor responsiveness. *Neuron* **35**, 657-669 (2002).
- Luche H, Weber O, Nageswara Rao T, Blum C, Fehling HJ. Faithful activation of an extra-bright red fluorescent protein in "knock-in" Cre-reporter mice ideally suited for lineage tracing studies. *Eur. J. Immunol.* **37**, 43-53 (2007).
- Muzumdar, M. D., Tasic, B., Miyamichi, K., Li, L. and Luo, L. A global double-fluorescent Cre reporter mouse. *Genesis* **45**, 593-605 (2007).

Nagoshi, N. et al. Ontogeny and multipotency of neural crest-derived stem cells in mouse bone marrow, dorsal root ganglia, and whisker pad. *Cell Stem Cell* **2**, 392-403 (2008).

Suh, H., Consiglio, A., Ray, J., Sawai, T., D'Amour, K. A., Gage, F. H. In vivo fate analysis reveals the multipotent and self-renewal capacities of Sox2⁺ neural stem cells in the adult hippocampus. *Cell Stem Cell* **1**, 515-528 (2007).

Suzuki, J., Yoshizaki, K., Kobayashi, T., Osumi, N. Neural crest-derived horizontal basal cells as tissue stem cells in the adult olfactory epithelium. *Neurosci. Res.* **75**, 112-120 (2013).

Takashima, Y., Era, T., Nakao, K., Kondo, S., Kasuga, M., Smith, A. G., Nishikawa, S. Neuroepithelial cells supply an initial transient wave of MSC differentiation. *Cell* **129**, 1377-1388 (2007).

Vidal, M., Maniglier, M., Deboux, C., Bachelin, C., Zujovic, V., Baron-Van Evercooren A. Adult DRG Stem/Progenitor Cells Generate Pericytes in the Presence of Central Nervous System (CNS) Developmental Cues, and Schwann Cells in Response to CNS Demyelination. *Stem Cells* **33**, 2011-2024 (2015).

Weber, M., Apostolova, G., Widera, D., Mittelbronn, M., Dechant, G., Kaltschmidt, B., Rohrer, H. Alternative generation of CNS neural stem cells and PNS derivatives from neural crest-derived peripheral stem cells. *Stem Cells* **33**, 574-588 (2015).

Wood, H. B. and Episkopou, V. Comparative expression of the mouse Sox1, Sox2 and Sox3 genes from pre-gastrulation to early somite stages. *Mech. Dev.* **86**, 197-201 (1999).

Yu, T. S., Zhang, G., Liebl, D. J. and Kernie, S. G. Traumatic brain injury-induced hippocampal neurogenesis requires activation of early nestin-expressing progenitors. *J. Neurosci.* **28**, 12901-12912 (2008).

Response to Reviewers

We sincerely thank the Reviewers for their constructive comments and insightful suggestions. We have added a significant amount of new data, leading to substantial improvements in our lineage tracing conclusions and the scRNA-seq analysis. The changes of manuscript are highlighted in blue colour. We believe that the requested datasets enhance the paper and further strengthen our conclusions.

Please find below a point-to-point response to the Reviewers' comments.

Reviewers' comments:

Reviewer #2 (Remarks to the Author):

In the current resubmission of the manuscript, Han et al., have performed additional experiments to support their conclusion that neuroepithelium derived multipotent stem cells can be enriched from various parts of the murine peripheral (non-CNS) body. The revised manuscript now provides evidence for the following findings. First, low-pH derived NSCs (ldNSCs) can be obtained from embryonic limb/ adult lung with NSC-like properties. In vitro, they have the proliferative and differentiation potential similar to brain derived NSCs. These cells are multipotent, survive transplantation without resulting in tumors, differentiating into neuron, astrocytes and oligodendrocytes at a degree similar to brain derived NSCs. Hierarchical clustering and microarray data presented do show the similarity between brain derived NSCs and ldNSCs. Second, from peripheral tissues (lung, limb, postnatal tail) the authors do isolate multipotent peripheral NSCs (pNSCs) which can be maintained in culture and have differentiation similar to brain derived NSCs. The in vivo transplantation of pNSCs, obtained without low pH treatment, expressing Cre, H2B-GFP into R26-loxp-stop-loxp-RFP does provide evidence to exclude cell fusion. Third, the pNSCs do not express NC markers in vitro and pNSCs isolated from postnatal lung from Sox1-GFP KI animals (and identified as Sox1-GFP⁺ cells) behave as control brain derived NSCs. Despite these new results, the concerns listed below need to be addressed.

Concerns

1. In Figure 7 the authors performed scRNA-seq of Sox1-GFP cells from the brain and the postnatal tail and argue that NSCs from both tissues overlap. This is a significant concern, not just for the current study but an inference which can be drawn is that cellular quiescence is similar irrespective of the niche and microenvironment, which is not the case- even for qNSCs from different brain regions. The authors attribute other cells populations to cytometry noise. For comparison between cells from the brain and the postnatal tail, these cells have to be excluded from Figure 7C. Their presence in the analysis, especially immune cells, will lead to a false sense of clustering (The "NSCs" do not really overlap either). Furthermore, immune cells in the tail would not be microglia; macrophages, at best. To succinctly show similarities, the authors need to show an UMAP just with the "NSCs". Next the authors need to perform far more detailed pair wise comparisons, along with hypergeometric tests, between the "NSC" populations to show greater than expected overlap (please present venn diagrams).

Answer: We have followed the Reviewer's suggestion and reanalyzed the scRNA-seq data after excluding noise cells. First, we removed ependymal cells, microglia/macrophages, and fibroblasts from the dataset and re-performed the analysis. We then excluded Krt1/Krt10 double-positive lung epithelial cells and Olig2/Plp1 double-positive OPC/Oligodendrocytes.

Finally, we used only the “NSC” population for cluster analysis. Our findings show that the tail pNSCs still clustered together with brain NSCs, indicating that tail pNSCs are indeed similar to brain NSCs (new Fig. 7f and 7g).

To further examine the similarity between tail pNSCs and brain NSCs, we conducted more detailed comparisons as suggested by the Reviewer. Pairwise-comparison analysis showed that the Pearson correlation coefficients of qNSC4, aNSC1, and aNSC2 clusters between brain NSCs and tail pNSCs were quite high (over 0.9), indicating significant similarity between these pNSC cluster cells and brain NSCs. The Pearson correlation coefficients of qNSC1, qNSC2, and qNSC3 clusters between brain NSCs and tail pNSCs were 0.63, 0.73, and 0.46, respectively (new Fig. 7h), indicating that these cells in the tail only partially resemble brain NSCs, perhaps suggesting a niche-dependent cellular quiescence.

In addition, the Venn diagram of conserved marker genes in each cluster between tail pNSCs and brain NSCs showed that while pNSCs shared many conserved markers with brain NSCs, there were still many brain NSC markers that were not expressed in tail pNSCs (new Extended Data Fig. 11d). This demonstrates that pNSCs are not exactly same as brain NSCs, which is expected, as these cells most likely follow different developmental trajectories and interact with different microenvironments.

Taken together, our new data shows that tail pNSCs cluster with and share many conserved markers with brain NSCs, suggesting that they are similar. However, pairwise comparison and marker gene Venn diagram analysis show that tail pNSCs are not identical to brain NSCs. We have replaced the word “overlapped” with “were clustered together” for better interpretation of the results, and added these findings to the main text (line 466-492).

We appreciate the Reviewer’s suggestion regarding the naming of immune cells in the tail. We have renamed these cells “macrophages,” as they would not be microglia.

It is not clear from the methods how identities were attributed to cell clusters, specially the cells from the tail.

Answer: We performed scRNA-seq experiments on postnatal tail and adult brain Sox1-GFP⁺ cells, and obtained data from 2,469 and 12,111 cells, respectively. All Sox1-GFP⁺ cells from both brain and tail were pooled together and separated into 9 different cell type clusters on the UMAP plot using the Seurat package. Based on the expression of some well-characterized marker genes, we attributed cell identity to each cluster. Abundant cell populations expressed many known NSC or intermediate neural progenitor (IPC) markers and were identified as qNSC1 (e.g. *Sparc*, *Aqp4*, *Apoe*, *Ednrb*, *Atp1a2*, *Apoe*, *Bcan*, *Slc1a3*, *Clu*, *Aldoc*), qNSC2 (e.g. *Tspan7*, *Mfge8*, *Clu*, *Slc1a3*, *Aldoc*, *Ptn*, *Bcan*, *Apoe*, *Ptprz1*, *Hopx*), qNSC3 (e.g. *Hopx*, *Fabp7*, *Vim*, *Dbi*, *Slc1a3*, *Fzd1*, *Tnc*, *Aldoc*, *Ednrb*, *Ptprz1*, *Mt1*, *Mt2*), qNSC4 (e.g. *Bcan*, *Atp1a2*, *Mt1*, *Neat1*, *Gfap*, *Mt3*), aNSC1 (e.g. *Hmgb2*, *Ptma*, *Hmgn2*, *Sox11*, *Eef1a1*, *Miat*, *Lima*, *Pabpc1*, *Rpl12*, *Rpsa*) and aNSC2 (e.g. *Baspl*, *Stmn2*, *Dcx*, *Fos*, *Sox11*, *Eef1a1*, *Ptma*, *Egr1*, *Miat*, *Egr1*, *Rpsa*, *Rpl12*, *Pabpc1*). Other very minor populations were identified as ependymal cells (Epend) (expressing e.g. *Tmem212*, *Hdc*, *Cdhr3*, *Fam216b*, *Odf3b*, *Ccdc153*, *Foxj1*, *Pifo*, *Dynlrb2*, *Pcp4ll*), fibroblasts (Fibro) (expressing e.g. *Col3a1*, *Col6a2*, *Col6a3*, *Pdgfra*, *MFAP5*, *Colla2*, *Colla1*, *Postn*, *Aspn*, *Penk*, *Col5a1*) and Microglia/Macrophages (Macro) (expressing e.g. *Ctss*, *Csf1r*, *Laptm5*, *Cd53*, *Itgam*, *Cyth4*, *Gpx1*, *Alox5ap*, *Fcrls*, *Clqc*). We believe that this may be due to flow cytometry noise or nonspecific sorting (Fig. 7a; Extended Data Fig. 10a-c; Supplementary Table4). These results have been added to the main text (line 419-435).

In the same figure, 7d panel is problematic. Sox2, even at the transcript level, is well published to be downregulated in brain neuroblasts. However, there is no observable difference between the brain clusters. This problem is far more exaggerated for Sox1. Given that the cells are obtained from Sox1-GFP knockin line, one would expect far more NSCs, rather than neuroblasts to be Sox1 positive.

Answer: We thank the Reviewer for raising this point. We agree that Sox2 expression is downregulated in brain neuroblasts. After re-examining Sox2 expression levels in different cluster cells (after removing the noise cells), we found that Sox2 expression in the labelled “NB” cluster was not downregulated (Figure below), indicating that our previous “NB” cluster designation may not be correct.

Upon revisiting our “NB” cluster, we found that these cells expressed some NB markers, such as *Stmn2*, *Dcx*, *Nrep*, *Cd24a*. However, they also expressed many more aNSC markers, such as *Baspl*, *Fos*, *Sox11*, *Eef1a1*, *Ptma*, *Dlx1*, *Dlx2*, *Egr1*, *Miat*, *Lima1*, *Egr1*, *Rpsa*, *Rpl12*, *Pabpc1*, *Rpl41*, *Ppia*, and *Hmgn2* (Supplementary Table4). In addition, *Stmn2* and *Dcx* are known to be expressed in both neuroblasts and neural progenitor cells (Artegiani et al., 2017; Basak et al., 2017; Yuzwa et al., 2017).

Pairwise-comparison analysis showed that the Pearson correlation coefficients between brain aNSCs and the previous brain “NB” cluster, as well as tail aNSCs and the previous tail “NB” cluster were 0.94 and 0.96, respectively (new Fig. 7h), indicating that these cells are very similar. The GO analysis of the previous “NB” cluster cells showed similar GO terms to aNSC cluster cells (New Fig. 7e). These results indicate that our previous “NB” cluster cells are actually aNSCs rather than mature neuroblasts. We renamed this cluster cells as aNSC2 (new Fig.7).

Therefore, most Sox1-GFP–positive cells from the Sox1-GFP knockin mouse line are NSCs rather than neuroblasts, consistent with previous reports that Sox1 marks neural stem cells/progenitor cells in the mouse brain (Barraud et al., 2005; Venere et al., 2012).

Figure: Violin plots showing the expression of NSC marker gene Sox2 in different populations of brain and tail Sox1-GFP⁺ cells.

2. The authors elude to a full-fledged neurogenic niche within the postnatal tail (with qNSCs, aNSCs and NBs) (Figure 7f). From the numbers, ~450 NBs (18%) were sequenced. While the authors show some very intriguing Sox1+Ki67+ cells within the tail, it is pertinent to perform staining for other NB markers (DCX, Calretinin, PSA-NCAM). Given the magnitude of the claim, a sufficient explanation has to be provided regarding the final fate of these pNSC derived NBs. It is recommended to follow this niche into adulthood.

Answer: From the above answers to point 1, the new results showed that our previous designation of the “NB” cluster was incorrect; these cells are active pNSCs, not neuroblasts. Therefore, our scRNA-seq analysis did not reveal a full-fledged neurogenic niche (with qNSCs-aNSCs-NBs) within the postnatal tail.

As suggested by the Reviewer, to examine the neurogenic niche within different tissues during embryogenesis and postnatal development, including the postnatal tail, we crossed Sox1-CreERT2 mice with R26-EYFP mice. Tamoxifen was injected into pregnant mothers at E8.5, E13.5, and E16.5, and the mothers were sacrificed at E19.5 for analysis. Newborn pups were also injected with tamoxifen and sacrificed one week later to study postnatal development. The inducible expression of Cre allows for the time-coordinated expression of EYFP and selective labelling of pNSCs and their progenies at the respective developmental stages (new Fig. 8g).

After sacrifice, multiple tissues were harvested, including the heart, thymus, lung, intestine, DRG, and tail. Labelled progenies were co-stained with the neuronal marker Tuj1 and functional neuronal markers, such as Synapsin 1 (Syn1) for functional synapse, GABA for sensory neurons, choline acetyltransferase (ChAT) for cholinergic/motor neurons, and tyrosine hydroxylase (TH) for peripheral sympathetic and enteric neurons.

In some tissues, such as the heart, a significant contribution of pNSC progenies was observed only in earlier time points (tamoxifen injection at E8.5 and E13.5), as indicated by EYFP⁺/Tuj1⁺/Syn1⁺, or EYFP⁺/Tuj1⁺/GABA⁺ neurons. Such cells were rarely observed when tamoxifen was injected at E16.5 or postnatally (new Extended Data Fig. 15b), suggesting that the deposited pNSCs have completed their differentiation and contribution to heart development at early organogenesis. Similarly, EYFP⁺/Tuj1⁺/Syn1⁺, EYFP⁺/Tuj1⁺/GABA⁺, and EYFP⁺/Tuj1⁺/TH⁺ neurons were found in lung tissue until tamoxifen was injected at E16.5, but such cells were rarely observed when tamoxifen was injected postnatally (new Extended Data Fig. 16b).

However, in other tissues, the contribution of pNSC progenies to neurogenesis persisted even after birth. In particular, pNSCs contributed significantly to the neurogenesis of the DRG (new Fig. 8h; new Extended Data Fig. 16a), accounting for a major proportion of developing neurons. This large-scale contribution continued into postnatal development, though it somewhat abated after E16.5 (new Fig. 8h; new Extended Data Fig. 16a). Similarly, significant contributions of pNSCs to neurogenesis were observed in the intestine and thymus across all time points assayed, indicating an important role of pNSCs in organ development (new Fig. 8h; new Extended Data Fig. 15a and c). Importantly, the EYFP⁺ neurons were also positive for functional neuronal markers synapsin 1, GABA, ChAT, and TH, suggesting that pNSCs have a functional contribution in the endogenous environment (Fig. 8h; Extended Data Fig. 15; Extended Data Fig. 16).

We did not observe any EYFP⁺ cells in peripheral DRG and intestine tissues, which should contain a very high percentage pNSCs in their progenies, in mice with the same genotype but

without tamoxifen injection (new Extended Data Fig. 15d). This excludes the possibility of weak leaky expression of the transgenic KI-construct, which could have led to sufficient Cre for recombination even in cells where Sox1 is not detected. In the postnatal tail, although the neurogenesis of pNSCs was much less compared to the aforementioned tissues, we did observe functional neurogenesis from pNSCs after birth, demonstrating the neurogenic niche of pNSCs in postnatal tail (new Extended Data Fig. 17).

In conclusion, our data demonstrates that pNSCs contribute to functional neurons that aid in the neurogenesis of multiple organs. pNSC neurogenesis follows different developmental trajectories for each tissue, persisting after birth for some or completing during early organogenesis for others. This continuous and intensive neurogenesis during fetal and postnatal development indicates both the intrinsic NSC properties of pNSCs and the significant functional impact of pNSCs on PNS development. We have added these results to the main text (line 573-623).

We thank the Reviewer for suggesting the follow-up study on pNSC neurogenesis into adulthood. However, our main conclusion that pNSCs with CNS-NSC properties exist in peripheral tissues remains valid without this additional investigation. We intend to address this question in future research.

3. An alternate explanation for pNSCs is de/trans-differentiation (a case can be made for either phenomena). The authors clearly aim to address this in Figure 8. Using constitutive Sox1 driven Cre does explain the observed GFP (mTmG) detection in the brain during embryonic development. However, an inducible version of Sox1-Cre had to be employed to rule out a late onset of Sox1 expression in the peripheral tissue, explaining GFP+/Tomato-cells in peripheral tissue, which negates a neuroepithelial origin. It is advised to do a far more stringent time course, if the current line is used.

Answer: We thank the Reviewer for raising this point. We have followed the Reviewer's suggestion and employed an inducible Sox1-CreERT2 mouse to rule out a late onset of Sox1 expression in peripheral tissues. We crossed Sox1-CreERT2 mice with R26-EYFP mice and injected tamoxifen only once at E8.5 into pregnant female mice. At E19, the pregnant mice were sacrificed, and the Sox1-CreERT2-R26-EYFP positive pups' tail tissues were dissociated into single cells. All the tail cells were cultured in NSC medium to derive tail pNSCs. Indeed, we were able to derive EYFP⁺ tail pNSCs.

qPCR and immunostaining analysis of NSC markers indicated a similar gene profile of the EYFP⁺ pNSCs colonies to brain NSC colonies (new Fig. 8f; new Extended Data Fig. 13a-c). Since we injected tamoxifen into pregnant female mice only once at E8.5, only the earliest neural tube neuroepithelial cells can be labelled, indicating that these tail pNSCs are derived from neural tube neuroepithelial cells. Thus, we exclude the possibility of a late onset of Sox1 expression in peripheral tissues.

We also observed many EYFP⁻ tail pNSC clusters from the same experiments (new Extended Data Fig. 13d), likely due to low Cre-loxP recombination efficiency resulting from the relatively low tamoxifen dosage injected into the pregnant female mice. Similarly, both EYFP⁺ and EYFP⁻ brain NSCs were observed (new Extended Data Fig. 13e). We have added these results to the manuscript (line 534-551).

4. Quiescence is not a singular state, but graded. The authors interpret that low recovery of colonies from pNSCs is indicative of their quiescence. While the authors perform low-pH treatment from Sox1-GFP[±] cells, it would be pertinent to check if the neurosphere/cluster

formation from GFP+ fractions from various tissues increases with the treatment, and compare this treating control brain NSCs with low-pH (which was not done in any experiments). This experiment would also try to shed light on why so few pNSCs, at least from the postnatal tail, form colonies.

Answer: We agree with the Reviewer that quiescence is not a singular state but rather graded. The colony-forming efficiency of Sox1-GFP+ tail cells is about 9.3%. However, the colony-forming efficiency of brain Sox1-GFP+ cells is much higher, suggesting that brain NSCs are easier to activate and more prone to proliferate than pNSCs (new Extended data Fig. 11f). These data suggest a graded quiescence, which is consistent with the reviewer's opinion.

Following the Reviewer's suggestion, we performed low-pH treatment on tail and brain Sox1-GFP+ cells. Low-pH treatment increased the efficiency of tail pNSC and brain NSC derivation (new Extended data Fig. 11f), further confirming the graded quiescence of pNSCs. We have added these results to the manuscript (line 503-511).

5. The observation that DRG houses pNSCs is intriguing. The authors do not show the FACS plot of DRG isolated cells for Tomato/GFP levels. A closer look at the image in Figure 5e shows some GFP+ staining. What is the proportion of Tomato+/GFP- cells? The expectation is that GFP+ cells would outnumber the GFP- cells. What fraction of the Tomato+GFP- cells form clusters? If GFP+ cells can be recovered from stringent Tomato+ gating, does this mean that a minority of the population cells start to express Wnt1 (and p75)? Can low-pH treatment of the GFP+ cells from DRG induce p75- "pNSC-like" cultures? If they don't, that could argue against trans/de-differentiation. Can you recover pNSCs from the Sox1-GFP mice's DRGs?

Answer: When we used DRG cells for the experiment, we did not perform a FACS experiment to sort the GFP+ cells out. Because we did not know whether pNSCs were derived from NC cells, we simply plated all the cells from the DRG tissues and cultured them in NSC medium. The cells included GFP+ and GFP- cells. Therefore, we could see some GFP+ cells in Figure 5e, but these GFP+ cells did not form NSC colonies and were lost within several passages.

We agree with the Reviewer that Wnt1-Cre might not 100% label all the NC cells, but the vast majority of NC cells are labelled by GFP. If the pNSCs were derived from NC cells, we should have a higher chance of finding the GFP+ NSC clusters. However, we observed only Tomato+/GFP- NSC cluster cells. This suggests that pNSCs do not originate from NC cells. We have added these points to the main text (line 271-295).

We also followed the Reviewer's suggestion and performed low-pH treatment on DRG GFP+ cells from Wnt1-Cre-R26-mTmG mice. We did not observe pNSC cluster cells from GFP+ cells (new Extended Data Fig. 5b), further arguing against trans/de-differentiation.

As suggested by the Reviewer, we checked whether we could recover pNSCs from the DRGs of Sox1-GFP mice by sorting the GFP+ cells from DRG tissue and culturing them in NSC medium. We were able to recover DRG pNSCs from the DRGs of Sox1-GFP mice (new Extended Data Fig. 9c-i).

6. In many experiments, the authors report that that sorted GFP+ cells were cultured on monolayers in NSC medium and clusters were picked. How does one isolate and pick adhered monolayer cells from the surrounding cells? (Specifically refer line 191-192; methods line 611-612) Please explain this more clearly in the methods.

Answer: As observed from the pictures of the pNSCs, the pNSC clusters were usually clonal due to the low colony formation efficiency. Consequently, we were able to pick each pNSC cluster with pipette tips. We made every effort to extract only the pNSC cluster cells by aspirating the central cells of the NSC clusters. While we could not 100% guarantee that we were sampling only pNSC cells without touching surrounding cells, the few surrounding cells that were inadvertently included were minimal and would be lost in multiple passages. We have provided a more detailed explanation of this process in the methods section (line 749-755).

7. The supplementary table with the raw results is useful. It is advised that the authors provide the raw numbers from the transplantation experiments employed in the manuscript.
Answer: We have followed the Reviewer's suggestion and provided the raw numbers from the transplantation experiments in the manuscript (line 234-237).

8. Please provide references for p75 (line 255-256)

Answer: We have followed the Reviewer's suggestion and provided the reference for p75 (Stemple et al., 1992; Bixby et al., 2002) (line 260).

9. In the experiment detailed in lines 123-127, the authors checked the methylation status for 2 genes- Colla1 and Nestin. To say that IdNSCs and brain derived NSCs are similar at the epigenetic level is a huge overreach.

Answer: We have followed the Reviewer's suggestion and removed the sentence "suggesting that IdNSCs are similar to control NSCs also at the epigenetic level" (line 129) for better interpretation of the results.

Reviewer #3 (Remarks to the Author):

The revised manuscript by Han et al. has been extensively revised and improved. Most of my previous technical concerns have been addressed, and the conclusion that multipotent cells with NSC-properties can be isolated from the lung and tail is supported by the data.

An important critique in my previous review was that the authors did not accurately distinguish the tail and lung NSCs from neural crest derived stem cells (NCSCs). One of the approaches was to use a Wnt-Cre transgenic mouse line to mark cells of neural crest origin. However, the data is based on lack of Wnt-Cre-induced floxing in neuroepithelial-derived NSCs, but not in neural crest-derived cells. However, it is likely that not 100% of NC-derived cells are accurately floxed and that a small proportion NC-derived cells can escape floxing and give rise to the misleading data. This is a common weakness in many lineage-tracing studies, e.g. emphasized in many of the strongly criticized astrocyte-to-neuron reprogramming studies that have been published in recent years. It is important that the authors explain the potential problems in interpretation of their lineage-tracing data. However, the conclusion that cultured lung and tail stem cells have properties distinct of NCSCs is now convincingly supported, not least by the inclusion of more solid scRNAseq data.

Answer 1: We used Wnt1-Cre-mTmG to lineage trace the NCCs and their daughter cells. When we performed experiments to obtain pNSCs from postnatal Wnt1-Cre-mTmG tail or DRG tissues (Figures 5d and 5e), we did not perform a FACS experiment to sort the GFP⁻ or GFP⁺ cells because we were uncertain whether the pNSCs are derived from neural crest (NC) cells. We simply plated all the cells from tail or DRG tissue at low density and cultured them in NSC medium. Given that there are many NC-derived cells in the DRG tissue, we observed some GFP⁺ cells in the culture. However, these GFP⁺ cells did not form NSC-like clusters (Fig. 5e) and were lost within several passages. Under these conditions, the GFP⁺ and GFP⁻ cells had the same chance of forming NSC-like clusters. Although Wnt1-Cre may not label all NC-derived cells 100%, most NC-derived cells are indeed labelled with GFP. If the pNSCs were derived from NC cells, we would expect to observe GFP⁺ NSC-like clusters rather than GFP⁻ NSC-like clusters, as most NC-derived cells are GFP⁺. In fact, we observed only GFP⁻ NSC-like clusters and never GFP⁺ NSC-like clusters, strongly suggesting that pNSCs are not derived from NC-derived cells. We have added these results to the main text (line 271-295).

We thank the Reviewer for recognizing that our cultured lung and tail neural stem cells have properties distinct from NCSCs.

Also, the new scRNAseq data derived from Sox1-GFP⁺ cells from the tail and compared to Sox1-GFP⁺ cells from the embryonic brain is an important new addition that strengthens the manuscript.

Answer 2: We thank the Reviewer for recognizing our new scRNA-seq data.

The authors also conclude that Sox1⁺ cells in the embryonic epithelium gave rise to the peripheral NSCs cultured in vitro and the cells with NSC-properties observed in tissue sections. However, as eluded to above, lineage tracing approaches are often leading to incorrect conclusions. The data in Fig. 8e-g can be explained either by Cre-expression in peripheral tissues leading to Cre-induced recombination outside of the neuroepithelium. Such expression could be a reflection of “real” Sox1 expression or it could be a result of weak expression of the transgenic KI-construct leading to sufficient Cre for recombination even in cells where Sox1 is not detected. Also, since this is not a tamoxifen-controlled Cre,

expression can lead to recombination at post-embryonic stages, e.g in the adult tissues. Thus, the evidence supporting the existence of neuroepithelial-derived quiescent NSCs in the periphery is rather weak. More importantly, what is the significance of such cells and are the neurons derived from them functional and of physiological importance?

Answer 3: We agree with the Reviewers that a tamoxifen-controlled inducible Cre system would provide stronger evidence for the existence of neuroepithelia-derived pNSCs in peripheral tissues. Following the Reviewer's suggestion, we crossed Sox1-CreERT2 mice with R26-EYFP mice and administered tamoxifen only once at E8.5 to pregnant female mice. At E19, the pregnant mice were sacrificed, and the Sox1-CreERT2-R26-EYFP positive pups' tail tissues were dissociated into single cells. We cultured all tail cells in NSC medium to derive tail pNSCs. We successfully derived EYFP⁺ tail pNSCs (Fig. 8f; Extended Data Fig. 13a-c). Since tamoxifen was administered only once, this labelling affected only the earliest neural tube neuroepithelial cells, indicating that these tail pNSCs are derived from neural tube neuroepithelial cells. Thus, we exclude the possibility of a late onset of Sox1 expression in peripheral tissues.

We also observed many EYFP⁻ tail pNSC clusters from the same experiments (Extended Data Fig. 13d) due to relatively low efficiency of Cre-loxP recombination, which was likely caused by the low dosage of tamoxifen injected into the pregnant female mice. Similarly, EYFP⁺ and EYFP⁻ brain NSCs were observed (Extended Data Fig. 13e). Additionally, we did not observe any EYFP⁺ cells in peripheral DRG and intestine tissues from mice of the same genotype without tamoxifen injection (Extended Data Fig. 15d). This finding excludes the possibility of weak leaky expression of the transgenic KI-construct leading to sufficient Cre for recombination in cells where Sox1 is not detected. We have added these results to the manuscript (line 534-551; line 607-611).

As suggested by the Reviewer, to address the significance of pNSCs and whether the neurons derived from them are functional and of physiological importance, we crossed Sox1-CreERT2 mice with R26-EYFP mice. Tamoxifen was injected into pregnant mothers at E8.5, E13.5, and E16.5, and the mothers were sacrificed at E19.5 for analysis. Newborn pups were also injected with tamoxifen and sacrificed one week later to study postnatal development. The inducible expression of Cre allows for the time-coordinated expression of EYFP and selective labelling of pNSCs and their progenies at the respective developmental stages (Fig. 8g).

Following sacrifice, multiple tissues were harvested, including the heart, thymus, lung, intestine, DRG, and tail. Labelled progenies were co-stained with the neuronal marker Tuj1 and functional neuronal markers, such as Synapsin 1 (Syn1) for functional synapses, GABA for sensory neurons, choline acetyltransferase (ChAT) for cholinergic/motor neurons, and tyrosine hydroxylase (TH) for peripheral sympathetic and enteric neurons. In some tissues, such as the heart, significant contributions of pNSCs were observed only in earlier time points (tamoxifen injection at E8.5 and E13.5), as indicated by EYFP⁺/Tuj1⁺/Syn1⁺, and EYFP⁺/Tuj1⁺/GABA⁺ neurons. Such cells were rarely observed when tamoxifen was injected at E16.5 or postnatally (new Extended Data Fig. 15b), suggesting that the deposited pNSCs have completed their differentiation and contribution to heart development during early organogenesis. Similarly, EYFP⁺/Tuj1⁺/Syn1⁺, EYFP⁺/Tuj1⁺/GABA⁺, and EYFP⁺/Tuj1⁺/TH⁺ neurons were found in lung tissue up to tamoxifen injection at E16.5; these cells were rarely observed when tamoxifen was injected postnatally (new Extended Data Fig. 16b).

In contrast, in other tissues, the contribution of pNSCs to neurogenesis persisted even after birth. Notably, pNSCs contributed significantly to neurogenesis in the DRG (new Fig. 8h; new Extended Data Fig. 16a), accounting for a major proportion of developing neurons. This large-scale contribution continued into postnatal development, though it somewhat abated after E16.5 (new Fig. 8h; new Extended Data Fig. 16a). Similarly, significant contributions of pNSCs to neurogenesis were observed in the intestine and thymus across all time points assayed, indicating a crucial role of pNSCs in organ development (new Fig. 8h; new Extended Data Fig. 15a and c). Importantly, the EYFP⁺ neurons were also positive for functional neuronal markers synapsin 1, GABA, ChAT, and TH, suggesting a functional contribution of pNSCs in the endogenous environment (new Fig. 8h; new Extended Data Fig. 15; new Extended Data Fig. 16).

In the postnatal tail, although the neurogenesis of pNSCs was less pronounced compared to other tissues, we still observed functional neurogenesis from pNSCs after birth, demonstrating the neurogenic niche of pNSCs in the postnatal tail (new Extended Data Fig. 17).

In conclusion, our data demonstrates that pNSCs contribute to functional neurons and play a role in the neurogenesis of multiple organs. pNSC neurogenesis follows different developmental trajectories for each tissue, persisting after birth for some or completing during early organogenesis for others. Such continuous and extensive neurogenesis from pNSCs during fetal and postnatal development, particularly the high neuronal contribution of pNSC progenies in DRG and intestine tissues (new Fig. 8h; new Extended Data Fig. 16a; new Extended Data Fig. 15a), highlights the intrinsic NSC properties of pNSCs and the functional impact of pNSCs on PNS development. We have added these results to the manuscript (line 573-623).

Ideally, specific deletion of pNSCs would be the best approach to assess their physiological importance. However, it is challenging to selectively delete pNSCs and their progenies without affecting CNS cells. Identifying specific markers that exclusively label pNSCs but not CNS NSCs will be valuable for a more detailed investigation into the developmental importance of pNSCs in the future. However, we believe that the main conclusion of the manuscript—that neuroepithelial-derived pNSCs with CNS NSC properties exist outside of CNS—is strongly supported by our data.

Comments on authors responses to referee 1:

Comments from referee 1 resembles many of my own concerns (referee 3). The authors have responded to all comments from referee 1 and they have also added new data that strengthens the claims.

Some comments:

- Point 2. The authors refer to the Wnt1-Cre tracing experiment which provides support for the conclusion that the analyzed cells were not derived from NCSCs. It should be noted, and pointed out in the ms, that this is a conclusion that is based on negative data. Thus, Wnt1-Cre labels NCCs and the lack of labeling is used to conclude that that cells are non-NCCs (see comment also to referee 3).

Answer: Please see the answer 1 for Reviewer 3.

- Point 5. Another lineage tracing approach used Sox1-Cre to label cells originating from the neuroepithelium. See comment from referee 3 concerning this approach. It is important to consider the many problematic studies using reprogramming approaches to make far-reaching claims that astrocytes and other cell types can be reprogrammed into therapeutically desired cell types in vivo. Many of these papers have not been possible to repeat and major problems were due to lineage tracing approaches that are not solid. In this manuscript, additional supporting data (e.g. scRNA-seq) provides independent strength to the conclusions, but cautionary comments in the ms on the tracing strategies would be important.

- The problem pointed out above is partly addressed by the scRNA-seq data from Sox1-Cre labeled cells. These data provide a strong addition addressing problems with the original conclusions.

Answer: Please see the answer 3 for Reviewer 3.

- I believe that most other specific points raised by Referee 1 have been addressed by the addition of new experimental data.

Answer: We thank Reviewer 3 for their efforts in reviewing our manuscript.

References:

- Artegiani, B. et al. A single-cell RNA sequencing study reveals cellular and molecular dynamics of the hippocampal neurogenic niche. *Cell Rep.* 21, 3271–3284 (2017).
- Basak, O. et al. Troy+ brain stem cells cycle through quiescence and regulate their number by sensing niche occupancy. *Proc. Natl. Acad. Sci.* 115, E610–E619 (2018).
- Yuzwa, S. A., Borrett, M. J., Innes, B. T., Voronova, A., Ketela, T., Kaplan, D., Bader, G.D., Miller, F.D. Developmental emergence of adult neural stem cells as revealed by single-cell transcriptional profiling. *Cell Rep.* 21, 3970–3986 (2017).
- Barraud, P., Thompson, L., Kirik, D., Björklund, A., Parmar, M. Isolation and characterization of neural precursor cells from the Sox1-GFP reporter mouse. *Eur. J. Neurosci.* 22, 1555–1569 (2005).
- Venere, M., Han, Y. G., Bell R., Song, J. S., Alvarez-Buylla, A., Belloch, R. Sox1 marks an activated neural stem/progenitor cell in the hippocampus. *Development* 139, 3938–3949 (2012).
- Stemple, D. L. & Anderson D. J. Isolation of a stem cell for neurons and glia from the mammalian neural crest. *Cell* 71, 937–985 (1992).
- Bixby, S., Kruger, G. M., Mosher, J. T., Joseph, N. M., Morrison, S. J. Cell-Intrinsic differences between stem cells from different regions of the peripheral nervous system regulate the generation of neural diversity. *Neuron* 35, 643–656 (2002).

Response to Reviewers

We sincerely thank the Reviewers for recognizing our revised manuscript as improved. We also thank the Reviewers for their new comments to make our manuscript even stronger. In the following, we address the Reviewers' concerns and explain points that support our claim.

REVIEWER #2

In the revised manuscript the authors have performed extensive experiments to address the previous concerns. The presented work shows that Sox1-expressing cells exist outside the CNS with properties similar to brain derived NSCs- in terms of colony formation, differentiation potential and marker expression. These cells, resident in many tissues, at the very least, seem to retain Sox1 expression and seemingly be able to differentiate into neurons (at different developmental speeds). The authors do provide compelling evidence using a single Tamoxifen injection strategy that the “pNSCs” identified are most likely from the neural tube early in the brain development. While the manuscript is improved, there are several remaining some points to be addressed:

1. A overall caveat is that these pNSCs, although possess stem like properties, may represent biological “noise”- a few cells escaping canonical developmental steps, ending up in the periphery. These cells do have the remarkable ability to generate terminally differentiated cells, which express makers which might suggest a neuron-like function. These cells have similar expression profile to brain NSCs. But it has to be recognized that the brain NSCs are an aggregate of NSCs from both the SGZ and SVZ. Hence the similarity could be driven, in part from the fact these are stem-like cells. To overcome this concern, can the authors can use two approaches- i. compare the expression profiles of tail Sox1+ cells to bona fide stem cells of the resident/other tissues, and repeat the venn diagram. Furthermore, regress the cell cycle genes from the expression profiles and regenerate the correlation map; ii. Post the low pH treatment expose the Sox1+ cells to medium favoring the differentiation into cell types of different origin. If other cell types are generated, an alternate hypothesis emerges- Sox1+ pNSCs are more akin to resident stem cells with a large degree of differentiation potential. The authors might appreciate that this hypothesis is not ruled out, as all experiments, including the transplantation heavily, if not exclusively, drive the “pNSCs” to a neuronal fate.

Response: We thank the reviewer for the interesting suggestion that pNSCs may represent biological “noise” cells or are other stem-like cells. Although the concern is valid, our data provide strong evidence that pNSCs are not “noise” cells and are true NSCs rather than other stem-like cells.

First, our new Sox1-CreERT2-R26-EYFP *in vivo* tracing results showed that pNSCs continuously and markedly contribute to neurogenesis in many organs. In particular, we observed a high neuronal contribution of pNSC progeny in DRG and intestinal tissues (Fig. 8h; Extended Data Fig. 15a and 16a). Even after birth, pNSCs made significant neuronal contributions in the DRG and intestine. If the pNSCs are “noise” cells that have escaped the canonical developmental pathways, we would expect a much smaller amount of these cells in the periphery, as the reviewer suggests. Moreover, these cells should bring forth little, if not no, developmental contribution, especially on such a wide and long-term scale across whole-body tissues and complete developmental trajectories. Therefore, the significant continuous and extensive neurogenesis from pNSCs during fetal and postnatal development, particularly the high neuronal contribution of pNSC progenies in DRG and intestine tissues, strongly argues against the expressed concern that pNSCs are simply “noise” cells. We have added this discussion to manuscript (line 697-718).

Second, if the pNSCs are “noise” cells and escape the canonical developmental steps, wouldn’t one expect such “noise” cells to be randomly distributed in the periphery? However, from the pNSC distributions in the lung and tail, we were able to show that the pNSCs are not distributed randomly, but in the form of clusters at specific locations. In the lung, pNSCs are distributed in the epithelial cell wall of the large bronchi. Even more interesting is that the pNSCs in the tail are not only distributed at a specific location, but also form a tubular structure (Fig. 6e and f). Therefore, the specific distributions and structure formation of pNSCs also indicate that pNSCs are not “noise” cells, but are a specific and integral part of mouse embryonic and postnatal development. We have added this discussion to manuscript (line 697-718).

Third, if the similarity between brain NSCs and pNSCs is based on the fact that both are stem-like cells, the conserved marker genes between brain NSCs and pNSCs would consist exclusively of genes related to stem cell maintenance, proliferation of cell populations, and cell cycle. However, analysis of the overlapping genes between brain NSCs and pNSCs shows that these genes fall under GO terms related to nervous system development and neurogenesis, which is consistent with our conclusion that pNSCs represent a group of peripheral neural stem cells. The overlap is not due to the fact that both cell groups are stem cells, but rather that both groups represent neural stem cells with very similar molecular profiles. We have included the GO analysis of the overlapping marker genes with the top 15 biological processes below:

Figure: Top 15 Gene Ontology (GO) terms of overlapping marker genes in different clusters.

As can be seen, the pathways involved are nervous system development, neuron differentiation, brain development, regulation of neuron projection development, axonogenesis, neuron migration, dendrite morphogenesis, and translation at postsynapse/presynapse. Only in the aNSC1 cluster, the conserved marker genes are more related to cell cycle and cell division genes, which is consistent with the active state of active NSCs. We have added these results to the manuscript (line 473-484; line 697-718).

Fourth, as suggested by the Reviewer, we compared the expression profiles of tail Sox1⁺ cells to bona fide tissue stem cells: lung AT1 and AT2 stem cells and embryonic hair follicle epithelium stem cells (HFESCs) (Curras-Alonso et al., 2023; Ge et al., 2020). Pairwise-comparison analysis showed that the Pearson correlation coefficients of tail pNSCs and AT1, AT2, HFESCs were quite low (0.3, 0.14, 0.38, respectively), indicating that pNSCs from the tail are not comparable to these tissue stem cells (see figures below). On the other hand, the Pearson correlation coefficients between tail pNSCs and brain NSCs was high (0.79), indicating significant similarity between tail pNSC and brain NSCs. Dendrogram analysis further confirmed that tail pNSCs are closer to brain NSCs but not tissue stem cells (Figures below). Furthermore, our latest version of the manuscript results (Fig. 7h) showed that the Pearson correlation coefficients of qNSC4, aNSC1, and aNSC2 clusters between brain NSCs and tail pNSCs were quite high (over 0.9), demonstrating the significant similarity between tail pNSC and brain NSCs. In addition, our scRNA-seq from lung Sox2-GFP⁺ cells (Fig. 6c) showed that Sox1⁺/Sox2⁺ neural cells were not clustered together with lung club stem cells and basal stem cells at all, demonstrating that lung Sox1⁺/Sox2⁺ pNSCs have no similarity to stem cells from lung tissue. Taken together, all these results show that pNSCs are similar to brain NSCs, but not to other bona fide tissue stem cells. We have added these results to the manuscript (line 512-524; line 697-718).

Figure: Pearson correlation coefficients (left) and Dendrogram analysis (right) of different types of stem cells.

Fifth, if pNSCs represent a random group of stem-like cells, we would expect that the differentiation ability of pNSCs into neural lineages is not the same as that of brain NSCs.

However, our results showed that the differentiation efficiency of pNSCs into neurons, astrocytes, and oligodendrocytes *in vitro* and *in vivo* was exactly the same as those of brain NSCs (Fig. 2d; Fig. 4c and g; Extended Data Fig. 5h; Extended Data Fig. 9i; Extended Data Fig. 12g). These results show that the pNSCs portray similar differentiation capacities as conventional NSCs, strongly arguing against the fact that they are a random group of stem-like cells. We have added this discussion to manuscript (line 697-718).

Sixth, in our Sox1-CreERT2-R26-EYFP *in vivo* tracing experiments, it is worth noting that the pNSCs *in vivo* were not treated with low-pH. These cells are completely endogenous and undergo normal *in vivo* developmental environment without being reprogrammed or pushed into any particular direction by exogenous factors. In this case, most of the GFP⁺ cells in DRG and intestine tissues were Tuj1⁺, indicating that pNSCs mostly differentiate into neurons rather than other types of cells *in vivo*. If pNSCs are other stem cells, we would expect their progenies to encompass a much more random assortment of cells rather than specific neural lineages.

Thus, we trust that our data provide comprehensive and compelling evidence that pNSCs represent a new group of NSCs that reside in the periphery. These cells are not simply “noise” cells that escape the canonical developmental pathway, but NSCs that have important developmental significance in mouse embryonic and postnatal neurogenesis. Specific evidence includes the newly added Sox1-CreERT2-R26-EYFP *in vivo* tracing results, the *in vivo* localization of pNSC clusters in the neonatal and adult mice, the *in vivo* and *in vitro* differentiation results, analysis of overlapping gene signatures between pNSCs and brain NSCs, as well as comparison of pNSCs with other tissue stem cells. We hope that our answer can clarify and allay the Reviewer’s concerns.

2. What intriguing is that the venn diagram presented shows that the tail pNSCs have a very small proportion of exclusively expressed genes. Is the overlap purely cell cycle genes? What pathways do brain NSCs, specially the quiescent ones, express which are not “tuned on” in the pNSCs? Can the authors generate GO analyses of DEGs enriched in brain specific NSCs?

Response: The total number of cells in tail pNSCs for scRNA-Seq is limited and much lower than in brain NSCs, therefore tail pNSCs have a very low proportion of exclusively expressed genes.

As mentioned above, the overlapping genes between pNSCs and brain NSCs are not just cell cycle genes. Instead, we observed a considerable number of neural stem cell markers among the overlapping genes, such as *Sox2*, *Sox1*, *Aldoc*, *Ednrb*, *Sparc*, *Mt1*, *Mt2*, *Ttyh1*, *Ptprz1*, *Id3*, *Sox9*, *Id4*, *ApoE*, *Dcx*, *Sox11*, *Stmn1*, *Stmn2*, *Tubb3*, *Ptma*, *Fabp7*, indicating the NSC property of pNSCs. We have also attached the GO analysis of conserved marker genes between brain NSCs and pNSCs, which shows that the top 15 biological processes are flooded with terms such as nervous system development, neuron differentiation, neuron migration, neuron projection development, brain development, axonogenesis, translation at postsynapse/presynapse. Only in an aNSC1 cluster, the conserved marker genes are more associated with cell cycle and cell division genes - consistent with the active state of active NSCs. Therefore, the conserved marker genes between pNSCs and brain NSCs indicate that pNSCs are similar to brain NSCs mainly due to gene expression for nervous system development rather than genes for general stem cell maintenance, cell cycle and cell proliferation.

As suggested by the Reviewer, we also reviewed the GO terms of brain NSC-specific “turned-on” genes. We found that the top 15 biological processes GO terms mainly consist of

protein translation, metabolic process, regulation of transcription, RNA splicing, and cell cycle/proliferation (Figure below), which may indicate the different status of pNSCs and brain NSCs. Several other GO terms are involved in nervous system development, neuron differentiation and development, which is consistent with the developmental potential of brain NSCs. We have added these results to the manuscript (line 501-507).

Figure: Top 15 Gene Ontology (GO) terms of brain NSC-specific “turn on” marker genes in different clusters.

3. The authors show that neurons are generated, survive and express mature markers and the authors claim that they are functional. Please tone down the claim of functionality, as the authors do not show innervation or functional properties in vivo.

Response: We have followed the Reviewer’s suggestion and toned down the claim of functionality of generated neurons. We change the word “functional” into “mature” in the abstract (line 48), and in the main text (lines 645, 649, 741, 1503, 1516, 1730, 1737, 1743).

4. Can the authors provide control images for Figure 8H?

Response: We have provided control images (new Extended Data Fig. 16d) for Figure 8h in our last revised manuscript.

5. Can the authors change the colour scheme in Figure 4d-f? It is very hard to separate red and magenta.

Response: We have followed the Reviewer's suggestion and changed the colour scheme in Figure 4d-f.

REVIEWER #4

In this manuscript, Han et al. have presented interesting results that the peripheral neural stem cells (pNSCs) exist which are found from the process of validating the stimulus-triggered acquisition of pluripotency (STAP) based on low-pH treatment of somatic cells. The authors provide the following evidence to prove the existence and the characteristics of pNSCs. First, the pNSCs can be obtained from the limbs and adult lung tissues by low-pH treatment which have the properties of CNS NSC. Second, the pNSCs can be directly isolated from tissue outside of CNS in NSC culture medium without low-pH treatment. Third, the evidence of distinct properties of pNSCs between neural crest cells (NCCs) and the non-NCCs derivation of pNSCs which are proved by lineage tracing method with multiple Cre mice strains are presented. Fourth, the results of scRNA-seq are also give the convincing evidence of the similarity between CNS NSCs and pNSCs.

Comments on authors responses to referee 3:

The most questions raised by referee 3 and referee 1 have been addressed by the latest version of manuscript, and the lineage tracing data with Sox1-CreERT2 is convincing. However, the lineage tracing by Wnt1-Cre is still the doubtful point. Can author provide any evidences to prove the NCCs and NECs are mutually exclusive? In other words, can any evidence prove that Cre⁺ cells in Wnt1-Cre mice are Sox1⁻ cells at embryonic or postnatal stages? Nevertheless, I support the publication of the paper.

Response: We thank the Reviewer for supporting our manuscript for publication. The *Wnt1-Cre* mouse transgenic line is widely used in combination with various Cre reporters for lineage tracing studies of neural crest cells and the midbrain. Its utility is based on the fact that the *Wnt1* gene is highly expressed in the dorsal neural tube prior to neural crest emigration and at the midbrain-hindbrain boundary (MHB) at the beginning of midbrain development (Brewer et al., 2004; Lewis et al., 2013). In *Wnt1-Cre* mice, Cre⁺ cells represent two kinds of cells: neural crest cells and midbrain cells. In the midbrain, many cells would be double-positive for Wnt1-Cre⁺ and Sox1⁺. Therefore, in *Wnt1-Cre* mice, the Cre⁺ cells do not completely exclude Sox1⁺ cells.

In *Wnt1-Cre* mice, Cre⁺ does label midbrain cells in tandem, but here is why we are suggesting that pNSCs are not derived from neural crest cells: We utilized this mouse line because Wnt1-Cre⁺ would theoretically label all neural crest cells. Our lineage tracing experiments have shown that pNSCs are not derived from Wnt1-Cre⁺ labelled cells. Because neural crest cells are contained in Wnt1-Cre⁺ cells, pNSCs are not derived from neural crest cells. In other words, our conclusion is that pNSCs are not derived from neural crest cells and midbrain cells. We think this is an exciting new finding of great interest. We have added this discussion to manuscript (line 676-690).

References:

Brewer, S., Feng, W., Huang, J., Sullivan, S., Williams, T. Wnt1-Cre-mediated deletion of AP-2alpha causes multiple neural crest-related defects. *Dev. Biol.* **267**, 135-152 (2004).

Curras-Alonso, S., et al. An interactive murine single-cell atlas of the lung responses to radiation injury. *Nat. Commun.* **14**, 2445 (2023).

Ge, W., Tan, S. J., Wang, S. H., Li, L., Sun, X. F., Shen, W., Wang, X. Single-cell Transcriptome profiling reveals dermal and epithelial cell fate decisions during embryonic hair follicle development. *Theranostics*, **10**, 7581-7598 (2020).

Lewis, A. E., Vasudevan, H. N., O'Neill, A. K., Soriano, P., Bush, J. O. The widely used Wnt1-Cre transgene causes developmental phenotypes by ectopic activation of Wnt signaling. *Dev. Biol.* **379**, 229-234 (2013).